# A Revised Phylogenetic Classification for *Viola* (Violaceae)

**DOI:** 10.3390/plants11172224

**Published:** 2022-08-27

**Authors:** Thomas Marcussen, Harvey E. Ballard, Jiří Danihelka, Ana R. Flores, Marcela V. Nicola, John M. Watson

**Affiliations:** 1Department of Biosciences, Centre for Ecological and Evolutionary Synthesis (CEES), University of Oslo, P.O. Box 1066 Blindern, NO-0316 Oslo, Norway; 2Department of Environmental and Plant Biology, Ohio University, Athens, OH 45701, USA; 3Department of Botany and Zoology, Masaryk University, Kotlářská 2, CZ-61137 Brno, Czech Republic; 4Czech Academy of Sciences, Institute of Botany, Zámek 1, CZ-252 43 Průhonice, Czech Republic; 5Independent Researcher, Casilla 161, Los Andes 2100412, Chile; 6Instituto de Botánica Darwinion (IBODA, CONICET-ANCEFN), Labardén 200, Casilla de Correo 22, San Isidro, Buenos Aires B1642HYD, Argentina

**Keywords:** *Viola*, Violaceae, taxonomic revision, nomenclature, fossils, morphology, phylogeny, monophyletic, polyploidy

## Abstract

The genus *Viola* (Violaceae) is among the 40–50 largest genera among angiosperms, yet its taxonomy has not been revised for nearly a century. In the most recent revision, by Wilhelm Becker in 1925, the then-known 400 species were distributed among 14 sections and numerous unranked groups. Here, we provide an updated, comprehensive classification of the genus, based on data from phylogeny, morphology, chromosome counts, and ploidy, and based on modern principles of monophyly. The revision is presented as an annotated global checklist of accepted species of *Viola*, an updated multigene phylogenetic network and an ITS phylogeny with denser taxon sampling, a brief summary of the taxonomic changes from Becker’s classification and their justification, a morphological binary key to the accepted subgenera, sections and subsections, and an account of each infrageneric subdivision with justifications for delimitation and rank including a description, a list of apomorphies, molecular phylogenies where possible or relevant, a distribution map, and a list of included species. We distribute the 664 species accepted by us into 2 subgenera, 31 sections, and 20 subsections. We erect one new subgenus of *Viola* (subg. *Neoandinium*, a replacement name for the illegitimate subg. *Andinium*), six new sections (sect. *Abyssinium*, sect. *Himalayum*, sect. *Melvio*, sect. *Nematocaulon*, sect. *Spathulidium*, sect. *Xanthidium*), and seven new subsections (subsect. *Australasiaticae*, subsect. *Bulbosae*, subsect. *Clausenianae*, subsect. *Cleistogamae*, subsect. *Dispares*, subsect. *Formosanae*, subsect. *Pseudorupestres*). Evolution within the genus is discussed in light of biogeography, the fossil record, morphology, and particular traits. *Viola* is among very few temperate and widespread genera that originated in South America. The biggest identified knowledge gaps for *Viola* concern the South American taxa, for which basic knowledge from phylogeny, chromosome counts, and fossil data is virtually absent. *Viola* has also never been subject to comprehensive anatomical study. Studies into seed anatomy and morphology are required to understand the fossil record of the genus.

## 1. Introduction

*Viola* L. is one of the largest angiosperm genera but has not been subject to taxonomic revision for nearly a century [1]. The genus comprises violets and pansies and is one of two temperate genera in the otherwise neotropical Violaceae Batsch family [2,3,4], besides *Cubelium* Raf. ex Britton & A. Br. for *C. concolor* (T. F. Forst.) Raf. ex Britton & A. Br. With its c. 664 species, *Viola* is the largest genus in the family, the fourth largest within Malpighiales (after *Euphorbia* with 2400 species, *Croton* with at least 1300 species, and *Phyllanthus* with 1200 species [5]) and among the 40–50 largest among angiosperms, despite not being among the genera listed by Frodin [6]. *Viola* is one of very few Malpighiales genera with large radiations in the temperate zone, next to *Hypericum* L., *Linum* L., *Salix* L. and *Populus* L.

Violets and pansies are well-known plants and have a long history in European folklore and the first records describing the use of *Viola* in Europe are from Ancient Greece [7]. Fragrant violets were sold in the Athenian agora, praised by Greek poets, such as in the writings of Sappho, used in medicine, had an active role in myths, such as in the abduction of Persephone, were used in garlands, and were present in The Odyssey’s garden of Calypso [7]. *Viola* continued to be used throughout the Middle Ages and species like *V. odorata* (Figure 1), *V. elatior*, and *V. tricolor* were described as medicinal plants in early modern period herbals (e.g., Matthiolus 1562 [8]). In Renaissance paintings and Christian traditions, violets were commonly associated with the Virgin Mary and had a symbolic meaning connected with humility [7].

Dried flowering shoots of *Viola arvensis* and *V. tricolor* are included in the European Pharmacopoeia as Violae tricoloris herba cum flore [9]. They are used as comminuted herbal substances for infusions for cutaneous and internal use, mainly in the treatment of various skin disorders. *Viola* and Violaceae in general are rich in cyclotides, a family of cyclic plant peptides involved in host defence (e.g., [10,11,12]). Given the chemical stability of the cyclotide framework, there is interest in using these peptides as scaffolds in drug design [13], and many species of *Viola* have been screened (e.g., [10,14,15]). *Viola odorata* (Figure 1) in particular has been cultivated for the production of essential oil for the perfume industry [16,17] but nowadays the fragrant compound, ionone, is usually synthesised chemically or endogenously from β-carotene [18]. From the leaves of the same species, absolutes with scent with floral and green notes, reminiscent of cucumber, are extracted and used in the perfume industry [17]. Several species of *Viola* are grown as ornamentals, such as the pansy hybrids *V*. ×*williamsii* and *V*. ×*wittrockiana* [19], and certain cultivars of *V. sororia*, *V. palmata* and *V. prionantha* for their floral display. Others are grown for their colourful or variegated decorative foliage, such as *V. variegata* and *V. riviniana f. purpurea* (often as *V. labradorica* hort. non Schrank). Some are grown for their fragrant flowers, such as *V. odorata*, filled forms of *V. alba* subsp. *dehnhardtii* known as ‘Parma violets’ [7,20], and *V. suavis* [21,22]. Pansy flowers have been used as garnishes on salads and cakes. Since ancient times the petals of blue- or purple-flowered species have been used to make syrups and jellies, and the young leaves of various species have been boiled as a vegetable [23]. *Viola sororia* is the state flower of the USA states of Illinois, Rhode Island, New Jersey, and Wisconsin. In Canada, *V. cucullata* is the provincial flower of New Brunswick. In the United Kingdom, *V. riviniana* is the county flower of Lincolnshire.

All phylogenetic studies to date have recovered *Viola* as monophyletic [3,4,24]. Unlike most other genera in the family, *Viola* is usually herbaceous and with a temperate distribution and is defined by several apomorphies with few exceptions, including the non-articulated peduncles (i.e., lacking an abscission zone at the level of bracteoles), solitary flowers, calycine appendages, bottom petal that is distinctly spurred (rarely scarcely saccate or gibbose), and with the blade shorter than to not much longer than the lateral and upper petals [25]. The spurred bottom petal is a shared feature with its sister lineage, the monotypic shrubby genera *Noisettia* and *Schweiggeria*, but this character is not unique within the family [3,25]. Cleistogamy is widespread and common in the genus (as well as in the family), and many of the lineages in the northern hemisphere have evolved seasonal cleistogamy [26,27].

*Viola* is distributed in most ice-free regions of the world except Antarctica, mainly in the temperate zones of both hemispheres and at high elevations in the mountain systems of the tropics [2,28] (Figure 2). The genus has its centres of taxonomic and morphological diversity in the Andes, in the Mediterranean area of Europe, in eastern Asia, and in North America. Three species, i.e., *V. biflora*, *V. suecica*, and *V. selkirkii*, have nearly circumboreal distributions. *Viola rostrata* is disjunctly distributed in Japan and eastern North America. *Viola palustris* is Amphiatlantic. *Viola arvensis*, *V. odorata*, and *V. tricolor* are near cosmopolites as a result of introductions.

*Viola*, like Violaceae as a whole, is assumed to have originated in South America [2,4,28,29]. Dating analysis associates the origin and beginning diversification of *Viola* with the Eocene-Oligocene cooling event [30,31,32] which, in combination with the formation of the Andes during the Eocene [33,34,35,36], may have given this temperate lineage opportunities to diversify [4,28].

An inherent feature of *Viola* is the lack of barriers against hybridisation, which occurs commonly between closely related species, especially in disturbed or transitional habitats, and which can make species identification difficult [37,38,39,40]. Speciation by allopolyploidisation, which occurs as a consequence of genome duplication in a hybrid, has been estimated to occur with a higher proportion in *Viola* (67% to 88% [28]) than in angiosperms in general (15% to 30% [41,42]). It is therefore no coincidence that the first polyploid series of chromosome numbers was discovered in *Viola*, with *n* = 6, 12, 18, 24, 36, 48 (Miyaji 1913 [43,44]). Allopolyploidisation has been instrumental in at least three major radiations within the genus, i.e., the first following dispersal into the northern hemisphere 18–20 Ma ago and the associated diversification into at least nine allopolyploid endemic lineages [28], the second following dispersal into North America c. 10 Ma ago and formation of the endemic allodecaploid sect. *Nosphinium* [45], and the third since c. 10 Ma within sect. *Melanium* in the western Palearctic [28].

The first taxonomic treatments of *Viola* were contributed by Frédéric C. J. Gingins de la Sarraz (1790–1873) in 1823 [46] and in the chapter on Violarieae in de Candolle’s Prodromus in 1824 [47]. Gingins realised that the shape of the style was a variable and reliable character to subdivide the genus, and based on that he grouped the 105 species known at the time into five sections, sect. *Nomimium* (=sect. *Viola*), sect. *Dischidium*, sect. *Chamaemelanium*, sect. *Melanium*, and sect. *Leptidium*. All but the last section covered the northern hemisphere taxa.

By the end of the 19th century, the number of known *Viola* species had doubled to 200. The treatment of *Viola* by Karl Reiche (1860–1922) for the first edition of Engler & Prantl’s Die Natürlichen Pflanzenfamilien [48] was the first to take into account the morphological distinction of the rosulate violets of South America (subg. *Neoandinium* in our circumscription). Reiche placed them in sect. *Rosulatae*, while uniting all of Gingins’ sections in sect. *Sparsifoliae* (subg. *Viola* in our circumscription). In addition, he erected sect. *Confertae* for five morphologically deviating species of both subgenera.

The treatment of *Viola* by Wilhelm Becker (1874–1928) for the second edition of Engler’s Die Natürlichen Pflanzenfamilien in 1925 [1] represented a leap forward in the understanding and classification of the genus, for which c. 400 species were known at the time. Summarising more than two decades of his taxonomic work on *Viola*, Becker recognised a total of 14 sections based on general morphology and biogeography, including five of Gingins’s [46] but, for some reason, none of Reiche’s [48]. Hence, Becker erected sect. *Delphiniopsis*, sect. *Nosphinium*, sect. *Sclerosium*, and sect. *Xylinosium* for northern hemisphere taxa, sect. *Andinium* (an illegitimate name for Reiche’s sect. *Rosulatae*), sect. *Chilenium*, sect. *Rubellium*, and sect. *Tridens* for South American taxa, and sect. *Erpetion* for the Australian taxa. In addition, he noted the need for additional sections to accommodate a few more, divergent species not included in his system, namely *V. abyssinica* and relatives in Africa, *V. filicaulis* in New Zealand, and *V. papuana* in New Guinea. Notably, Becker subdivided the large and heterogeneous sect. *Nomimium* (=sect. *Viola*) into a total of 17 unranked greges, denoted A through R, many of which have since been combined at the subsection or section level.

The treatment of *Viola* by Wilhelm Becker (1874–1928) for the second edition of Engler’s Die Natürlichen Pflanzenfamilien in 1925 [1] represented a leap forward in the understanding and classification of the genus, for which c. 400 species were known at the time. Summarising more than two decades of his taxonomic work on *Viola*, Becker recognised a total of 14 sections based on general morphology and biogeography, including five of Gingins’s [46] but, for some reason, none of Reiche’s [48]. Hence, Becker erected sect. *Delphiniopsis*, sect. *Nosphinium*, sect. *Sclerosium*, and sect. *Xylinosium* for northern hemisphere taxa, sect. *Andinium* (an illegitimate name for Reiche’s sect. *Rosulatae*), sect. *Chilenium*, sect. *Rubellium*, and sect. *Tridens* for South American taxa, and sect. *Erpetion* for the Australian taxa. In addition, he noted the need for additional sections to accommodate a few more, divergent species not included in his system, namely *V. abyssinica* and relatives in Africa, *V. filicaulis* in New Zealand, and *V. papuana* in New Guinea. Notably, Becker subdivided the large and heterogeneous sect. *Nomimium* (=sect. *Viola*) into a total of 17 unranked greges, denoted A through R, many of which have since been combined at the subsection or section level.

Becker’s taxonomic treatment from 1925 [1] remains the last comprehensive taxonomic treatment of *Viola*. Although comprehensive, it was only a summary, with very short descriptions of infrageneric taxa only and incomplete lists of taxa. Becker probably considered this treatment provisional, as it is known that he was working on a monograph of the genus when he died after a short illness in 1928, aged only 54 [49,50]. His notes were lost and never published. His *Viola* herbarium, containing approximately 4300 specimens and acquired by the Herbarium berolinense (B) in 1929, was destroyed by a fire in early March 1943 after a bombing by Allied forces [51,52]. These unfortunate events, along with the mere size of *Viola* which renders the genus difficult to study in its entirety, are likely reasons why *Viola* has not been subject to full revision in nearly a century. 

In the late 1920s and early 1930s, numerous studies on chromosome cytology were published on *Viola* in the northern hemisphere [29,43,44,53,54,55,56]. Based on these findings, along with observations on general morphology, biogeography, and crossing experiments [57,58], Jens C. Clausen (1891–1969) suggested two considerable changes to Becker’s system [29,56,59]. The first was introducing the concept of a widely defined sect. *Chamaemelanium* that united all yellow-flowered taxa having the base chromosome number *x* = 6, i.e., including sect. *Dischidium* and greges *Orbiculares* and *Memorabiles* of sect. *Nomimium*. The second change was splitting in two the large and heterogeneous sect. *Nomimium*, i.e., into sect. *Plagiostigma*, having a margined style and the base chromosome number *x* = 12, and sect. *Rostellatae* (=sect. *Viola*), having an unmargined, rostellate style and *x* = 10. Although this subdivision was backed up by substantial evidence and later also confirmed phylogenetically, Clausen’s revision was not implemented in any treatment of the genus for the next 90 years [2,28,45,60,61].

Only a few monographs have been published dealing comprehensively with particular groups, i.e., on sect. *Chilenium* [62,63], sect. *Melanium* [64,65,66], subsect. *Borealiamericanae* [67], and most recently on subg. *Neoandinium* [68]. The remaining major post-Beckerian taxonomic treatments of *Viola* by specialists have been regional, e.g., for North America [69,70], Peru [71], the former Soviet Union and Russia [21,61,72], Europe [73], Malesia [74], China and Taiwan [75,76,77,78], Iran and parts of adjacent countries [79], Norden [19], and Argentina [80]. In general, the Russian and Asian taxonomic treatments have combined Becker’s sections at the subgenus level and used higher taxonomic ranks for all the infrageneric groups of *Viola*. There is currently no taxonomic consensus.

Of the numerous phylogenetic studies that have been published for *Viola* [2,28,45,60,81,82,83,84,85,86,87,88,89,90,91,92,93,94] only two have been near-comprehensive in terms of sampling of infrageneric groups [2,28]. The ITS phylogeny of Ballard et al. [2] was the first phylogeny for *Viola* and covered eight of Becker’s 14 sections. The species-level phylogenies of Marcussen et al. [28,45] covered all of Becker’s sections, and based on three low-copy genes and a chloroplast marker, allowed also for the reconstruction of reticulate, allopolyploid history of the genus. Among other things, the phylogenetic findings lended support to Clausen’s [29,56,59] suggestions for a re-circumscription of the large and heterogeneous sect. *Nomimium* and to Reiche’s [48] early suggestion to recognise the South American rosulate violets at a higher taxonomic level. In addition, numerous new infrageneric segregates have been identified (or confirmed) in recent years that require taxonomic recognition, i.e., *V. abyssinica* and relatives and *V. decumbens* in Africa [28], the recently discovered *V. hybanthoides* in China, which has been assigned to the monotypic sect. *Danxiaviola* [90], *V. kunawurensis* for which a reference genome is on the way (NCBI accession PRJNA805692, as *V*. “*kunawarensis*”), and *V. spathulata* and relatives [28] in Eurasia, and a large clade of North American and Hawaiian allodecaploids provisionally referred to as sect. *Nosphinium* s.lat. [2,45,81].

In summary, the knowledge that has been accumulating for nearly a century, since the last revision of *Viola* by Becker in 1925 [1], has not been revised and systematised. This has beyond doubt hindered the testing of new hypotheses and obtaining new knowledge. Since the last revision in 1925, the number of known species in *Viola* has increased by 60% and numerous new infrageneric segregates have been identified using molecular methods and morphology. Among the amended classifications that do exist, no consensus exists for use of rank, delimitation, or nomenclature, mostly because each of these classifications covered only a small part of the genus and taxon delimitation and rank had not been defined in the context of the total variation within *Viola*. Furthermore, none of the hitherto proposed classifications have been phylogenetic by nature and aimed at reconciling taxon monophyly and the extensive reticulate evolution due to allopolyploidy [28] in the genus. Finally, it is now known that a substantial proportion of the known species of *Viola* are narrow endemics and endangered species and are as such at risk of extinction due to human-induced changes in land use and climate [95]. *Viola* (sect. *Melanium*) *cryana*, is considered extinct in Europe and globally [96] and *V.* (sect. *Plagiostigma*) *stoloniflora* is considered extinct in the wild in the Ryukyus Islands [97], and it is to be feared that up to 27 species within subg. *Neoandinium*, most of which have not been seen since the type collection, have become extinct [68].

The aim of this revision was to generate an updated infrageneric taxonomy for *Viola* based on modern principles of phylogenetics and monophyly and the accumulated information since Becker’s previous morphology-based classification from 1925 [1]. The revision is presented as (1) a global checklist of species of *Viola* accepted by us and annotated with infrageneric taxonomy, (2) an updated multigene phylogenetic network and an ITS phylogeny with denser taxon sampling, (3) a brief summary of the taxonomic changes from Becker’s classification and their justification, (4) an account of each infrageneric group with justifications for delimitation and rank including a description, a list of apomorphies, molecular phylogenies where possible or relevant, and a list of accepted species, and (5) a morphological binary key to the accepted subgenera, sections and subsections. It is our intention and hope that this synthesis, by summarising what is known and what remains to be known for *Viola*, will serve as both a foundation and an inspiration for further studies on this large, diverse and insufficiently understood genus.

## 2. Results and Discussion

### 2.1. Phylogeny and Classification

We recognise 664 known species of *Viola*, 43 of which have not yet been described. The global species checklist, annotated with infrageneric taxonomy, is presented in Appendix B. We subdivide the genus into two subgenera, 31 sections, and 20 subsections. Subgenus *Neoandinium* comprises 139 species in 11 sections, and subg. *Viola*, 525 species in 20 sections and 20 subsections (Table 1; Figure 3). Photographs of representatives of each accepted infrageneric segregate are shown in Figure 3; these images are also presented in a downloadable and printable poster in Appendix A. Section *Plagiostigma* is by far the most species-rich section with 142 species, followed by sect. *Melanium* with 110 species. Nearly half of the sections, 15 of 31, include three species or less (Figure 4). We propose subg. *Neoandinium* as a replacement name for the illegitimate subg. *Andinium* (W. Becker) Marcussen, and erect 13 new infrageneric taxa within subg. *Viola*, i.e., six new sections (sect. *Abyssinium*, sect. *Himalayum*, sect. *Melvio*, sect. *Nematocaulon*, sect. *Spathulidium*, and sect. *Xanthidium*), and seven new subsections (subsects. *Australasiaticae*, *Bulbosae*, and *Formosanae* within sect. *Plagiostigma*, subsect. *Clausenianae* within sect. *Nosphinium*, and subsects. *Cleistogamae*, *Dispares*, and *Pseudorupestres* within sect. *Melanium*). Justifications for erecting new taxa are given under each taxon in the taxonomic section and in the form of a binary key (Chapter 5).

#### 2.1.1. Genus Phylogeny

We updated the allopolyploid phylogenetic network obtained by Marcussen et al. (2015 [28]), based on homoeologs of three low-copy nuclear genes, with new information on chromosome counts and sequences (Figure 5). A dated phylogeny of the ITS marker, with denser sampling for selected taxa, is shown in Figure 6. New ITS sequences provided a new and older crown node age for subg. *Neoandinium* (c. 20.3 Ma) compared to Marcussen et al. [28], and also allowed placing the two novel sections *Danxiaviola* [90] and *Himalayum* as distinct lineages within the North Hemisphere CHAM + MELVIO allopolyploid tangle in Figure 5. We have re-evaluated the phylogenetic placement of *Viola* (sect. *Melvio*) *decumbens* (Figure 5), after discarding the erroneous *trnL-trnF* sequence that placed it next to *V. arborescens* in sect. *Xylinosium* [28]. *Viola decumbens* appears to be hexaploid, as each of the three low-copy genes analysed by [28] has three MELVIO homoeologs that coalesce around 17–22 Ma. These homoeologs coalesce slightly shallower (on average 1.6 Ma) with one another than with the rest of the MELVIO clade, suggesting that the subgenomes of *V. decumbens* constitute a monophyletic sister to the rest of the MELVIO lineage. No chromosome counts exist for *V. decumbens*. The updated and corrected chromosome counts on *V*. (sect. *Erpetion*) *banksii* (2*n* = 50, not 60) and *V*. (sect. *Tridens*) *tridentata* (2*n* = 40, not 80) are reconcilable with the molecular data without the need to formulate complex hypotheses of homoeolog loss and duplication (cf. [28]). Both homoeolog number and chromosome count for sect. *Erpetion* indicate that this lineage is allo-octoploid (Figure 5); the recent count of 2*n* = 50 in *V. banksii* [98] is very close indeed to the expected 2*n* = 48 based on *x* = 6 in the diploid ancestor of sects. *Chamaemelanium* and *Rubellium*. Similarly, for sect. *Tridens* both homoeolog number and chromosome count agree with allohexaploidy (Figure 5); the count of 2*n* = 40 [99] is very close to the expected 2*n* = 38 based on *x* = 6 in the two genomes shared with sect. *Erpetion* and *x* = 7 in the one shared with sect. *Leptidium*.

#### 2.1.2. Justification for Taxonomic Levels and Classification

The phylogenetic history of *Viola* is reticulated to such an extent that monophyletic groups can be delimited at three hierarchical levels only (Figure 5). The highest hierarchical level corresponds to subgenus in our treatment and delimitates two monophyletic taxa, i.e., subg. *Neoandinium* and subg. *Viola*. The intermediate hierarchical level corresponds to the section level. The lowest hierarchical level delimits subsections. Below the level of subsection, taxa are interconnected by allopolyploidy and the taxonomic level of series is not applicable as a result of non-monophyly. In addition, we have chosen to apply the levels of subgenus, section and subsection because this use maximises taxonomic stability by minimising the number of changes from Becker’s [1] treatment and by allowing us to keep most of his sections.

The alternative to treating *Neoandinium* at the subgenus level would be to recognise it as a separate genus (e.g., as *Andinium*). This could have been justified both morphologically and phylogenetically. However, this change would be phylogenetically unnecessary, as monophyly is not affected, and there is also no need for additional taxonomic levels within *Viola*, considering that we here abandon the taxonomic level of series for reasons of monophyly. Recognising a separate genus for subg. *Neoandinium* would further disrupt taxonomic stability and require numerous new taxonomic combinations to be made.

In our taxonomic treatment, sect. *Nosphinium* is the only exception to the rule of strict monophyly, which cannot be enforced due to the conceptual conflict between reticulate evolution, as a result of allopolyploidy, and the hierarchical system of classification. This conflict occurs because sect. *Nosphinium* is an allodecaploid lineage that originated by hybridisation between taxa deeply nested within the sections *Chamaemelanium*, *Plagiostigma*, and *Viola*, and that during its diversification acquired several additional *Plagiostigma* genomes by further allopolyploidisation [45]. Enforcing strict monophyly in this case would, by a domino effect, have the undesirable consequence that all sections within subg. *Viola* were rendered non-monophyletic.

#### 2.1.3. Changes to Becker’s Original System for *Viola*

The comprehensive classification of *Viola* presented here is the first since that proposed by Becker [1] nearly a century ago. Changes in classification from Becker’s system are summarised and displayed as a “wire” diagram in Figure 7. We give justifications for these changes under each taxon in Chapter 2.4. (Taxonomic Treatment of *Viola*).

Becker [1] recognised 14 sections and numerous infrasectional greges within *Viola*. Here, we suggest recognising two subgenera, subg. *Neoandinium* (Becker [1]: sect. *Andinium*) with 11 sections and subg. *Viola* with 20 sections and 18 subsections. Recently, Watson et al. [68] proposed a provisional classification of subg. *Neoandinium* (as subg. *Andinium*) with 11 sections based on general morphology. In the absence of phylogenetic data and a good understanding of character polarity in the two subgenera, we tentatively follow this classification. Within subg. *Viola*, we make the largest changes in circumscription to Becker’s sections *Nomimium*, *Dischidium*, *Nosphinium*, and *Chamaemelanium*, where Becker’s [1] species groups are now re-distributed among six sections. Section *Chamaemelanium* now comprises the former sect. *Dischidium* and greges *Memorabiles* and *Orbiculares* of sect. *Nomimium*. Section *Viola* corresponds to the former sect. *Nomimium* s.str. and unites greges *Repentes*, *Umbraticolae* and *Rostratae* in subsect. *Rostratae*, and greges *Uncinatae*, *Lignosae* and *Serpentes* (pro parte) in subsect. *Viola*. Section *Plagiostigma* unites greges *Serpentes* pro parte, *Vaginatae*, *Bilobatae* and *Stolonosae* in subsect. *Australasiaticae* and subsect. *Stolonosae*, retains grex *Diffusae* as subsect. *Diffusae* and retains most of grex *Adnatae* as subsect. *Patellares*. Section *Nosphinium*, which in the original sense comprised the Hawaiian *Viola* only, is here considerably expanded and comprises subsect. *Borealiamericanae*, subsect. *Mexicanae*, subsect. *Pedatae*, and subsect. *Langsdorffianae* (all previously greges of sect. *Nomimium*), next to subsect. *Nosphinium* (Becker [1]: sect. *Nosphinium*) and subsect. *Clausenianae*.

In subg. *Viola* six new sections have been erected to accommodate the following taxa: sect. *Abyssinium* for the African species *V. abyssinica* and allies (Becker [1]: mentioned but not formally classified); sect. *Himalayum* for *V. kunawarensis* in the Himalayas (Becker [1]: sect. *Nomimium* grex *Adnatae*); sect. *Melvio* for the South African Cape endemic *V. decumbens* (Becker [1]: sect. *Xylinosium*); sect. *Nematocaulon* for the New Zealand endemic *V. filicaulis* (Becker [1]: mentioned but not formally classified); sect. *Spathulidium* for *V. spathulata* and allies in southwestern Asia (Becker [1]: sect. *Nomimium* grex *Adnatae*); and sect. *Xanthidium* for the *V. flavicans* group in southern South America (Becker: not included in the monograph [1] but mentioned elsewhere [100,101,102]). Section *Danxiaviola* has already been published to accommodate the newly described *V. hybanthoides* endemic to Yunnan, China [90]. These six new sections comprise in total about 11 species only, indicating that Becker’s [1] century-old classification provided a remarkably good overview of the genus.

### 2.2. Patterns of Evolution within Viola

#### 2.2.1. Historical Biogeography of *Viola*

We reconstructed the historical biogeography of *Viola* (Figure 8) using a simplified approach based on four biogeographic categories, a single-rate transition model, and 50 operational taxonomic units as defined in the diploid multilabelled phylogenetic timetree that is the counterpart of the phylogenetic allopolyploid network in Figure 5. Our result gives the strongest possible support (*pp* = 1.0) to the previously proposed, but never actually tested, hypothesis that *Viola* originated in South America [2,28,29]. Subgenus *Neoandinium* has remained within the ancestral range in South America. Within subg. *Viola*, it is inferred that the CHAM and MELVIO lineages dispersed independently into the Northern Hemisphere 20–25 Ma ago where they eventually met and formed allopolyploids. Intersectional biogeographic relationships within the Northern Hemisphere are not resolvable due to the basal polytomy. However, it seems likely that the diploid CHAM lineage dispersed northwards from South America into North America, where it gave rise to sect. *Chamaemelanium* which at present has its diversity centre along the Pacific coast of North America; this scenario was proposed already by Clausen nearly a century ago [29]. The dispersal history of the diploid MELVIO lineage remains unknown, as it is represented by a single species (*V.* (sect. *Melvio*) *decumbens*) that occurs allopatrically in the Cape of South Africa. It seems clear, however, that members of CHAM and MELVIO both dispersed into Eurasia where they by hybridisation gave rise to numerous allopolyploid lineages, most of which correspond to sections in our treatments. Western Eurasia appears to have been the cradle of early allopolyploid diversification, as the majority of these sections are endemic or have diversity centres here; only three sections have diversity centres in eastern Eurasia (sects. *Danxiaviola*, *Himalayum*, and *Plagiostigma*). Both the ancestral diploids (CHAM and MELVIO) have since become extinct in western Eurasia.

Concerning the several dispersals out of South America, it would not be surprising if *Viola* was shown to have migrated northwards within the continent, following the progressive rise of the Andes and the advent of new alpine habitats, as known for many other taxa. Although the current mountain elevations were reached relatively recently during the Plio-Pleistocene [104,105,106], the southern and central Andes date back to the Cretaceous, and the northern Andes to the Eocene [107,108]. Regarding the dispersal of *Viola* from South America to North America, first, there is evidence that the Isthmus of Panama in Central America was an uninterrupted chain above sea level from the late Eocene until at least the late Miocene [109]. Second, although the distances were presumably too long to allow for direct dispersal between North and South America, comparing these patterns with those involving dispersal to the Pacific Islands, Carlquist [110] suggested that dispersal by birds could account for many of the disjunct distributions between North and South America. Third and finally, even though *Viola* species do not have palatable fruits or seeds, a large number of genera showing disjunctions do not have obvious effective long-distance dispersal mechanisms either. Some birds may eat and thus internally carry fruits or seeds other than those that are big, fleshy, and strikingly coloured [110].

Regardless of mode, *Violas* are apparently quite capable of long-distance dispersal and have successfully colonised remote oceanic islands like Hawaii, New Zealand, and the Azores, the temperate “sky islands” of tropical Africa, and the South African Cape province, or have even dispersed back to southern South America (*V. huidobrii*). A few extant species occur on more than one continent (e.g., *V. biflora*, *V. palustris*, *V. rostrata*, *V. selkirkii*, *V. suecica*).

#### 2.2.2. Hybridisation and Allopolyploidy

Interspecific hybridisation is common in *Viola* and is well studied for the sections in the Northern Hemisphere. Hybridisation occurs most commonly between pairs of closely related species, especially among those that share a genome due to allopolyploidy, such as *V. epipsila* (4*x*) and *V. palustris* (8*x*) and European members of subsect. *Rostratae* (4*x*/8*x*/12*x*) [19,38,39,111,112,113,114,115,116]. As a result, spontaneous hybrids occur nearly exclusively between taxa within the same subsection, more rarely between species belonging in different subsections, and only occasionally between species in different sections. The most phylogenetically distant taxa to form spontaneous hybrids are members of sect. *Plagiostigma* subsect. *Patellares* and sect. *Viola* subsect. *Rostratae*, which are estimated to have diverged some 19 Ma ago (Figure 5). Their hybrids are extremely rare and have been reported from single individuals only, of which *V. japonica* × *V. rostrata* is the only one that has been confirmed by DNA data [61,117,118]. Artificial hybrids are, however, easily made between members of these two sections and also with sect. *Nosphinium* subsect. *Borealiamericanae*, to a lesser degree with sect. *Chamaemelanium* [57,58]. The genomic compatibility of these lineages most likely reflects their comparatively slow evolutionary rates [28].

The symplesiomorphic, retained ability of taxa to interbreed for millions of years after they diverged has evidently played an important role in the phylogenetic history of the genus by allowing for extensive allopolyploid speciation (Figure 5; [28,45,60]. Although historically most allopolyploidisations have involved recently diverged parental taxa, their divergence may have been more than 10 Ma for mesopolyploids such as sect. *Leptidium* and sect. *Tridens* (Figure 5) and widespread neo-octoploids such as *V. blanda*, *V. incognita*, *V. pluviae*, and *V. palustris* [45,93]. All these four neo-octoploids have Boreal distributions and their origins coincide with the climate cooling and repeated glaciations in the last 5 Ma [119]. More than anything, this shows that the ability to hybridise and speciate by allopolyploidisation can be a rapid mode of diversification to fill vacant niches (e.g., [120]).

The association of long-distance dispersal with polyploidy is striking in *Viola*. In each of the seven cases of long-distance dispersals older than a few Ma (Figure 5 and Figure 8), the colonist taxon has a higher ploidy than its sister taxon or, if known, ancestor. This is seen on a massive scale in connection with the colonisation of the Northern Hemisphere by the CHAM and MELVIO lineages, which occurred c. 19 Ma ago and gave rise to more than 400 species [28], and with the decaploidisation that gave rise to sect. *Nosphinium* following independent dispersal to North America of its ancestors in sect. *Plagiostigma* and sect. *Viola*, which occurred c. 10 Ma ago and gave rise to 61 species [45]. The same pattern of increased ploidy in the colonist taxon is seen on a smaller scale for sect. *Erpetion* in Australia within the last 7 Ma (11 species), for subsect. *Nosphinium* in the Hawaiian islands within the last 5 Ma (9 species), for sect. *Abyssinium* in tropical African mountains within the last 5 Ma (3 species), for sect. *Melvio* (i.e., *V. decumbens*) in South Africa possibly 20 Ma ago, and for sect. *Nematocaulon* (i.e., *V. filicaulis*) in New Zealand, age unknown. In the four cases where there is sufficient phylogenetic resolution, polyploidisation seems to have occurred after colonisation (CHAM + MELVIO, sect. *Nosphinium*, sect. *Erpetion*, sect. *Melvio*). This indicates that polyploidy is linked with colonisation rather than dispersal, an association that is general across angiosperms and may reflect that speciation by polyploidy gets to dominate during phases of colonisation because it is a much faster process than homoploid speciation (e.g., [120]).

The phylogenetic network for *Viola* (Figure 5) contains 13 homoploid speciations and 23 allopolyploid speciations, which means that allopolyploidy may have accounted for 64% (=23/(13 + 23)) of the speciations above the section level. This proportion is lower than the estimate of 67–88% by Marcussen et al. [28] as a result of new and re-interpreted information for numerous sections, as well as an expanded set of taxa, but the estimate is still far higher than that for angiosperms as a whole, estimated to 15% [41] or 30% [42].

The reason why polyploidisation is more common in *Viola* than in other lineages is probably linked to the ability to hybridise in combination with cleistogamy. The retained ability for lineages to form hybrids, in some cases up to 15 Ma or more, provides the raw material for allopolyploid formation. Regular selfing through cleistogamy might help the nascent allopolyploid in the early phases of establishment.

#### 2.2.3. Base Chromosome Numbers in *Viola*

The limited number of chromosome counts appears to indicate that *x* = 7 may be the base chromosome number for *Viola* as a whole. The two counts in subg. *Neoandinium* both show 2*n* = 14 [121]. For subg. *Viola*, *x* = 6 was long assumed because 2*n* = 12 is shared by its two diploid sections, *Chamaemelanium* [29,43,44] and sect. *Rubellium* [60]. However, the two deepest lineages of subg. *Viola*, which are now extinct as diploids, may rather have had *x* = 7, which is indicated by ploidy and chromosome counts for the two polyploid sections *Leptidium* (*x* = 6.75, based on 2*n* = 54 [53] and 8*x*; Figure 5) and *Tridens* (*x* = 6.67, based on 2*n* = 40 [99] and 6*x*; Figure 5). The reduction from *x* = 7 to *x* = 6 may therefore be a synapomorphy for the most recent common ancestor of sects. *Chamaemelanium* and *Rubellium*. This hypothesis needs to be tested with additional counts for the South American lineages of *Viola*, and also from the sister genera, *Noisettia* and *Schweiggeria*, for which data are lacking but would be relevant for understanding character polarity.

#### 2.2.4. Morphology, Anatomy, Adaptations

Perhaps the most striking finding in our phylogeny of *Viola* is the lack of a clear correspondence between macromorphology and phylogeny, with the exception of the distinction between subg. *Viola* and subg. *Neoandinium*, and style morphology to some extent (Figure 9). There are two likely causes for this—the highly reticulate phylogeny, which has allowed for the redistribution of apomorphies and adaptations, and the large polytomy at the base of the Northern Hemisphere taxa, which precludes the existence of synapomorphies among these taxa (Figure 8).

Style shape is variable in *Viola* (Figure 9) and has historically been a key character to subdivide the genus [1,29,46,47,48,59,61,68,90,122,130]. While broad diversity of style morphologies have been used previously for limited studies of taxa within subsections or sections of the genus, we sought to greatly expand the sampling to encompass the main “phenotypes” of style morphology within the two subgenera and all sections and subsections, and to evaluate the efficacy of style traits for delimiting higher-level taxa in addition to morphology. We recognised broad types of styles, first as “undifferentiated” (styles cylindrical, often straight, lacking apical ridges or processes and terminating in the stigmatic orifice) and “differentiated” (clavate or capitate, with processes or apical ridges or lobes, the stigmatic orifice on a rostellum). Additional traits were noted, such as the presence/absence of papillae or trichomes; the shape, orientation and fusion of apical ridges or lobes; and the thickness, prolongation and orientation of the rostellum supporting the stigmatic orifice. In subg. *Neoandinium*, the bulk of species display conspicuous and remarkable types of crests and processes, each species often dramatically distinct in these stylar adornments. We speculate that the divergent stylar crests or processes among related species serve a role in pollinator specificity, in a region where the paucity of pollination vectors could drive selection for diverse pollinator behaviours to reduce hybridisation. In subg. *Viola*, the range of style morphologies within some larger sections such as *Chamaemelanium* and *Plagiostigma* is very broad, whereas the range within *Leptidium*, *Melanium*, and *Viola* is generally quite narrow. Variability within subsections is generally quite narrow and readily characterised. In all higher-level taxa (sections and subgenera), the range of style morphologies can be discretely described and used to support the recognition of higher-level taxa based on morphology and chromosome number. In particular, style morphology can provide distinctive apomorphies where certain morphological features may be homoplasious in comparing some higher-level taxa, especially in sect. *Nosphinium* and sect. *Plagiostigma*.

Some of the variations in style morphology are geographically structured and might reflect adaptation to special modes of pollination and/or pollinators. Undifferentiated, filiform styles occur exclusively in tropical-montane and south-temperate taxa, i.e., sect. *Erpetion*, sect. *Leptidium*, sect. *Tridens*, sect. *Nematocaulon*, and in single species within sect. *Chilenium* (*V*. *commersonii*), and sect. *Viola* (*V. papuana*). Trichomatous-bearded styles occur exclusively in north-temperate taxa, i.e., sects. *Chamaemelanium*, *Melanium*, and *Viola*.

Shoot morphology has been given much attention in previous classifications, at least among the herbaceous Northern Hemisphere taxa, notably the presence or absence of leaf rosettes, aerial stems, or stolons. Taxa have accordingly been described and classified as rosulate or arosulate, caulescent or acaulescent, and stolonose or estolonose (e.g., [1,131,132,133,134]). This classification is, however, artificial and does not reflect phylogenetic relationships. In addition, this classification is problematic because of the logical flaw of defining taxa based on the absence of a structure (e.g., acaulescence), and it also eludes the possibility that aerial stems in one “caulescent” taxon could be homologous with stolons in another “acaulescent stolonose” taxon, as otherwise suggested by the intermediate morphology of interspecific hybrids (e.g., *V. canina* × *V. uliginosa*, *V. odorata* × *V. riviniana* [58], *V. epipsila* × *V*. *riviniana*; T.M., unpublished data from crossing experiments). In any case, our data show that shoot morphology is quite labile and that loss, gain, or transitions among character states have occurred repeatedly in the four sections *Nosphinium*, *Plagiostigma*, *Viola*, and *Chamaemelanium* to the extent that it is not possible to infer which state(s) was ancestral; the exception is sect. *Chamaemelanium* where nearly all species have aerial stems and this character state seems to be ancestral. The loss of lateral stems presumably has a simple genetic basis, but these structures appear to be gained almost as easily. For instance, within sect. *Plagiostigma*, aerial stems have been invented from an ancestor that lacked them in subsect. *Diffusae* within the last 3 Ma (*V. guangzhouensis*) and in subsect. *Stolonosae* within the last 5 Ma (*V. moupinensis*). Similarly, stolons have been invented de novo in sect. *Erpetion* within the last 7 Ma. Within sect. *Viola* subsect. *Rostratae* all character states (i.e., aerial stem, stolon, or absence of both) may have evolved within the last 7 Ma.

Another conspicuous characteristic is woodiness. This was most obviously the ancestral character state at the stem node of the genus, given that the sister lineage of *Viola* (*Noisettia* and *Schweiggeria*) and nearly all other genera in Violaceae are woody. However, the most recent common ancestor of *Viola* was probably not a lignose. Shrubby and subshrubby taxa occur scattered throughout the genus, and the fact that shrubbiness is most definitely derived in the taxa of subsect. *Nosphinium*, which arrived in the Hawaiian Islands some 5 Ma ago (see Chapter 5) [45,81,85], indicates that this too is a plastic character. Furthermore, none of the shrubby taxa of *Viola* (except for the Hawaiian ones) have retained the differentiated shoot architecture found in *Noisettia* and *Schweiggeria* as well as woody seed plants in general, with growth axes differentiated in orthotropic vegetative axes and plagiotropic reproductive axes [135].

A suite of characters appears to have evolved in the ancestor of the Northern Hemisphere taxa, perhaps in part as adaptations to increased seasonality as compared to South America. These include a shoot architecture with differentiated growth axes, seasonal cleistogamy, and a bearded style. All three characters are expressed in the diploid sect. *Chamaemelanium* and might therefore be adaptations associated with the ancestral CHAM genome, but they are not expressed in all of the allopolyploids having CHAM and MELVIO genomes. In sect. *Chamaemelanium* shoot differentiation is extreme, with the perennating axis usually being a deep-buried rhizome and lateral stems annual, aerial and floriferous; this differentiation is less extreme, but present in large sections such as *Viola*, *Plagiostigma* and *Nosphinium*. Another character is cleistogamy, which is common in *Viola*. *Viola* has the type of cleistogamy referred to as dimorphic, i.e., the primordial bud is already predetermined to develop into either a chasmogamous or cleistogamous flower [27]. Cleistogamy is facultative in the Southern Hemisphere lineages in sects. *Leptidium*, *Chilenium*, and *Nematocaulon*, and at least in the last two may occur as reproductive assurance under unfavourable conditions [26,136]. Many of the Northern Hemisphere lineages have instead evolved seasonal cleistogamy by which production of flower type is determined by photoperiod and temperature: during long-day conditions, cleistogamous flowers are produced and during short-day conditions, chasmogamous flower buds are produced that remain dormant until the following spring [137,138,139,140,141,142,143,144]. Seasonal cleistogamy is known from sects. *Chamaemelanium*, *Himalayum*, *Melanium*, *Nosphinium*, *Plagiostigma*, and *Viola*.

There have been no comprehensive anatomical studies of *Viola* (cf. [145]), but investigations have been conducted on particular species or groups of species (e.g., [145,146,147,148,149,150,151,152,153,154]). Shoot architecture has been studied for a few European species [155].

Pollen in Violaceae is generally tricolporate [156]. In *Viola*, however, about one third of the species are heteromorphic with regard to pollen aperture number, which has been explained as a consequence of neopolyploidy [157]. Hence, up to five and six apertures occur in the high-polyploids (4*x* to 20*x*) of sect. *Melanium* whereas three and four apertures occur in the other investigated sections [157]. It may be noted that [157] severely underestimated the ploidy of most of the investigated taxa; e.g., the 12-ploid *V. tricolor*, 16-ploid *V. arvensis*, and 18-ploid *V. langsdorffii* were all interpreted to be diploid. Gavrilova & Nikitin [147] found that East European species in the sections *Chamaemelanium*, *Plagiostigma*, and *Viola* have 3–(4)-colp(oroide)ate pollen with long colpa and mostly complex exine ornamentation, while sect. *Melanium* has (3–)4–5(–6)-colporate pollen with shorter colpa and simple exine ornamentation. No palynological data exist on South American members of the genus.

*Viola* comprises numerous metallophytes, i.e., species or populations of species with the capacity to tolerate metal toxicities as well as survive and reproduce on metalliferous soils. While the situation in subg. *Neoandinium* is not known, within subg. *Viola* at least 20 taxa are known to hyperaccumulate heavy metals and other toxic elements (As, Cd, Ni, Pb, Sb, Tl, Zn) [158,159]. Most of these taxa belong in sect. *Melanium*, but all the four largest sections in the northern hemisphere are represented, i.e., sect. *Chamaemelanium* (*V. cuneata*, *V. brevistipulata*), sect. *Melanium* (*V. albanica*, *V. allchariensis*, *V. arsenica*, *V. beckiana*, *V. dukadjinica*, *V. elegantula*, *V. kopaonikensis*, *V. lutea*, *V. raunsiensis*, *V. tricolor*, *V. vourinensis*, *V.* ×*wittrockiana*), sect. *Nosphinium* (*V. howellii*), sect. *Plagiostigma* (*V. baoshanensis*, *V. philippica*, *V. principis*), and sect. *Viola* (*V. sacchalinensis*, *V. kizildaghensis*). Additionally, some members of sect. *Erpetion* (*V. banksii*, *V. serpentinicola*) may well prove to be hyperaccumulators [98,160]. The high prevalence of metallophytes in sect. *Melanium* reflects the general ability of members of this section to adapt to extreme abiotic conditions.

Although no species of subg. *Neoandinium* are known to be hyperaccumulators, it is reasonable to assume that several species are, especially those known from metalliferous soils in the immediate proximity of Andean copper mines in Chile, such as *V*. *escarapela*, *V. exilis*, and *V. gelida* (sect. *Rosulatae*), *V. godoyae* (sect. *Relictium*), *V. uniquissima* (sect. *Triflabellium*), and *V. vallenarensis* (sect. *Subandinium*) [68,161,162]. In fact, *V*. *godoyae* and *V*. *uniquissima* were both discovered during the initial surveys of the immediate areas where their respective mines are located. It is also noteworthy that many *Neoandinium* taxa have an affinity for extreme abiotic conditions in much the same way as sect. *Melanium*.

In Violaceae, hyperaccumulators occur within another four genera outside of *Viola*, i.e., *Afrohybanthus*, *Agatea*, *Pigea*, and *Rinorea* [163,164,165,166,167]. These five genera are phylogenetically scattered and not closely related [3], indicating that the ability of metal hyperaccumulation evolved independently in each lineage, presumably from a set of shared preadaptations. The possible occurrence of hyperaccumulators in both subgenera of *Viola* points to a set of preadaptations being shared across the genus.

#### 2.2.5. Biotic Interactions

The chasmogamous flowers of *Viola* are visited and pollinated primarily by solitary bees (Anthophoridae, Halictidae, Andrenidae; Hymenoptera), but also bumblebees (Apidae, Hymenoptera), hoverflies (Syrphidae, Diptera), bee flies (Bombyliidae, Diptera), butterflies (Lepidoptera), and in some species beetles (Coleoptera) [168,169,170,171,172,173]. Many species appear to be generalists but the degree of specialisation varies considerably among species and even populations and appears to be evolutionary plastic. For instance, among the closely related species of subsect. *Rostratae*, *V. adunca* and *V. striata* are visited mostly by solitary bees, *V. reichenbachiana* mostly by hoverflies, and the longer-spurred *V. rostrata* mostly by bee flies [169]. Populations in natural and disturbed sites can attract widely differ pollinators, as shown for *V.* (sect. *Rubellium*) *portalesia* [173]. Among the most highly specialised are *V.* (sect. *Leptidium*) *arguta* [174] which has a red corolla and a saccate spur and is the only *Viola* pollinated by hummingbirds (Trochilidae, Aves), and the three species of sect. *Delphiniopsis* which are characterised by bright pink corollas with a very long, thin spur and are pollinated by a single species of day-flying hawk-moth (*Macroglossum stellatarum*, Sphingidae, Lepidoptera) [175,176]. 

Spurred, nectar-producing flowers occur also in the sister genera of *Viola*, *Noisettia* and *Schweiggeria*, and must therefore be considered the ancestral state in *Viola*. In certain sections of subg. *Viola* with species adapted to pollination by solitary bees there has been a transition from nectar to pollen as a pollinator reward. Contrary to “nectar flowers”, which produce nectar from the nectariferous appendage of the two bottom anthers and store it in the spur of the bottom petal, these “pollen flowers” do not produce nectar and have reduced anther appendages and petal spur [171,177,178]. This transition has occurred independently in sect. *Leptidium* [171] and sect. *Erpetion* [177,178], but morphology suggests that several other short-spurred groups may have evolved into “pollen” flowers; at least *V. sumatrana* (sect. *Plagiostigma*, subsect. *Australasiaticae*) appears to produce no nectar [171]. In sect. *Leptidium* the bottom pair of stamens have prolonged “u”-shaped anther connective appendages that the female bee (*Anthrenoides*, Andrenidae) holds onto while harvesting pollen by vibration (“buzz-pollination”) [171]; curiously, the only bird-pollinated *Viola* species, *V. arguta*, belongs to the same section and does produce copious amounts of nectar (4 µL per 24 h) [174].

Caterpillars of Nymphalidae butterflies, subfam. Heliconiinae, are specialised to feeding on the foliage of members of the parietal Malpighiales, which includes Violaceae, Passifloraceae, and Salicaceae [179]. *Viola* are frequently predated by the larvae of tribe Argynnini whose diversification in time and space appears to have been tracking that of *Viola*. The ancestral Argynnini used *Passiflora* and Violaceae as larval host plants but specialising on Violaceae appears to have occurred prior to c. 23 Ma on the branch subtending the stem node of *Yramea* [180,181,182] in South America, where *Yramea* occurs today. The crown node of *Viola* is estimated to be just a few Ma older, c. 31 Ma [28]. Dispersal into Eurasia of the common ancestor of *Boloria*, *Issoria*, *Brenthis*, *Argynnis* (Argynnina clade) may have happened around c. 15 Ma ago [181,182,183], which again corresponds well with the appearance of *Viola* seeds in the Eurasian fossil record 17–18 Ma [184,185]. Finally, further dispersal and diversification of *Argynnis* (*Speyera* subclade) into North America is estimated to have occurred c. 5 Ma ago [183,186], apparently tracking the diversification of the decaploid sect. *Nosphinium* lineage [45]. Host switches from *Viola* to other cold-temperate taxa have occurred in *Boloria* (to, e.g., *Dryas*, *Vaccinium*, *Salix)* and in *Brenthis* (to mainly Rosaceae) [180,186].

While tribe Argynnini have larvae that feed mostly on Violaceae, the other lineages within subfam. Heliconiinae have specialised on the closely related Passifloraceae (the clade consisting of tribes Heliconiini and Acraeini) and Salicaceae (tribe Vagrantini). A Malpighialean larval host appears to be ancestral in the common ancestor of Heliconiinae and its sister lineage, subfam. Limenitidinae [179].

#### 2.2.6. Fossil Record of *Viola*

*Viola* is represented in the fossil record of Eurasia from the Miocene onwards, by both pollen [187,188,189,190,191,192,193] and seeds [185,194,195,196,197,198,199,200,201,202]. There are in addition unconfirmed records of *Viola* macrofossils from the Pliocene and Pleistocene of North America [203,204,205,206]. *Viola* has no known fossil record in South America although this continent is where the genus has the longest history.

Records of *Viola* fossils older than Miocene are deemed questionable [200,207]. Significantly, one of the most frequently cited fossils (e.g., [208]), “*Viola*” *rimosa* P. Nikit. from the Oligocene and Miocene of western Eurasia, was reidentified as *Poliothyrsis* (Salicaceae) [207].

Seeds of *Viola* can be recognised by the relatively large chalaza, the transverse cellular pattern of the inner surface of the testa, and the existence of a layer with rhomboid crystals within the testa [197,199,200]. Fossil seeds of *Viola* are common in western Eurasian sediments from the Miocene onwards, where a total of c. 19 extinct morphotypes have been described [185,194,195,196,197,198,199,200,201,202]. Most of these are known from single fossil sites only but two have a wide stratigraphic range, i.e., *V. miocenica* (20.44–5.333 Ma, western Siberia [202]) and *V. neogenica* (15.97–2.58 Ma, Germany and Italy [200,201,209]). The oldest fossils of *Viola* are seeds from the Lower Miocene of Europe and comprise several unnamed morphotypes, one from the Burdigalian (18–17 Ma [184]) of Austria [185] and three from the Upper Karpatian (17.5–16.5 Ma [210]) of Poland [197]. Additionally four morphotypes, two of which are closely similar to one of the Polish ones [197], have been described from western Siberia [196,202] from about the same time interval (20.44–11.63 Ma [211]).

Seed fossils closely similar to, and possibly attributable to, extant species of *Viola* are known back to the Pliocene (5.333–2.58 Ma) of Europe, i.e., *V. palustris* back to the Lower Pliocene (5.333–3.6 Ma) of Germany [198,200] and European Russia [202], *V. tricolor* back to the Upper Pliocene (3.6–2.58 Ma) of Germany [198], and *V.* cf. *uliginosa* back to the Pliocene of Poland [194,197]. Seeds attributed to the extant *V. canina* and *V. rupestris* (probably incorrectly so) have been reported from the Tortonian (10–9 Ma [212]) of Germany [199]. Seed morphotypes comparable to sect. *Viola* have been reported from the Miocene of western Siberia [202], i.e., *V. miocenica* and *Viola* [Arbuzova] *sp. 6* (both compared to *V. alba*, *V. collina*, *V. mirabilis*, *V. riviniana*, and *V. suavis*). Seed morphotypes comparable to either of the two subsections of sect. *Viola* are younger, from the Pliocene (5.333–2.58 Ma) of the southern Urals [202]; i.e., *Viola* [Arbuzova] *sp. 1* to *3* are compared to species of subsect. *Viola* (*V. alba*, *V. ambigua*, *V. collina*, and *V. suavis*); and *Viola* [Arbuzova] *sp. 4* is compared to species of subsect. *Rostatae* (*V. mirabilis*, *V. reichenbachiana*, and *V. tanaitica*). Three among the oldest seed morphotypes (20.44–11.63 Ma) from western Siberia were reported to bear no similarity to extant taxa, i.e., *Viola* [Arbuzova] *sp. 5*, *Viola* [Arbuzova] *sp. 8*, and *V. kireevskiana* [202]. 

The assignments of these fossils to extant infrageneric taxa of *Viola* should be considered tentative as none has been justified by apomorphies or phylogenetic analysis. As noted by Łańcucka-Środoniowa [197], the taxonomic distinction of species in the genus *Viola* is difficult because the structure of seeds is very similar, at least among the European sections. Indeed, in a survey of seed morphology in East European angiosperms, Bojňanský & Fargašová [213] found no significant differences in seed morphology among the four sections of *Viola* studied by them, based on 28 species. However, their survey employed rather superficial morphological features observable using a light microscope, and it is therefore possible that more detailed studies using scanning electron microscope (SEM) micrographs on a more comprehensive sample of *Viola* sections could reveal apomorphies, e.g., such as seen within subsect. *Borealiamericanae* [214]. The only infrageneric group that stands out as distinct is the obligate myrmecochorous [215] subsect. *Viola* with its apomorphic large seeds, 2.0–3.0 × 1.3–2.0 mm (vs. 1.3–2.9 × 0.7–1.7 mm in other species), with a large elaiosome covering about half of the length of the raphe (vs. <1/3 in other taxa) [57,113,213]. The three fossil seed morphotypes with possible affinity to subsect. *Viola*, from the Pliocene of the southern Urals, are somewhat smaller (1.8–2.4 × 1.3–1.6 mm [202]) than seeds of extant species of this subsection [213]. However, at least within sect. *Viola*, seeds derived from chasmogamous flowers are often larger and heavier than seeds from cleistogamous flowers [216], up to almost twice as heavy in *V. odorata* [19]. 

The sudden appearance of *Viola* in the fossil record of western Eurasia and its almost immediate diversification into several recognisable morphotypes [185,196,197,202] agree with both the rapid radiation inferred from nuclear gene sequences [28,45] and the reconstruction of historical biogeography for both *Viola* (Figure 8) and Violaceae [4].

The perceived absence of *Viola* fossils in South America must be seen in the light of fossil recovery rates not being constant in time and space and across lineages. In fact, the exception in Violaceae is the “burst” in occurrence rates of *Viola* fossils in Eurasia from 17–18 Ma. Apart from that, the fossil record of Violaceae is practically non-existent. There are for instance no records from North America from the same period even though we must assume that *Viola* was present there. Possible explanations for the lack of identified, older *Viola* fossils in South America, despite the existence of geological formations of an appropriate age (e.g., Abanico from Eocene-Miocene, Río Turbio from Eocene-Oligocene, La Leona from Oligocene, etc.), include possibly low fossilisation probabilities due to *Viola* growing far from the fossilisation sites, and a lack of paleobotanical studies enforced by the absence of reference anatomical studies on extant taxa.

### 2.3. The “Known Unknowns”: Outstanding Research in Viola

The level of knowledge of the genus *Viola* has a strong geographic bias towards the northern hemisphere, primarily Europe, where taxonomic research has the longest history and where taxa have been most intensively studied. This has resulted in a “eurocentric” understanding of the diversity of the genus, its evolution, and its classification. The most significant gaps in our knowledge of *Viola* are for the South American taxa, notably subg. *Neoandinium*, for which classification, diversity and phylogeny are still poorly (or not) understood, all being based on morphological characters and geography. Because *Viola* originated in South America, understanding the evolutionary patterns here is key to understanding patterns within the genus as a whole. 

This is the first, comprehensive taxonomy for *Viola* in the last 97 years, since that of Becker (1925 [1]). It is beyond doubt that the century-long absence of systematised information that an updated classification would have represented has hindered the formation and testing of new hypotheses—and therefore accumulation of new knowledge. Below we discuss the most imminent gaps in our knowledge of *Viola*.

#### 2.3.1. Phylogeny of *Viola*

Phylogenetic data are completely lacking for the monotypic sect. *Nematocaulon* from New Zealand (*V. filicaulis*), sect. *Xanthidium* (*V. flavicans*) from South America, both in subg. *Viola*, and for most of subg. *Neoandinium* from South America. As subg. *Neoandinium* comprises a minimum of 140 known species and currently makes up some 21% of the diversity within the genus, this is beyond comparison the biggest knowledge gap within the genus. In addition, a large proportion of the species are narrow endemics that are critically endangered [68]. The monotypic sect. *Danxiaviola* is known from ITS and chloroplast sequences only which means that its ploidy and exact placement within the polyploid CHAM × MELVIO tangle remain unknown. While the occurrence of the polyploid CHAM × MELVIO tangle in the Northern Hemisphere has been well established, the same can not be said about the occurrence of similar tangles in the southern hemisphere involving the polyploid sections *Chilenium*, *Tridens*, *Leptidium*, *Erpetion*, and probably also *Nematocaulon* and *Xanthidium*. For these taxa inference of the species-level phylogeny in the study of Marcussen et al. [28] was rendered difficult by gene duplication and loss, even though three low-copy nuclear genes were used, and the lack of supporting data on chromosome numbers and ploidy. Though there is a large number of chromosome counts within the species-rich and probably also highly polyploid sect. *Melanium*, these numbers do not allow for reliable inferences on ploidy level in particular taxa. This lack of knowledge is combined with very limited information about the phylogeny of this section as the phylogenetic analyses, using a combination of ITS and ISSR markers [217] and more recently a combination of nuclear ITS and ETS and plastid *trnS–trnG* intergenic spacer sequences [94], have yielded poor resolution.

#### 2.3.2. Chromosome Counts and Ploidy

Chromosome number is an important taxonomic character and also gives information on ploidy. Chromosome counts are completely lacking for the sections *Chilenium*, *Melvio*, *Spathulidium*, and *Xanthidium*, and for most of subg. *Neoandinium*. Numerous other sections are represented only by a single count that is in need of confirmation (i.e., sects. *Abyssinium*, *Danxiaviola*, *Erpetion*, *Himalayum*, *Leptidium*, *Nematocaulon*, and *Rubellium*). Genome size has been measured by flow-cytometry mainly on European taxa [218,219,220,221] but is ploidy-informative within sections only.

#### 2.3.3. Fossil Record

Despite *Viola* having a rich seed fossil record from the Miocene (18–17 Ma) onwards of Europe and western Siberia, interpretations on phylogeny, evolution, and biogeography are limited by the lack of detailed knowledge of variation and apomorphies among extant species and sections of the genus, e.g., based on SEM micrographs. To this date, the only comparative study of seed morphology [213] covered only parts of the European territory and taxa and did not use SEM. Furthermore, the seed fossil record outside of western Eurasia is limited to unconfirmed records from the Pliocene and Pleistocene of North America, and there are no seed fossil records for *Viola* in South America although the genus has its longest history there. There are several geological formations in or near the Andes with fossiliferous horizons assignable to the Eocene-Oligocene boundary onwards. However, there are also no comprehensive studies on the morphology and anatomy of pollen, seeds, and other plant structures on the extant South American species of *Viola* that can serve as a solid basis for fossil surveys.

#### 2.3.4. Alpha Taxonomy

In recent years, a better understanding has been acquired of difficult groups such as subg. *Neoandinium* in South America (e.g., [68,80,222,223,224]), sect. *Nosphinium* subsect. *Borealiamericanae* in North America [67,214,225], sect. *Erpetion* in Australia [98,160,177,226,227], as well as the genus as a whole in China [76,78]. The last remaining blank spot seems to be the southeastern Asian and Malayan species, which comprises relatively few, but morphologically specialised and probably not closely related species that do not fit seamlessly with the taxonomic system, as indicated by the few treatments available [74,228,229,230,231].

#### 2.3.5. Transcriptomes and Genomes

Thus far, reference sequence genome has been published for the diploid *Viola* (sect. *Chamaemelanium*) *pubescens* [232] and the octoploid *V*. (sect. *Himalayum*) *kunawurensis* (as *V*. “*kunawarensis*”; NCBI accession PRJNA805692), but numerous *Viola* genomes are planned sequenced by the Earth Biogenome Project during the next decade [233]. Transcriptomes have been published for at least the four most widespread sections within subg. *Viola*, i.e., sects. *Chamaemelanium*, *Melanium*, *Plagiostigma*, and *Viola* (e.g., [234,235,236]), but to date no transcriptomes exist for taxa from outside of Eurasia and North America.

### 2.4. Taxonomic Treatment of Viola


**
*Viola*
**


*Viola* L., Sp. Pl. 2: 933 (1753).—Type (Brainerd 1913 [237], page: 546): *Viola odorata* L.

*Description*.—Annual or perennial acaulescent or caulescent herbs, shrubs or very rarely treelets. Axes morphologically differentiated or not. Stipules free or adnate, small or foliaceous, margin entire, laciniate, dentate, or fimbriate. Lamina linear to reniform, more or less petiolate, margin entire, crenulate, serrate, pinnate, or pedate. Flowers axillary and solitary, rarely in cymes. Peduncle non-articulated, lacking an abscission zone at the level of the bracteoles. Corolla white to yellow, orange or violet or multicoloured with or without yellow throat, strongly zygomorphic. Calycine appendages present. Bottom petal slightly to much shorter than others and weakly differentiated, rarely larger than others. Spur scarcely exserted to very long, rarely absent. Filaments free, two lowest stamens calcarate, dorsal connective appendage large, oblong-ovate, entire. Style filiform, clavate, or capitate, variously crested or not, bearded or not, often rostellate at tip. Capsule thick-walled. Seeds few to many per carpel, obovoid to globose, often arillate. Cleistogamous flowers often produced. Base chromosome numbers *x* = 6, 7.

*Diagnostic characters*.—Flowers axillary and solitary AND peduncle non-articulated AND plant herbaceous AND temperate distribution AND bottom petal slightly to much shorter than others and weakly differentiated.

*Ploidy and accepted chromosome counts*.—2*x*, 4*x*, 6*x*, 8*x*, 10*x*, 12*x*, 14*x*, 16*x*, 18*x*, 20*x*, >20*x*. 2*n* = 4, 8, 10, 12, 14, 16, 18, 20, 22, 24, 26, 28, 34, 36, 40, 44, c. 44, 46, 48, 50, 52, 54, 58, 60, c. 64, 72, 76, 80, c. 80, 82, c. 96, 102, c. 120, 128.

*Age*.—Crown node age 30.9 (29.8–31.3) Ma [28].

*Included species*.—664.

*Distribution*.—Temperate regions and montane areas in the tropics worldwide; all continents except Antarctica (Figure 2).

*Discussion*.—The two main lineages of *Viola* are here treated as subgenera, *Neoandinium* and *Viola*. The two subgenera differ rather consistently in aspects of growth form, leaf shape, degree of emargination of the bottom petal, shape of the anther appendages, style shape, and also in base chromosome number for the diploids investigated so far. Reiche [48,122,130] was the first to notice the fundamental distinction between these two sublineages of the genus. He recognised three sections, the first corresponding to subg. *Viola* (as sect. *Sparsifoliae*), the second to subg. *Neoandinium* (as sect. *Rosulatae*), and a third small section with four deviant taxa from both subgenera (sect. *Confertae*) [48]. Becker [1], however, treated subg. *Neoandinium* as one of 14 sections of the genus (as sect. *Andinium*).

#### 2.4.1. Key to the Subgenera, Sections, and Subsections of *Viola*

Conventions and definition of terms:*An “M” dash (“—”)* is used to identify uncommonly expressed traits/separate characters that have no counterpart in the antithesis.*Arosulate acaulescent*: with leaves scattered on stem, not in rosettes. Aerial stems and stolons (e.g., *V. filicaulis*).*Arosulate caulescent*: with leaves on aerial stems. Rosettes and stolons absent (e.g., *V. abyssinica*, *V. arborescens*, *V. stagnina*).*Beard*: tuft of hairs on the lateral petals (and sometimes upper or bottom petals) located at the throat of the chasmogamous flower, also a tuft of trichomes near the apex of the style in some species or groups. Organs with or without a beard are referred to as bearded or glabrous, respectively.*Calycine appendage*: Appendage at base of the sepal; synonymous with “sepal auricle” or “sepal appendage”.*Caulescent/acaulescent*: with/without aerial stems.*Flower colour*: base colour of the petals in living plants excluding the nectar guides, unless otherwise noted.*Foliaceous*: used to describe stipules that are green and often large and leaf-like (e.g., *V. elatior*, *V. raddeana*, *V. tricolor*).*Papilla*: lateral expansion of the cell wall to form a short conical structure up to 3 times as long as wide. For instance, a pad of papillae is found on the lateral petals of sect. *Erpetion* in place of a beard of trichomes exhibited in some other lineages.*Rosulate/arosulate*: with/without leaves in rosette.*Rosulate acaulescent*: with leaves in rosettes. Aerial stems and stolons absent (e.g., *V. hirta*, *V. pedata*, *V. selkirkii*).*Rosulate caulescent*: with leaves in rosettes, aerial stems present. Stolons absent (e.g., *V. canadensis*, *V. riviniana*).*Rosulate stoloniferous*: with leaves in rosettes, stolons present. Aerial stems absent (e.g., *V. banksii*, *V. odorata*, *V. palustris*).*Stolon*: lateral, specialised procumbent stem producing adventitious roots and new plantlets. We restrict the term to taxa in which the shoot axes are differentiated.*Trichome*: elongate hair-like structure usually more than 3 times as long as wide and typically linear or distinctly broader above the base.*Violet*: colour of the corolla and petal striation in many species. In the literature, this colour is often referred to, rather ambiguously, as “blue” or “purple”.

 


1a.Herbs, usually forming subacaulous imbricate or loose rosettes, very rarely erect-cauline, rarely woody based, or dwarf ericoid shrublets. Margin of juvenile laminas flat, not involute. Peduncle shorter or as long as mature lamina. Bottom petal usually cleft, more rarely emarginate or entire. Nectariferous appendage of the two bottom stamens filiform. Style at apex capitate, beardless, usually crested; crest 1–3 lobes or flanges at sides or top of style apex, or a continuous sharp dorsolateral rim, very rarely crest absent. Cleistogamous flowers not produced. (Subg. *Neoandinium*) ................................................................................................................................................................................................................................................................................................. 2.1b.Herbs, subshrubs or shrubs, with leaves scattered on stem or in rosette, rarely cushions with imbricated distichous leaves (sect. *Tridens*). Margin of juvenile laminas usually involute. Peduncle often longer than mature lamina. Bottom petal entire or emarginate, very rarely cleft. Nectariferous appendage of the two bottom stamens various in shape, very rarely filiform. Style filiform, clavate or (sub)capitate, not crested (lateral lobes present: sect. *Sclerosium*) but top of style apex often flattened or concave with more or less raised edges, sometimes bearded. Cleistogamous flowers often produced. (Subg. *Viola*) .............................................................................................................................................................................................................................................................................................. 13.2a.Underground part of stems conspicuously elongated, leafless and stolon-like, branching or not. ........................................................................................ **sect. *Rhizomandinium***2b.Stems without basal stolon-like segment. ........................................................................................................................................................................................................................ 3.3a.Leaves glabrous, except occasionally for minute cilia on margins, rarely glabrescent or pubescent. Lamina usually more or less rigid, thick or coriaceous; margins usually entire, rarely crenulate. ....................................................................................................................................................................................................................................................... 4.3b.Leaves with indumentum, or if glabrous, then with prominently raised veins above. Lamina flexible, thick or thin; margins usually crenate or incised, rarely entire. ...... 7.4a.Plant a dwarf ericoid shrublet. ...................................... ................................................................................................................................................**sect. *Ericoidium*** (*V. fluehmannii*)4b.Plants other. ......................................................................................................................................................................................................................................................................... 5.5a.Plant caulescent. ............................................................................................................................................................................................................... **sect. *Confertae*** (*V. nassauvioides*)5b.Plants subacaulous, rosulate. ............................................................................................................................................................................................................................................. 6.6a.Bottom petal longer than or equal to the other petals. ........................................................................................................................................................................ **sect. *Sempervivum***6b.Bottom petal much shorter than the other petals. ............................................................................................................................ **sect. *Inconspicuiflos***, in part (*V. membranacea*)7a.(4). Style crest as one apical and two lateral lobes. .............................................................................................................................................................................. **sect. *Triflabellium***7b.Style crest lateral, or lateral and frontal, or apical only, or a sharp dorsolateral rim. .................................................................................................................................................. 8.8a.Plant with short woody aerial stems. ........................................................................................................................................................................................... **sect. *Xylobasis*** (*V. beati*)8b.Plants completely herbaceous............................................................................................................................................................................................................................................ 9.9a.Corolla large, four times wider than lamina width or more. ................................................................................................................................................................. **sect. *Grandiflos***9b.Corolla small, usually as wide or up to twice as wide as lamina width, exceptionally up to four times wider than lamina width. ................................................................................................................................................................................................................................................................................................ 10.10a.Cilia long, surrounding entire lamina margin, strongly deflexed. ....................................................................................................................................................... **sect. *Relictium***10b.Cilia short, more or less patent. ..................................................................................................................................................................................................................................... 11.11a.Bottom petal much smaller than the other petals. .............................................................................................................................................. **sect. *Inconspicuiflos***, in largest part11b.Bottom petal not smaller than the other petals.............................................................................................................................................................................................................. 12.12a. Annuals. Lamina linear, oblanceolate or obovate; margin entire or shallowly and remotely crenulate. ................................................................................................................................................................................................................................................................. **sect. *Subandinium***12b.Annuals or perennials. Lamina elliptical, narrowly to broadly obovate, orbicular, or rhomboid; margin deeply to shallowly crenate, sinuate, incised, pinnatifid, or rarely entire when plant perennial. ...................................................................................................................................................................................................................... **sect. *Rosulatae***13a (1).Style slender and slightly clavate, with a pair of apical or subapical lateral lobes. Corolla white to violet with yellow-green throat. Stipules minute. Annual herbs or subshrubs. (northeastern Africa, southern and eastern Arabia, southwestern Asia) ...................................................................................................................**sect. *Sclerosium***13b.Style tubular, clavate or (sub)capitate, lacking lateral processes, but sometimes at apex bearded, or margined with +/− raised edges, or bilobate. Corolla yellow throughout, or white to violet with syncolorous or yellow-green throat, or multicoloured. Stipules prominent. Perennial herbs, sometimes annuals (in sect. *Melanium*), occasionally shrubs or subshrubs. (more widely distributed) .................................................................................................................................................................................. 14.14a.Bottom petal (excluding spur) more than twice as long and broad as lateral and upper petals. Subshrub. (southern China: Guangdong) ....................................................................................................................................................................................................................................... **sect. *Danxiaviola*** (*V. hybanthoides*)14b.Bottom petal (excluding spur) subequal to somewhat smaller or larger than lateral and upper petals. Herbs, subshrubs or shrubs. (not restricted to China) .................. 15.15a.Spur 12–30 mm long. Petals pink to magenta. Arosulate caulescent. Leaves sessile, seemingly ternate to palmate, with 3–5 lanceolate, entire segments (lamina and 2 or 4 stipule segments similar). (southern Europe) .................................................................................................................................................................................. **sect. *Delphiniopsis***15b.Spur <20 mm long (to 16 mm in sect. *Melanium* and in subsect. *Rostratae*). Petals of various colours, very rarely pink to magenta. Rosulate or arosulate, caulescent or acaulescent. Leaves commonly petiolate. (not restricted to southern Europe) ..................................................................................................................................................... 16.16a.Lamina subulate, somewhat succulent, margin entire. Style sigmoid, dorsiventrally flattened at base, tapering in width and becoming filiform towards apex, with an apical stigmatic opening. Subshrub. (South Africa) ......................................................................................................................................................... **sect. *Melvio*** (*V. decumbens*)16b.Lamina broader, margin usually crenate. Style filiform, clavate or capitate. Herbs, subshrubs, or shrubs. (not South Africa) .......................................................................... 17.17a.Style filiform, protruding, straight or somewhat geniculate at base, with an apical stigmatic opening. Spur reduced to a swelling (gibba), or short, 0.5–1.5 mm, as long as tall (spur 4 mm long and as tall, half length of petal blade, and corolla bright red: *V*. (sect. *Leptidium*) *arguta*). Herbs, more rarely subshrubs or shrubs. ........................ 18.17b.Style clavate or (sub)capitate, monosymmetric (style filiform and spur 4–9 mm long: *V*. (sect. *Viola*) *papuana*). Spur well developed, as long as tall or longer. Herbs or subshrubs............................................................................................................................................................................................................................................................................ 21.18a.Leaves 2–10 mm long. Petiole indistinct. Lamina obovate, at apex tridentate, sometimes bilobate or entire. Phyllotaxis distichous. (s South America) ......................... ..................................................................................................................................................................................................................................................... **sect. *Tridens*** (*V. tridentata*)18b.Leaves >10 mm long. Petiole distinct. Lamina of various shapes, crenate. Phyllotaxis polystichous. .................................................................................................................... 19.19a.Stipules long, densely short-fimbriate, broad and sheathing the stem. Subshrubs or herbs. Arosulate caulescent, with reclining or weakly ascending to erect stems. Corolla with a white throat, rarely throat red (*V. arguta*). (Latin America) ......................................................................................................................................... **sect. *Leptidium***19b.Stipules rather small, entire or sparingly lacerate to laciniate with few long processes, not sheathing the stem. Herbs. Rosulate stoloniferous or arosulate acaulescent. Corolla with a yellow throat or with a green blotch on bottom petal. ..................................................................................................................................................................... 20.20a.Stem creeping, remotely noded, branched. Stolons absent. Corolla with a yellow throat. Spur distinct, 0.5–1.5 mm long, yellow. Lateral petals sparsely bearded. Cleistogamous flowers produced. (New Zealand) ................................................................................................................................................... **sect. *Nematocaulon*** (*V. filicaulis*)20b.Stem usually densely noded (usually rosettes). Stolons present, sympodial. Corolla without a yellow throat, but bottom petal with a green blotch inside. Spur absent, reduced to a swelling (gibba). Lateral petals with a broad dense pad of papillae. Cleistogamous flowers not produced, but some species have flowers with a small corolla. (Australia) ......................................................................................................................................................................................................................................... **sect. *Erpetion***21a(17). Corolla white on the inside, rarely pale violet, lacking violet striation. Shrubs, usually with lateral, leafless 1–few-flowered inflorescences, rarely herbs with solitary flowers (*V. kauaensis*). (Hawaiian Islands).—Lower stipules ovate or triangular, partially sheathing the stem. Style apex with weak subapical dorsolateral swelling (where distinct rim occurs in several other groups), rostellum formed by bent apex tall and blunt at tip. ....................................................... **sect. *Nosphinium* subsect. *Nosphinium***21b.Corolla variously coloured, usually with violet striation. Herbs or subshrubs. Flowers solitary, not in inflorescences. (not Hawaiian Islands) ............................................................................................................................................................................................................................................................................................. 22.22a.Small subshrubs. Lamina lanceolate or spathulate. .................................................................................................................................................................................................... 23.22b.Herbs, sometimes with a woody rhizome. Lamina shape and style shape variable. Cleistogamous flowers produced or not. ....................................................................... 24.23a.Leaf base decurrent. Petiole indistinct. Stipules entire or with one or two basal segments, sometimes foliaceous. Corolla violet, white or yellow. Style apex scarcely to weakly bent ventrad. (Mediterranean) ................................................................................................................................................................................................. **sect. *Xylinosium***23b.Leaf base cuneate. Petiole distinct. Stipules small, bract-shaped, fimbriate. Corolla violet or magenta. Style apex strongly bent ventrad or with stigma on ventral side. (Chile) ........................................................................................................................................................................................................................................................... **sect. *Rubellium***24a.Corolla with a yellow throat. Petals yellow or variously coloured. Style clavate or (sub)capitate. (Throat white, corolla white with reddish-violet striation, stipules free, spur as long as tall, style more or less filiform: *V. commersonii*. Throat white or cream, petals violet, style capitate: *V. nummulariifolia*, *V. cornuta*, *V. orthoceras*.) .............. 25.24b.Corolla with a white or cream, violet, or yellowish-green throat. Petals usually violet or white, occasionally pink, never yellow. Style clavate or cylindrical, rarely filiform, never capitate. .................................................................................................................................................................................................................................................... 32.25a.Usually caulescent. Perennial or annual. Stipules entire or with a few irregular teeth, or deeply pinnatifid. Petals yellow or variously coloured. Style usually capitate and bearded. (Northern Hemisphere, naturalised elsewhere) ........................................................................................................................................................................................... 26.25b.Acaulescent. Perennial. Stipules glandular-lacerate to glandular-laciniate. Petals yellow (white in *V*. (sect. *Chilenium*) *commersonii*). Style usually concave or flattened at apex, glabrous or bearded. (Style ellipsoid with broadly rounded apex when fresh in sect. *Xanthidium*, bearded: *V. flavicans*.) (South America) ......................................... 31.26a.Perennial. Rosulate, caulescent, rarely stoloniferous (*V. kusnezowiana*) or acaulescent (*V. barroetana*). Perennating stem a monopodial rhizome, often deeply buried. Stipules not distinctly foliaceous, margins entire or with 1–2(–4) irregular shallow teeth on either margin. Spur usually very short to short (less than twice as long as tall), rarely longer (in 2 Asian species). Calycine appendages short (<2 mm). Bottom petal (including spur) typically <15 mm. Style various at apex, often (sub)capitate and bearded, occasionally bifid, but lacking shallow reflexed lateral lobes. Lamina margin subentire, crenate, lobed or divided.—................................... **sect. *Chamaemelanium***26b.Perennial to annual. Arosulate caulescent, sometimes indistinctly caulescent (*V. alpina*). Perennating stem a sympodially branching pleiocorm. Stipules usually large and foliaceous, pinnatifid or palmately divided, rarely small with entire or dentate margins. Spur very short to very long (0.9–16 mm, often much longer than tall). Calycine appendages very short to very long (0.3–4.7 mm). Bottom petal (including spur) 2–34 mm. Style capitate and bearded at apex, with a pair of inconspicuous or prominent shallow reflexed lateral lobes. Lamina margin entire or crenate, never lobed or deeply divided. (sect. *Melanium*) ...................................................................................................................................................................................................................................................................................... 27.27a.Cleistogamous flowers produced in summer. Annual or biennial. (eastern North America) ........................................ **sect. *Melanium* subsect. *Cleistogamae*** (*V. rafinesquei*)27b.Cleistogamous flowers not produced. Annual to perennial. (Palaearctic, naturalised elsewhere) ......................................................................................................................... 28.28a.Corolla violet, with a cream-coloured throat. Stipules ovate-lanceolate, dentate. Bottom petal 9.5–10.5 mm. Low, high-Alpine perennial. (southwestern Alps and Corsica) ......................................................................................................................................................................................... **sect. *Melanium* subsect. *Pseudorupestres*** (*V. nummulariifolia*)28b.Corolla colour various, often yellow or violet, with a bright yellow throat (if throat cream or white, then lateral petals directed horizontally or downwards: *V. cornuta* and *V. orthoceras*). Stipules variable, often foliaceous, rarely dentate. Bottom petal 2–34 mm. Annual or perennial. ......................................................................................... 29.29a.Annual. Basal leaves entire or indistinctly crenulate. Bottom petal 2–11.5 mm. Spur 0.9–3 mm. ............................................................. **sect. *Melanium* subsect. *Ebracteatae***29b.Annual to perennial. Leaves crenate or entire, but in annual species basal leaves crenate. Bottom petal 5–34 mm. Spur 1–16 mm. ................................................................. 30.30a.Calycine appendages 0.3–1.0 mm. Bottom petal 5–13 mm. Spur 1–3.5 mm. (Mediterranean area) .................................................................. **sect. *Melanium* subsect. *Dispares***30b.Calycine appendages 0.9–4.7 mm. Bottom petal 5.4–34 mm. Spur 1.8–16 mm. ............................................................................................ **sect. *Melanium* subsect. *Bracteolatae***31a.Rosulate, perennating stem a short monopodial rhizome. Style ellipsoid and broadly rounded at apex when fresh, when dried clavate with flattened apex, bearded (*V. flavicans*) or at most occasionally papillate (*V. pallascaensis*). Stipules adnate at base or for most of their length, narrow, shallowly glandular-lacerate. Bracteoles narrow, shallowly glandular-lacerate. ................................................................................................................................................................................................................. **sect. *Xanthidium***31b.Variably rosulate or arosulate, perennating stems multiple, elongate and deeply buried. Style clavate or straight fresh or dried, apex concave, flattened or slightly acute with sharp dorsolateral rim, sometimes with a short subapical ventrad or incurved rostellum bearing the stigma, usually beardless (white-hairy in *V. rudolphii*). Stipules free, broad, deeply glandular-laciniate, rarely entire. Bracteoles broad, deeply glandular-laciniate. ............................................................................................. **sect. *Chilenium***32a(24). Stipules adnate at least in the lower 1/3, rarely in the lower 1/4 or less (subsect. *Clausenianae*). Rosulate acaulescent, estoloniferous. ..................................................... 33.32b.Stipules free or adnate at base only. Rosulate caulescent, rosulate stoloniferous, rosulate acaulescent, or arosulate caulescent. (Stipules partly adnate and plant stoloniferous: subsect. *Bulbosae* and *V*. (subsect. *Rostratae*) *uliginosa*. Stipules (1/2–)2/3 or more adnate and flowers white with bottom petal blade densely striated, spur shorter than tall: *V*. (subsect. *Mexicanae*) *humilis*.) ....................................................................................................................................................................................................... 34.33a.Lamina of various shape but not spathulate, undivided with margin crenulate or serrate, or incised to dissected. ......................................................................................... 34.33b.Lamina spathulate, undivided, margin entire or indistinctly and remotely crenulate (southern and western Asia). .......................................................................................... 36.34a.Lamina deeply pedately dissected. Calycine appendages entire. Spur short, as long as tall. Style with long dorsolateral margin closely following style body as a narrowly rounded rim running laterally and ventrally at an acute angle from dorsum of apex to a more proximal point on the ventral surface, the stigma hidden in the narrow cavity created by the rim. Cleistogamous flowers not produced. .................................................................................................... **sect. *Nosphinium* subsect. *Pedatae*** (*V. pedata*)34b.Lamina undivided, incised to pinnatifid, or ternately to triternately dissected. Calycine appendages dentate or entire. Spur as long as tall or longer. Style with dorsolateral margin obsolete, or short and more or less perpendicular to dorsum, or produced as a thick or swollen continuous rim at an acute angle from dorsum of apex to the centre of the ventral surface. Cleistogamous flowers produced. ............................................................................................................................................................................................................................................................................................. 35.35a.Stipules adnate in lower 1/4, margins glandular-lacerate. Calycine appendages short, triangular, narrowly rounded at apex, entire. Spur short, as long as tall. Style apex protruded dorsally as a thickened broadly truncate or slightly emarginate rim, continuous laterally and ventrally at an acute angle from dorsum of apex to a proximal point on the ventral surface, ending in a strongly incurved rostellum. Lamina deltoid-triangular. (western North America: Utah) .............................................................................................................................................................................................. **sect. *Nosphinium* subsect. *Clausenianae*** (*V. clauseniana*)35b.Stipules 1/3–3/4 adnate to petiole, margins entire or indistinctly crenulate. Calycine appendages short or elongated, usually oblong, truncate or emarginate at apex, usually dentate. Spur longer than tall, usually 1/5 to 1/2 of total length of bottom petal, 2–10 mm. Style apex with dorsolateral margin obsolete, or dorsolateral margin slightly thickened or produced as a pair of short lobes more or less perpendicular to dorsum but not continuous laterally to the straight ventrad rostellum. Lamina of various shape, undivided, deeply incised, lobed or dissected. (not restricted to North America) ................................. **sect. *Plagiostigma* subsect. *Patellares***, in largest part36a(33). Petiole indistinct, about as long as lamina. Style apex with thickened dorsal margin and a ventral rostrum. Cleistogamous flowers not produced. Spur 1.5–4 mm, longer than tall. (southwestern Asia) .................................................................................................................................................................................................. **sect. *Spathulidium***36b.Petiole distinct, at least twice as long as lamina. Style lacking distinct margins. Cleistogamous flowers produced. ........................................................................................................................................................................................................................................................................................... 37.37a.Spur c. 1.5 mm, as long as tall. Plant with stems subterranean from deeply buried rhizome, appearing aboveground as proximal or tufted rosettes. (Rim of the Tibetan Plateau) ........................................................................................................................................................................................................................ **sect. *Himalayum*** (*V. kunawurensis*)37b.Spur 3–7.5 mm, longer than tall. Rhizome usually at soil surface, with leaf rosette. ...... **Sect. *Plagiostigma* subsect. *Patellares***, in part (*V. alaica*, *V. dolichocentra*, *V. turkestanica*)38a(32). Spur longer than tall. ........................................................................................................................................................................................................................................39.38b.Spur shorter than tall. ................................................................................................................................................................................................................................................ 42.39a.Bottom petal (excluding spur) conspicuously longer than the other petals, emarginate, 6–11 mm. Stolons leafless, terminated by a leafy rosette. (Taiwan, Ryukyu islands) .......................................................................................................................................................................................................................... **sect. *Plagiostigma* subsect. *Formosanae***
39b.Bottom petal (excluding spur) not longer than the other petals. Stolons, if present, with scattered leaves along the length. (not restricted to southeastern Asia) ........... 39.40a.Spur saccate, less than twice as long as tall. Calycine appendages very short or obsolete, 0–0.5 mm. Arosulate caulescent. Stems creeping to reclining or suberect, proximally rooting. Style clavate, apex sharply bent 90° ventrad into a prolonged rostrum, beardless. Cleistogamous flowers not produced. (Africa) .........**sect. *Abyssinium***40b.Spur not saccate, pronounced to very long, (much) more than twice as long as tall, 2–20 mm long. Calycine appendages short or long, >0.5 mm long. Rosulate caulescent, rosulate stoloniferous, rosulate acaulescent, or arosulate caulescent. Style cylindrical or subclavate, apex straight to slightly curved or abruptly bent ventrad, bearded or beardless above. Cleistogamous flowers usually produced (sect. *Viola*). ............................................................................................................................................................. 41.41a.Capsule trigonous-ellipsoid, usually glabrous, forcibly ejecting the seeds after dehiscence, borne on erect peduncles at maturity. Style often bearded above, nearly straight to weakly bent at apex with rostellum. Usually rosulate caulescent, more rarely rosulate stoloniferous, rosulate acaulescent, or arosulate caulescent. .................. ............................................................................................................................................................................................................................................... **sect. *Viola* subsect. *Rostratae***41b.Capsule globose, usually hairy, non-dehiscent, borne on decumbent to prostrate peduncles at maturity (cleistogamous flowers and capsules often underground). Style beardless, often strongly bent at apex with pronounced rostrum. Rosulate acaulescent or rosulate stoloniferous. ................................................... **sect. *Viola* subsect. *Viola***42a(38). Corolla pale pink or pale violet, rarely white. Bottom petal 2.5–12 mm long (including spur), conspicuously shorter and narrower than the others, usually acute, with distinct violet striation or reticulation. Style apex bilobate. Stipules linear to broadly lanceolate, densely or remotely fimbriate, free or 1/3 adnate. Stolons produced. .............................................................................................................................................................................................................................................................................................. 43.42b.Corolla white or violet, occasionally pink. Bottom petal 7–25 mm (including spur), not usually conspicuously smaller than the others. Style apex bilobate or distinctly margined. Stipules lanceolate to ovate, entire or remotely denticulate to fimbriate-dentate, free or adnate. Stolons produced or not. ........................................................... 4443a.Lateral petals not bearded. Peduncles glabrous; plant usually glabrous or nearly so. Rhizome long and remotely noded or short and densely noded. Stolons present or rarely absent, with (many) scattered leaves. Stipules free or adnate at base only, often brownish, long-fimbriate to laciniate. Corolla usually pale violet to whitish, without a greenish throat. Perennials. ............................................................................................................. **sect. *Plagiostigma* subsect. *Australasiaticae***43b.Lateral petals usually bearded. Peduncles with patent hairs, rarely glabrous (in *V. nanlingensis*); plant usually hairy. Rhizome short, densely noded. Stolons with 1–2 (smaller) leaves and a leaf rosette at apex. Stipules adnate in the lower 1/3 (stipules on stems free in *V. guangzhouensis*), remotely or rarely densely fimbriate. Corolla usually pale pink to pale violet, with a greenish throat. Perennials or rarely annuals (*V. diffusa*). ......................................................... **sect. *Plagiostigma* subsect. *Diffusae***44a.Bottom petal 7–12 mm including the spur. Corolla usually white with violet striation. Style strongly bilobate or distinctly margined all around. ...................................... 45.44b.Bottom petal 12–25 mm including the spur. Corolla violet, rarely white (*V. grahamii*, some *V. hookeriana*, some *V. moupinensis*, *V. oxyodontis*, *V. brevipes*, some *V. thomsonii*) or rose-violet (*V. rossii*). Style with weak to pronounced dorsolateral rim or not, not strongly bilobate. .............................................................................................................. 47.45a.Stem vertical, growing from underground bulbil. Stolons underground, branched, leafless, with cleistogamous flowers. Outer stipules adnate, inner stipules free.—Style bilobate. ................................................................................................................................................................................................................. **sect. *Plagiostigma* subsect. *Bulbosae***45b.Bulbils absent, rhizome oblique to vertical. Stolons different than above, or absent. Stipules usually free. ............................................................................................................................................................................................................................................................................................ 46.46a.Lateral stems creeping, ascending or erect. Stipules green, margins entire, remotely denticulate, or 1–3-toothed on either side, teeth eglandular. Style apex bilobate. ................................................................................................................................................................................................................................. **sect. *Plagiostigma* subsect. *Bilobatae***46b.Lateral stems absent, or present as stolons. Stipules membranous, glandular-lacerate. Style apex margined or rarely bilobate. .................................................................................................................................................................................................. **sect. *Plagiostigma* subsect. *Stolonosae***, in largest part47a(44). Lateral petals glabrous, rarely with a few hairs. Calycine appendages dentate or entire. Style with or without a distinct dorsolateral rim, if present this short and weakly spreading or oriented apically, usually not extending much laterally. .............. **sect. *Plagiostigma* subsect. *Stolonosae***, in part (*V. suecica*, *V. bissettii*, *V. brevipes*, *V. diamantiaca*, *V. epipsila*, *V. moupinensis*, *V. palustris*, *V. pluviae*, *V. rossii*, *V. thomsonii*, *V. vaginata*)47b.Lateral petals densely bearded (glabrous or sparsely bearded in certain species of subsect. *Mexicanae*). Calycine appendages entire (dentate in a few species of subsect. *Borealiamericanae*). Style apex sharp-edged without a distinct dorsolateral rim or with a pronounced and thickened spreading rim commonly extending laterally to the rostellum. ........................................................................................................................................................................................................................................................................ 48.48a.Aerial stems present. Lower stipules ovate, shallowly glandular-fimbriate, sheathing the stem. Style apex broadly rounded, with or without a weak dorsal or dorsolateral swelling in place of a distinct rim. (Amphiberingian) .................................................................................................. **sect. *Nosphinium* subsect. *Langsdorffianae***48b.Aerial stems absent. Stipules linear-lanceolate to lanceolate (ovate and glandular-laciniate in *V. guatemalensis* and *V. nubicola* of subsect. *Mexicanae*). Style apex abruptly flattened or concave with a sharp edge, or flanked by a prominent truncate to emarginate or bilobate spreading to dorsad thickened rim. .................................................................................................................................................................................................................................................................................... 49.49a.Stolons present or absent, if absent then lateral petals glabrous or sparsely bearded. Lateral petals glabrous or sparsely bearded (densely bearded in stoloniferous white-flowered *V. grahamii* and *V. oxyodontis*). Corolla white or violet, rarely dark violet (*V. beamanii*). Stipules in some species basally or mostly adnate. Calycine appendages short and entire. Bottom petal glabrous. Style apex merely sharp-edged or scarcely thickened, apically oriented or slightly inrolled, not prolonged, not strongly thickened or spreading and not extending much laterally (somewhat prolonged and slightly thickened dorsally in *V. hookeriana*). (Mexico to northern South America) ........................................................................................................................................................................................................................... **sect. *Nosphinium* subsect. *Mexicanae***49b.Stolons absent. Lateral petals densely bearded. Corolla violet to dark violet. Stipules free. Calycine appendages short or elongated, entire or dentate. Bottom petal glabrous or bearded. Style apex with pronounced thickened ascending to spreading rounded or strongly emarginate dorsolateral rim extending ventrad partly or fully to rostellum. (North America, *V. nuevoleonensis* in northern Mexico) ............................................................................................ **sect. *Nosphinium* subsect. *Borealiamericanae***


#### 2.4.2. Descriptions of Subgenera, Sections, and Subsections of *Viola*

**[1] *Viola* subg. *Neoandinium* (Figure 3a**–**l)**

*Viola* subg. *Neoandinium* Marcussen, Nicola, Danihelka, H. E. Ballard, A. R. Flores, J. S. Watson, subg. nov.—Type: *Viola rosulata* Poepp. & Endl.

*Description*.—Perennial or annual herbs, usually forming subacaulous imbricate or loose rosettes, very rarely either caulescent, woody based, or dwarf ericoid subshrublet (in sect. Ericoidium). Axes not morphologically differentiated. Stems vertical, branched or not, occasionally arising from a buried branching “rhizome” (stolon-like persistent axes). Stipules inconspicuous or sometimes absent. Lamina usually spathulate, tapering into the petiole (pseudopetiole); margin entire, hyaline, crenulate, or lobed to pinnate; margin of juvenile laminas flat, not involute. Peduncle shorter or as long as mature laminas. Bottom petal usually cleft, rarely emarginate or entire. Spur present or rarely absent. Nectariferous appendage of the two bottom stamens filiform. Style at apex capitate and crested; crest 1–3 lobes or flanges at sides or top of style apex, or a continuous sharp dorsolateral rim, very rarely crest absent. Cleistogamous flowers not produced. Diploid. Base chromosome number *x* = 7.

*Diagnostic characters*.—Margin of juvenile laminas not involute OR peduncles not longer than mature leaves OR style capitate and crested OR cleistogamous flowers absent. 

*Ploidy and accepted chromosome counts*.—2*x*; 2*n* = 14.

*Age*.—Crown node age c. 20.3 Ma (Figure 6).

*Included species*.—139.

*Distribution*.—From the equator (Ecuador) to southern Patagonia (Argentina) (Figure 10).

*Etymology*.—The well-established but illegitimate sectional name *Andinium* refers to the majority of species of the subgenus (90%) inhabiting the Andes mountains (Figure 10). Instead of combining the little used name *Viola* sect. *Rosulatae* to the subgenus level, we are deliberately describing a new subgenus, *Neoandinium*, with a name that clearly indicates a connection to Becker’s sect. *Andinium*.

*Discussion*.—The subgenus state of subg. *Neoandinium* is justified by its phylogenetic sister position to the rest of *Viola* and by its morphological distinctness, notably in the frequently imbricate rosettes and conspicuously and variably crested style. In spite of the high species diversity (21% of the total diversity of *Viola*) and wide distribution in the Andes, subg. *Neoandinium* is incompletely known. Dozens of species await description [68] and the subgenus lacks both a phylogeny and until recently a taxonomic treatment. The data presented here are a synopsis of the recent monograph by Watson et al. [68] who recognised 11 morphological sections within subg. *Neoandinium*. Hitherto all species studied have proven diploid (four species in two sections) but unpublished data on gene homoeolog numbers indicate allopolyploidy at least within sect. *Sempervivum* (T.M., unpublished). 

Both Reiche [48] and Becker [1] subdivided subg. *Neoandinium* in annual and perennial species, but this classification does not appear to be natural [68]. However, this difference in life cycles is reflected in a difference in the growth form. Annual species have a taproot and only one rosette, while perennial species present a taproot usually branching below the ground, and various degrees of transition between rosettes, pleiocorm, and alpine cushion plants. Stolon-like persistent axes can also rarely be found among perennial species (sect. *Rhizomandinium*). A constant character within the subgenus is the margins of the leaf lamina. On the one hand, there is a group of species that present entire margins (sects. *Confertae*, *Ericoidium*, *Rhizomandinium*, and *Sempervivum*) and, on the other hand, another group of species with crenulate, crenate, lobed, even incised margins (sects. *Grandiflos*, *Inconspicuiflos*, *Relictium*, *Rosulatae*, *Subandinium*, *Triflabellium*, and *Xylobasis*). Generally, hairiness and the presence/absence of glands are correlated with this character; the entire leaves being generally glabrous without glands, and the leaves with non-entire margins often having hairs, glands, and raised veins. Because several characters are correlated, it can be hypothesised that these two morphological groups reflect phylogeny at least to some degree, but it is currently not known whether they are phylogenetic sisters or whether one is nested within the other.

The undescribed *Viola quasichilenium* J. M. Watson & A. R. Flores, ined., is superficially similar to sect. *Chilenium* of subg. *Viola* in having an extended petiole and in corolla colour and shape, but belongs in subg. *Neoandinium* on the basis of having abaxial lamina glands and a style with a significant crest, apparently apical. The specimen is known from photograph only, without geographical information.

 


**[1.1] *Viola* sect. *Confertae* (**
Figure 3
**a and **
Figure 9
**a)**


*Viola* sect. *Confertae* Reiche in Nat. Pflanzenfam., ed. 1 [Engler & Prantl], 3(6): 335. 1895.—Lectotype (Watson et al. [68], page: 189): *Viola nassauvioides* Phil.

*Diagnostic characters*.—Perennial erect, caulescent, glabrous herb. Fertile stem enveloped in short, acaulous laminas, apex as expanded, imbricate rosette. Sterile rosettes basal, subacaulous, imbricate.

*Included species*.—1. *Viola nassauvioides* Phil.

*Distribution*.—Unknown (probably central Chile) [68].

 


**[1.2] *Viola* sect. *Ericoidium* (**
Figure 3
**b and **
Figure 9
**b)**


*Viola* sect. *Ericoidium* J. M. Watson, A. R. Flores & Marcussen in Watson et al., Viola Subg. Andinium: 189. 2021.—Type: *Viola fluehmannii* Phil.

*Diagnostic characters*.—Perennial dwarf ericoid shrublets.

*Included species*.—1. *Viola fluehmannii* Phil.

*Distribution*.—Southern Chile, central-western Argentina.

 


**[1.3] *Viola* sect. *Grandiflos* (**
Figure 3
**c and **
Figure 9
**c)**


*Viola* sect. *Grandiflos* J. M. Watson, A. R. Flores & Marcussen in Watson et al., Viola Subg. Andinium: 190. 2021.—Type: *Viola truncata* Meyen.

*Diagnostic characters*.—Perennial subacaulous, rosette-forming herbs. Rosette loose, irregular, not imbricated, radiating, not depressed. Lamina narrow, oblanceolate-spathulate, flexible, acute, entire, dentate or pinnatifid, never crenate. Corolla large, prominent, c. 15 × 15 mm, twice as wide as lamina or more.

*Included species*.—6. *Viola acanthophylla* Leyb. ex Reiche, *V. angustifolia* Phil., *V. belovorum* J. M. Watson & A. R. Flores, ined., *V. bustillosia* Gay, *V. cheeseana* J. M. Watson, *V. truncata* Meyen

*Distribution*.—Central Chile.

 

**[1.4] *Viola* sect. *Inconspicuiflos* (**Figure 3**d**–**e and **Figure 9**d**–**e)**

*Viola* sect. *Inconspicuiflos* J. M. Watson & A. R. Flores in Watson et al., Viola Subg. Andinium: 192. 2021.—Type: *Viola lilliputana* Iltis & H. E. Ballard

*Diagnostic characters*.—Dwarf, cushion forming plants, glabrous or with indumentum. Corolla notably small, the upper and lateral petals distinctly larger than the bottom one.

*Included species*.—8. *Viola blefescudiana,* ined., *V. diminutiva,* ined., *V. enmae* P. Gonzáles, *V. lilliputana* Iltis & H. E. Ballard, *V. membranacea* W. Becker, *V. quasimelanium* H. Beltrán & J. M. Watson, ined., *V. quercifolia,* ined., *V. weibelii* J. F. Macbr.

*Distribution*.—Peru.

 


**[1.5] *Viola* sect. *Relictium* (**
Figure 3
**f and **
Figure 9
**f)**


*Viola* sect. *Relictium* J. M. Watson, A. R. Flores & Marcussen in Watson et al., Viola Subg. Andinium: 193. 2021.—Type: *Viola huesoensis* Martic.

*Diagnostic characters*.—Annual rosulate herbs. Cilia long, surrounding entire lamina margin, strongly deflexed.

*Distribution*.—Northern Chile.

*Included species*.—8. *Viola**dandoisiorum* J. M. Watson & A. R. Flores, *V. deflexa,* ined., *V. godoyae* Phil., *V. huesoensis* Martic., *V. johnstonii* W. Becker, *V. marcelorosasii* J. M. Watson & A. R. Flores, *V. ovalleana* Phil., *V. simulans,* ined.

 


**[1.6] *Viola* sect. *Rhizomandinium* (**
Figure 3
**g and **
Figure 9
**g)**


*Viola* sect. *Rhizomandinium* J. M. Watson, A. R. Flores & Marcussen in Watson et al., Viola Subg. Andinium: 193. 2021 (“*Rhizomandimium*”).—Type: *Viola escondidaensis* W. Becker

*Diagnostic characters*.—Perennial herbs. Stem arising from the apex of long, creeping, stolon-like segment.

*Distribution*.—Northern Argentine Patagonia.

*Included species*.—2. *Viola**anitae* J. M. Watson, *V. escondidaensis* W. Becker

 


**[1.7] *Viola* sect. *Rosulatae* (**
Figure 3
**h and **
Figure 9
**h–m)**


*Viola* sect. *Rosulatae* Reiche in Nat. Pflanzenfam., ed. 1 [Engler & Prantl], 3(6): 335. 1895.—Type (Shenzhen Code Art. 22.2): *Viola rosulata* Poepp. & Endl.

≡ *Viola* sect. *Andinium* W. Becker in Nat. Pflanzenfam., ed. 2 [Engler & Prantl], 21: 374. 1925, nom. illeg. superfl. (Shenzhen Code Art. 52.1).—Type (Shenzhen Code Art. 7.5): *Viola rosulata* Poepp. & Endl.

*Diagnostic characters*.—Perennial or annual subacaulous, more or less hairy rosette-forming herbs. Lamina flexible, elliptical, narrowly to broadly obovate, orbicular, or rhomboid, deeply to shallowly crenate, sinuous-crenate, dentate, incised, pinnatifid, or rarely entire when plant perennial.

*Ploidy and accepted chromosome counts*.—2*x* (*V. congesta*); 2*n* = 14 (*V. montagnei*, *V. roigii*).

*Age*.—Crown node unknown; stem node c. 13.9 Ma (Figure 6).

*Distribution*.—Central-northern Peru to northern Patagonia.

*Included species*.—55. *Viola**(Rosulatae)* sp. 02, ined., *V. (Rosulatae)* sp. 04, ined., *V. (Rosulatae)* sp. 05, ined., *V. (Rosulatae)* sp. 06, ined., *V. argentina* W. Becker, *V. aurantiaca* Leyb., *V. calchaquiensis* W. Becker, *V. chamaedrys* Leyb., *V. cistanthe,* ined., *V. congesta* Gillies ex Hook. & Arn., *V. decipiens* Reiche, *V. escarapela* J. M. Watson & A. R. Flores, *V. evae* Hieron. ex W. Becker, *V. exilis* Phil., *V. exsul* J. M. Watson & A. R. Flores, *V. farkasiana* J. M. Watson & A. R. Flores, *V. ferreyrae* P. Gonzáles, *V. friderici* W. Becker, *V. frigida* Phil., *V. gelida* J. M. Watson, M. P. Cárdenas & A. R. Flores, *V. glechomoides* Leyb., *V. granulosa* Wedd., *V. hillii* W. Becker, *V. hippocratica* J. M. Watson & A. R. Flores, ined., *V. imbricata* J. M. Watson & A. R. Flores (et al.), ined., *V. kermesina* W. Becker, *V. lanifera* W. Becker, *V. lilloana* W. Becker, *V. llullaillacoensis* W. Becker, *V. longibracteata* P. Gonzáles & J. M. Watson, ined., *V. montagnei* Gay, *V. multiflora,* ined., *V. nazarenoensis,* ined., *V. neuquenensis* J. M. Watson & A. R. Flores, ined., *V. niederleinii* W. Becker, *V. ornata* D. Montesinos & J. M. Watson (et al.), ined., *V. philippiana* Greene, *V. philippii* Leyb., *V. replicata* W. Becker, *V. rhombiloba* H. E. Ballard, ined. [Monheim s. n.], *V. rodriguezii* W. Becker, *V. roigii* Rossow, *V. rosulata* Poepp. & Endl., *V. rubromarginata* J. M. Watson & A. R. Flores, *V. rugosa* Phil. ex W. Becker, *V. singularis* J. M. Watson & A. R. Flores, *V. spegazzinii* W. Becker, *V. stellaris,* ined., *V. tectiflora* W. Becker, *V. tholiformis,* ined., *V. tovarii* P. Gonzáles & Molina-Alor, *V. trochlearis* J. M. Watson & A. R. Flores, *V. umbrina,* ined., *V. volcanica* Gillies ex Hook. & Arn., *V. xanthopotamica* J. M. Watson & A. R. Flores

 


**[1.8] *Viola* sect. *Sempervivum* (**
Figure 3
**i and **
Figure 9
**n–x)**


*Viola* sect. *Sempervivum* J. M. Watson & A. R. Flores in Watson et al., Viola Subgenus Andinium: 188. 2021.—Type: *Viola atropurpurea* Leyb.

*Diagnostic characters*.—Perennial or annual subacaulous, glabrous, imbricated rosette-forming herbs. Lamina entire or shallowly subcrenulate, apex acute to obtuse.

*Ploidy and accepted chromosome counts*.—Unknown; gene homoeolog numbers indicate allopolyploidy in some species (T.M., unpubl.).

*Age*.—Crown node c. 13.3 Ma; stem node c. 20.3 Ma (Figure 6).

*Distribution*.—Ecuador to southern Patagonia.

*Included species*.—34. *Viola**abbreviata* J. M. Watson & A. R. Flores, *V. aizoon* Reiche, *V. atropurpurea* Leyb., *V. auricolor* Skottsb., *V. bangii* Rusby, *V. beckeriana* J. M. Watson & A. R. Flores, *V. columnaris* Skottsb., *V. comberi* W. Becker, *V. coronifera* W. Becker, *V. cotyledon* Ging., *V. cupuliformis* H. E. Ballard, ined. [T. Hofreiter & T. Franke 1/104], *V. dasyphylla* W. Becker, *V. hieronymi* W. Becker, *V. leyboldiana* Phil., *V. lologensis* (W. Becker) J. M. Watson, *V. marcelae,* ined., *V. micranthella* Wedd., *V. nigriflora* H. E. Ballard, ined. [T. Hofreiter & T. Franke 1/103], *V. nobilis* W. Becker, *V. obituaria* J. M. Watson & A. R. Flores, *V. pachysoma* M. Sheader & J. M. Watson, *V. petraea* W. Becker, *V. polycephala* H. E. Ballard & P. Jørg., *V. portulacea* Leyb., *V. pusillima* Wedd., *V. pygmaea* Juss. ex Poir., *V. regina* J. M. Watson & A. R. Flores, *V. rossowiana* J. M. Watson & A. R. Flores, *V. sacculus* Skottsb., *V. santiagonensis* W. Becker, *V. sempervivum* Gay, *V. skottsbergiana* W. Becker, *V. turritella* J. M. Watson & A. R. Flores, *V. vortex,* ined.

 


**[1.9] *Viola* sect. *Subandinium* (**
Figure 3
**j and **
Figure 9
**y–ac)**


*Viola* sect. *Subandinium* J. M. Watson & A. R. Flores in Watson et al., Viola Subg. Andinium: 193. 2021.—Type: *Viola subandina* J. M. Watson

*Diagnostic characters*.—Annual rosulate herbs. Lamina flexible, linear, oblanceolate or obovate, entire or shallowly long-crenulate. Diploid.

*Ploidy and accepted chromosome counts*.—2*x* (*Viola pusilla*); no chromosome counts.

*Age*.—Crown node c. 4.8 Ma; stem node c. 13.9 Ma (Figure 6).

*Distribution*.—Southern Chile to southern Peru.

*Included species*.—15. *Viola**araucaniae* W. Becker, *V. aurata* Phil., *V. auricula* Leyb., *V. domeikoana* Gay, *V. minutiflora* Phil., *V. nubigena* Leyb., *V. polypoda* Turcz., *V. pulvinata* Reiche, *V. pusilla* Poepp., *V. rhombifolia* Leyb., *V. subandina* J. M. Watson, *V. taltalensis* W. Becker, *V. vallenarensis* W. Becker, *V. weberbaueri* W. Becker, *V. yrameae* J. M. Watson & A. R. Flores, ined.

 


**[1.10] *Viola* sect. *Triflabellium* (**
Figure 3
**k and **
Figure 9
**ad)**


*Viola* sect. *Triflabellium* J. M. Watson, A. R. Flores & Marcussen in Watson et al., Viola Subg. Andinium: 192. 2021.—Type: *Viola triflabellata* W. Becker

*Diagnostic characters*.—Perennial rosette-forming herbs. Style crest as one apical and two lateral extended lobes.

*Distribution*.—Northern Chile to northwestern Argentina.

*Included species*.—7. *Viola**(Triflabellium)* sp. 1, ined., *V. flos-idae* Hieron., *V. joergensenii* W. Becker, *V. mesadensis* W. Becker, *V. triflabellata* W. Becker, *V. tucumanensis* W. Becker, *V. uniquissima* J. M. Watson & A. R. Flores

 

**[1.11] *Viola* sect. *Xylobasis* (**Figure 3l **and **Figure 9**ae)**

*Viola* sect. *Xylobasis* J. M. Watson & A. R. Flores in Watson et al., Viola Subg. Andinium: 191. 2021.—Type: *Viola beati* J. M. Watson & A. R. Flores

*Diagnostic characters*.—Perennial hairy, rosette-forming herbs. Stem shortly woody-branched.

*Distribution*.—Northwestern Argentina.

*Included species*.—1. *Viola beati* J. M. Watson & A. R. Flores

 


**[2] *Viola* subg. *Viola* (Figure 3m–av and Figure 9af–fe)**


=*Viola* sect. *Sparsifoliae* Reiche in Nat. Pflanzenfam., ed. 1 [Engler & Prantl], 3(6): 334. 1895, nom. inval. (Shenzhen Code Art. 22.2; *Viola odorata* L.)

*Description*.—Annual or perennial herbs, subshrubs or very occasionally treelets. Axes morphologically differentiated or not. Leaves scattered on stems or in rosettes, very occasionally imbricated with distichous phyllotaxy (sect. Tridens). Stipules free or partially adnate, sometimes large and foliaceous. Lamina usually petiolate; young laminas with involute margins (rarely folded in narrow leaves). Peduncles often longer than mature leaves. Bottom petal usually entire or shallowly emarginate, very rarely cleft. Spur absent to very long (34 mm). Nectariferous appendage of the two bottom stamens of various shape, rarely filiform. Style filiform, clavate, or capitate, not crested (but lateral lobes present in sect. Sclerosium) but top of style apex often flattened or with more or less raised edges, bearded or beardless. Cleistogamous flowers often produced.

*Diagnostic characters*.—Young laminas with involute margins OR peduncles longer than mature leaves OR style not crested OR cleistogamous flowers present.

*Ploidy and accepted chromosome counts*.—2*x*, 4*x*, 6*x*, 8*x*, 10*x*, 12*x*, 14*x*, 16*x*, 18*x*, 20*x*, >20*x*. 2*n* = 4, 8, 10, 12, 14, 16, 18, 20, 22, 24, 26, 28, 34, 36, 40, 44, c. 44, 46, 48, 50, 52, 54, 58, 60, c. 64, 72, 76, 80, c. 80, 82, c. 96, 102, c. 120, 128.

*Age*.—Crown node age 29.0 (28.3–29.4) Ma; stem node age 30.9 (29.8–31.3) Ma [28].

*Included species*.—525.

*Distribution*.—All continents except Antarctica. Diversity centres in e Asia, w Eurasia and N America.

*Discussion*.—Within subg. *Viola* we recognise 20 sections which can be grouped in three well-separated biogeographic clusters and allopolyploid tangles. The first cluster occurs in South and Central America and Australasia and comprises 43 species in 7 sections (sects. *Chilenium*, *Erpetion*, *Leptidium*, *Nematocaulon*, *Rubellium*, *Tridens*, and *Xanthidium*). The second cluster occurs primarily in the northern hemisphere and comprises 470 species in 12 sections (sects. *Abyssinium*, *Chamaemelanium*, *Danxiaviola*, *Delphiniopsis*, *Himalayum*, *Melanium*, *Nosphinium*, *Plagiostigma*, *Sclerosium*, *Spathulidium*, *Viola*, and *Xylinosium*). The third cluster occurs in South Africa with a single allopolyploid section and species (sect. *Melvio*; *V. decumbens*). The last two clusters are phylogenetically nested within the first one. Sections *Chamaemelanium* and *Rubellium* (2*n* = 12) are the only diploid lineages within subg. *Viola* (no data for sect. *Xanthidium*).

 


**[2.1] *Viola* sect. *Abyssinium* (**
Figure 3
**m and **
Figure 9
**af)**


*Viola* sect. *Abyssinium* Marcussen, sect. nov.—Type: *Viola abyssinica* Steud. ex Oliv.

*Description*.—Perennial herbs. Axes not morphologically differentiated. All stems ascending or trailing, rooting at proximal nodes. Stipules deeply dentate-laciniate to entire. Lamina crenulate, petiolate. Flowers c. 1 cm, peduncles produced only from some leaf axils. calycine appendages very short or absent. Corolla violet or white, with a white throat, bottom petal with violet striations. Spur saccate. Style clavate, laterally compressed, at base geniculate, at apex galeiform and distally margined, beardless. Cleistogamous flowers not produced. Allododecaploid (CHAM + MELVIO). Secondary base chromosome number *x’* = c. 36. ITS sequence of MELVIO type. 

*Diagnostic characters*.—All stems ascending or trailing AND corolla violet or white with white throat AND style clavate.

*Ploidy and accepted chromosome counts*.—12*x*; 2*n* = c. 72 (*Viola abyssinica*).

*Age*.—Crown node age c. 2 Ma; stem node age 3.6 (1.8–5.0) Ma [28].

*Included species*.—3. *Viola**abyssinica* Steud. ex Oliv., *V. eminii* (Engl.) R. E. Fr., *V. nannae* R. E. Fr.

*Distribution*.—High mountains of central and eastern Africa and Madagascar (Figure 11): *Viola abyssinica* throughout the range; *V. eminii* in eastern Congo, Rwanda, Burundi, Uganda to central and southern Kenya, northern Tanzania south to the Uluguru Mountains; V. nannae in central and southern Kenya [238].

*Etymology*.—The name Abyssinium refers to the main distribution area in and around Ethiopia (=Abyssinia).

*Discussion*.—Sect. *Abyssinium* is one of just two endemic African lineages of *Viola* (the other is the South African sect. *Melvio*). The count of 2*n* = c. 72 in *V. abyssinica* [239] is the only count for the section and needs confirmation. Section *Abyssinium* has an African distribution but is phylogenetically nested within the north hemisphere tangle of allopolyploid lineages. It appears to have originated in the Pliocene, from an allopolyploid of sect. *Spathulidium* (8*x*) and one of the 4*x* ancestors of that lineage (Figure 2 and [28]), which is distributed in southwestern Asia. The relatively recent origin of sect. *Abyssinium* from Eurasian ancestors fits a pattern commonly observed in Afrotemperate/Afromontane floral elements [240]. Becker [1] made a note that this group of species would merit a separate section, but he did not provide one. Possible hybridisation among the three species of sect. *Abyssinium* is briefly discussed by Grey-Wilson [238].

 


**[2.2] *Viola* sect. *Chamaemelanium* (**
Figure 3
**n and **
Figure 9
**ag–ay)**


*Viola* sect. *Chamaemelanium* Ging. in Mém. Soc. Phys. Genève 2(1): 28. 1823 ≡ *Viola* subg. *Chamaemelanium* (Ging.) Juz. in Schischk. & Bobrov, Flora URSS 15: 446. 1949—Type: *Viola canadensis* L.

≡*Lophion* Spach, Hist. Nat. Vég. [Spach] 5: 516. 1836 ≡ *Lophion* subg. *Eulophion* Nieuwl. & Kaczm. in Amer. Midl. Naturalist 3: 215. 1914, nom. inval. (Shenzhen Code Art. 22.2)—Type: *Viola canadensis* L.

=*Viola* sect. *Dischidium* Ging. in Mém. Soc. Phys. Genève 2(1): 28. 1823 ≡ *Dischidium* (Ging.) Opiz in Bercht. & Opiz, Oekon.-Techn. Fl. Böhm. [Berchtold & al.] 2(2): 7. 1839 ≡ *Viola* subg. *Dischidium* (Ging.) Peterm., Deutschl. Fl.: 65. 1846; (Ging.) Kupffer in Kusnezow et al., Fl. Caucas. Crit. 3(9): 172. 1909 (isonym); (Ging.) Juz. in Schisk. & Bobrov, Flora URSS 15: 441. 1949 (isonym) ≡ *Viola* [unranked] (”Gruppe”) *Dischidium* W. Becker in Beih. Bot. Centralbl., Abt. 2, 36: 38. 1918—Type: *Viola biflora* L.

≡*Chrysion* Spach, Hist. Nat. Vég. [Spach] 5: 509. 1836.—Type: *Viola biflora* L.

≡*Viola* [unranked] §.5. *Dischidieae* Boiss., Fl. Orient. 1: 452. 1867 ≡ *Viola* subsect. *Dischidieae* (Boiss.) Rouy & Foucaud, Fl. France [Rouy & Foucaud] 3: 36. 1896—Type: *Viola biflora* L.

=*Crocion* Nieuwl. & Kaczm. in Amer. Midl. Naturalist 3: 215. 1914—Type: *V. pubescens* Aiton

=*Viola* (sect. *Nomimium*) [unranked] (“Gruppa”) *Memorabiles* W. Becker in B. Fedtsch., Fl. Aziat. Ross. 8: 19. 1915 ≡ *Viola* sect. *Memorabiles* (W. Becker) Juz. in Schischk. & Bobrov, Flora URSS 15: 407. 1949—Type: *Viola kusnezowiana* W. Becker

=*Viola* “class” *Orbiculares* Pollard in Bot. Gaz. 26: 330. 1898, nom. inval. (Shenzhen Code Art. 33.9) ≡ *Viola* [unranked] *Orbiculares* W. Becker in Nat. Pflanzenfam., ed. 2 [Engler & Prantl], 21: 369. 1925 ≡ *Viola* subsect. *Orbiculares* (“Pollard“) Brizicky in J. Arnold Arb. 42: 326. 1961, nom inval. (Shenzhen Code Art. 41.5)—Type: *Viola orbiculata* Geyer ex Holz.

=*Viola* [unranked] D. *Erectae* W. Becker in Nat. Pflanzenfam., ed. 2 [Engler & Prantl], 21: 370. 1925 ≡ *Viola* sect. *Erectae* (W. Becker) Ching J.Wang, Fl. Reipubl. Popularis Sin. 51: 123. 1991.—Lectotype (designated here): *Viola acutifolia* (Kar. & Kir.) W. Becker

=*Viola* [unranked] (Untergruppe) *Longicalcaratae* W. Becker in Beih. Bot. Centralbl. 36(2): 38. 1918 ≡ *Viola* [sect. *Dischidium*; unranked] A. *Longicalcaratae* W. Becker in Nat. Pflanzenfam., ed. 2 [Engler & Prantl], 21: 370. 1925 ≡ *Viola* subsect. *Longicalcaratae* (W. Becker) W. Becker in Acta Horti Gothob. 2: 288. 1926 ≡ *Viola* subsect. *Longicalcaratae* (W. Becker) Ching J. Wang, Fl. Reipubl. Popularis Sin. 51: 119. 1991.—Lectotype (designated here): *Viola wallichiana* Ging.

*Description*.—Perennial herbs. Axes usually morphologically differentiated into a perennial rhizome and annual aerial stems. Rhizome usually deep-buried with a few-leaved apical rosette. Lateral stems aerial, rarely stolons, sometimes reduced or absent. Stipules partially to completely herbaceous or rarely membranous, margins entire or irregularly dentate with a few teeth. Lamina cordate to lanceolate, margin crenate, lobed, or pedately divided, usually long-petiolate. Corolla yellow, white, or violet, always with a yellow throat. Spur very short, rarely longer in a few Asian species. Style clavate or capitate, variable, usually bearded at apex. Cleistogamous flowers usually produced; cleistogamy seasonal. Diploid. Base chromosome number *x* = 6. ITS sequence of CHAM type.

*Diagnostic characters*.—Corolla with a yellow throat AND base chromosome number *x* = 6.

*Ploidy and accepted chromosome counts*.—2*x*, 4*x*, 6*x*, 8*x*, 12*x*; 2*n* = 12, 24, 36, 48, 72. 

*Age*.—Crown node age 19.0 (18.0–19.3) Ma [28].

Included species.—69. *Viola acutifolia* (Kar. & Kir.) W. Becker, *V. alliariifolia* Nakai, *V. allochroa* Botsch., *V. angkae* Craib, *V. aurea* Kell., *V. bakeri* Greene, *V. barroetana* W. Schaffn. ex Hemsl., *V. beckwithii* Torr. & A. Gray, *V. biflora* L., *V. brevistipulata* (Franch. & Sav.) W. Becker, *V. californica* M. S. Baker, *V. cameleo* H. Boissieu, *V. canadensis* L., *V. caucasica* (Rupr.) Kolen. ex Juz., *V. charlestonensis* M. S. Baker & J. C. Clausen, *V. coahuilensis* H. E. Ballard, ined. [P. Fryxell 2692], *V. confertifolia* C. C. Chang, *V. crassa* (Makino) Makino, *V. cuneata* S. Watson, *V. delavayi* Franch., *V. dimorphophylla* Y. S. Chen & Q. E. Yang, *V. douglasii* Steud., *V. eriocarpa* Schwein., *V. fischeri* W. Becker, *V. flagelliformis* Hemsl., *V. flettii* Piper, *V. franksmithii* N. H. Holmgren, *V. galeanaensis* M. S. Baker, *V. glabella* Nutt., *V. glaberrima* (Ging. ex Chapm.) House, *V. guadalupensis* A. M. Powell & Wauer, *V. hallii* A. Gray, *V. hastata* Michx., *V. hediniana* W. Becker, *V. kitamiana* Nakai, *V. kusnezowiana* W. Becker, *V. lithion* N. H. Holmgren & P. K. Holmgren, *V. lobata* Benth., *V. majchurensis* Pissjauk., *V. muehldorfii* Kiss, *V. muliensis* Y. S. Chen & Q. E. Yang, *V. nuttallii* Pursh, *V. ocellata* Torr. & A. Gray, *V. orbiculata* Geyer ex Holz., *V. orientalis* (Maxim.) W. Becker, *V. painteri* Rose & House, *V. pedunculata* Torr. & A. Gray, *V. pinetorum* Greene, *V. praemorsa* Douglas, *V. pubescens* Aiton, *V. purpurea* Kellogg, *V. quercetorum* M. S. Baker & J. C. Clausen, *V. rockiana* W. Becker, *V. rotundifolia* Michx., *V. rugulosa* Greene, *V. scopulorum* (A. Gray) Greene, *V. sempervirens* Greene, *V. sheltonii* Torr., *V. szetschwanensis* W. Becker & H. Boissieu, *V. tenuipes* Pollard, *V. tenuissima* C. C. Chang, *V. tomentosa* M. S. Baker & J. C. Clausen, *V. trinervata* (Howell) Howell ex A. Gray, *V. tripartita* Elliott, *V. uniflora* L., *V. urophylla* Franch., *V. utahensis* M. S. Baker & J. C. Clausen, *V. vallicola* A. Nelson, *V. wallichiana* Ging.

*Distribution*.—North America and east Asia; only *V. biflora* is roughly circumpolar (Figure 12).

*Discussion*.—Sect. *Chamaemelanium* is the only diploid representative of the CHAM genome; intrasectional allopolyploids are frequent but there was no hybridisation with the MELVIO lineage. The lineage is characterised karyologically by the base chromosome number *x* = 6 and morphologically by a plesiomorphic yellow corolla (variously coloured but always with a yellow throat in the *Canadenses* and *Chrysanthae* greges), shoots differentiated in a perennial (often deep-buried) rhizome with an apical (often few-leafed) leaf rosette and annual lateral floriferous stems, and the presence of seasonal cleistogamy. The lateral stems are usually more or less erect and aerial, in some reclining or prostrate and leafy or leafless (*V. kusnezowiana* in northeastern Asia, *V. orbiculata*, *V. rotundifolia* and *V. sempervirens* in North America), or entirely missing (*V. barroetana* in Mexico). Stipules in some species are semi-membranous or membranous, and are commonly entire or with one to few irregular teeth on one or both margins. Leaf lamina is usually crenate or crenulate but deeply divided in some taxa (greges *Chrysanthae* and *Nudicaules* in North America, the *V. biflora* group in northeastern Asia). Style shape is variable [29] but most species groups have a capitate, bearded style. Members of the *V. biflora* group (the former sect. *Dischidium*) have a bilobate style, while a few other species have style shapes resembling those found in other sections, such as sect. *Viola* (*V. kitamiana* and *V. kusnezowiana* in northeastern Asia) or sect. *Plagiostigma* (*V. rotundifolia* in eastern North America). Elaiosomes are highly reduced to obsolete in at least some species of the Canadenses grex. Cleistogamous flowers are missing in some taxa adapted to arid habitats (notably grex *Chrysanthae* and *V. guadalupensis*).

We recognise a broadly defined sect. *Chamaemelanium* that includes sect. *Dischidium* Ging. (i.e., the *V. biflora* group), grex *Orbiculares* Pollard (i.e., *V. orbiculata*, *V. sempervirens* and *V. rotundifolia*) and grex *Memorabiles* W. Becker (i.e., *V. kusnezowiana*) previously placed in sect. *Nomimium* by Becker [1], and *V. kitamiana*. The inclusion of *Dischidium* and *Orbiculares* in sect. *Chamaemelanium*, first suggested nearly a century ago by Clausen [29,59], is supported by morphology, chromosome counts, and by phylogeny (Figure 13) [28,60]. *Viola kusnezowiana* is included in sect. *Chamaemelanium* on basis of flower and stipule characters [61]; the somewhat emarginate lamina apex is particularly reminiscent of the *V. biflora* group. *Viola kitamiana* is included in sect. *Chamaemelanium* based on its corolla with a yellow throat (otherwise white) and the diploid chromosome number 2*n* = 12 [61]. 

We do not recognise infrasectional groups within sect. *Chamaemelanium* because its extant sublineages, at least the North American ones (Figure 13), are interconnected by allopolyploidy and therefore non-monophyletic [59,60,241,242]. Furthermore, the 7–8 diploid deep lineages do not correspond to any recognised morphological greges [1] and their interrelationships are deep and largely unresolved. For instance, both the capitate-bearded style shape and the rhizomatous habit with lateral, aerial floriferous stems, the two characters that define Becker’s grex *Erectae*, appear to be ancestral and plesiomorphic within sect. *Chamaemelanium* (Figure 13).

The initial radiation of sect. *Chamaemelanium* appears to have coincided with that of the CHAM + MELVIO allopolyploids in the northern hemisphere c. 19 Ma ago [28]. It has not been established whether the CHAM genomes involved in these allopolyploidisations were derived from within the extant sect. *Chamaemelanium* or from a lineage sister to it. It is, however, clear that the version of the CHAM genome present in the southern hemisphere sect. *Chilenium* and sect. *Erpetion* is sister to all other CHAM genomes.

The report of 2*n* = 20 in *V. kusnezowiana* [243] is at odds with the other counts in the section, all of which are based on *x* = 6, and in need of confirmation.

 


**[2.3] *Viola* sect. *Chilenium* (**
Figure 3
**o and **
Figure 9
**az–bc)**


*Viola* sect. *Chilenium* W. Becker in Nat. Pflanzenfam., ed. 2 [Engler & Prantl], 21: 376. 1925.—Lectotype (designated here): *Viola maculata* Cav.—Note: *Viola maculata* is indicated as “Haupttypus” in the protologue (Becker 1925: 376).

≡*Viola* [unranked] § I. *Bicaules* Reiche in Fl. Chile [Reiche] 1: 139. 1896—Lectotype (designated here): *Viola maculata* Cav.

*Description*.—Perennial herbs. Axes not morphologically differentiated. Stem creeping, more or less densely noded, deeply seated. Stipules free, broad and glandular-laciniate, rarely entire. Lamina oblong, elliptical or rhombic-lanceolate to reniform, margin crenate, long-petiolate. Bracteoles broad, deeply glandular-laciniate. Corolla yellow, rarely white (*V. commersonii*). Bottom petal at least twice as broad as top petals, rarely only slightly broader than the top petals (*V. rudolphii*), with brown striation, rarely reddish-violet striation (*V. commersonii*). Spur shorter than tall, rarely much longer than tall (*V. rudolphii* and *V. stuebelii*). Style clavate or straight, concave, flattened or slightly acute apically with a continuous sharp dorsolateral rim, the rim truncate and hiding the stigma (*V. stuebelii*) or slightly prolonged on the upper side with an upcurved visible rostellum bearing the stigma (other species), usually beardless (white-hairy in *V. rudolphii*). Cleistogamous flowers produced; cleistogamy facultative. Allopolyploid.

*Diagnostic characters*.—All stems rhizomes AND corolla yellow with brown striation or white with reddish-violet striation AND facultative cleistogamy.

*Ploidy and accepted chromosome counts*.—≥4*x*; no chromosome counts.

*Age*.—Crown node age unknown, stem node age 7.4 (6.5–7.7) Ma [28].

*Included species*.—7. *Viola**commersonii* DC. ex Ging., *V. germainii* Sparre, *V. maculata* Cav., *V. magellanica* G. Forst., *V. reichei* Skottsb. ex Macloskie, *V. rudolphii* Sparre, *V. stuebelii* Hieron.

*Distribution*.—Disjunct in southern (Argentina and Chile) and northern South America (Colombia, Ecuador, and Peru) (Figure 14).

*Discussion*.—We here modify Becker’s [1] original delimitation of sect. *Chilenium* by including *V. stuebelii* (=*V. glandularis* H. E. Ballard & P. Jørg.) based on shared diagnostic characters, and excluding *V. huidobrii* (=*V. brachypetala* Gay). Reiche [122,130] was the first to recognise this group, which he circumscribed under an invalid taxonomic rank (i.e., the unranked *Bicaules* within the invalid “Divisio” *Sparsifoliae*). Later, Sparre [62,63] revised the section and recognised eight southern South American species (some of them were later synonymised), which he distributed among three subsections, *Maculatae* (*V. germainii*, *V. maculata*, *V. reichei*), *Magellanicae* (*V. commersonii*, *V. magellanica*), and *Lanatae* (*V. rudolphii*), based on characteristics of the spur, style, and nectariferous appendages. We transfer the distinctive, violet-flowered *V. huidobrii* (Sparre as subsect. *Coeruleae*) to sect. *Viola* subsect. *Rostratae*. The new delimitation of sect. *Chilenium* renders the section geographically disjunct, with *V. stuebelii* in northern South America and the rest of the species in southern South America. Section *Chilenium* comprises only seven species, some closely related (e.g., *V. maculata* and *V. reichei*) and others known only from the type specimen (*V. germainii* and *V. rudolphii*), and in the absence of molecular data we choose not to keep Sparre’s subsections.

The South American sect. *Chilenium* is sister lineage of the Australian sect. *Erpetion* [28].

 


**[2.4] *Viola* sect. *Danxiaviola* (**
Figure 3
**p and **
Figure 9
**bd)**


*Viola* sect. *Danxiaviola* W. B. Liao & Q. Fan in Phytotaxa 197: 19. 2015—Type: *Viola hybanthoides* W. B. Liao & Q. Fan

*Description*.—Subshrub. Axes not morphologically differentiated. All stems erect or ascending. Stipules free, conspicuous, oblong-lanceolate, remotely long-fimbriate. Lamina elliptic or ovate-lanceolate, margin serrate, short-petiolate. Corolla whitish to pale violet. Bottom petal clawed, much larger than the other, reduced petals, whitish to pale violet with a yellowish green blotch at base. Spur short and saccate. Style capitate, at apex slightly bilobate, beardless, not beaked and with a stigmatic opening in front and with a lamellar processus below the opening. Cleistogamous flowers not produced. Chromosome number *x* = 10. ITS sequence of CHAM type.

*Diagnostic characters*.—Bottom petal clawed, much larger than the other petals.

*Ploidy and accepted chromosome counts*.—Probably 4*x*; 2*n* = 20.

*Age*.—Crown node age not applicable (monotypic section), stem node age probably 17.8–19.3 Ma.

*Included species*.—1. *Viola**hybanthoides* W. B. Liao & Q. Fan

*Distribution*.—Southeastern China (northern Guangdong). Known only from two sites on Mt. Danxia (Figure 15).

*Discussion*.—The single species in the section, *V. hybanthoides*, is phylogenetically isolated within the north hemisphere allopolyploid tangle, based on both ITS and chloroplast sequences [90]. The combined features of the larger bottom petal and much smaller lateral and upper petals is unique in *Viola* but found recurrently in most bilaterally symmetrical genera of the Violaceae, such as genera currently being segregated from the former polyphyletic *Hybanthus*, and sisters to *Viola*, *Noisettia* and *Schweiggeria* [3]. We infer that *V. hybanthoides* is probably a CHAM + MELVIO meso-allotetraploid, judging from its chromosome number (2*n* = 20) which in sect. *Viola* and sect. *Delphiniopsis* reflects 4*x*, the small size of its chromosomes and tricolporate pollen which both reflect a certain time since the polyploidisation, and its phylogenetic placement nested within a tetraploid clade [90] (Figure 6).

This morphologically highly unusual species was discovered as late as 2012 and published in 2015 [90] and was therefore included neither in the morphological treatment of Becker [1] nor in the phylogeny of Marcussen et al. [28].

 


**[2.5] *Viola* sect. *Delphiniopsis* (**
Figure 3
**q and **
Figure 9
**be–bf)**


*Viola* sect. *Delphiniopsis* W. Becker in Nat. Pflanzenfam., ed. 2 [Engler & Prantl], 21: 373. 1925—Type (Shenzhen Code Art. 10.8): *Viola delphinantha* Boiss.

≡*Viola* [unranked] §.1. *Delphinoideae* Boiss., Fl. Orient. 1: 451. 1867.—Type: *Viola delphinantha* Boiss.

≡*Viola* [unranked] c. *Lobulariae* Nyman, Consp. Fl. Eur. 1: 79. 1878.—Type: *Viola delphinantha* Boiss.

*Description*.—Perennial herbs with a woody base. Axes not morphologically differentiated. All stems aerial, annual, growing in fascicles from a woody and sometimes thick rhizome (pleiocorm). Leaves sessile, consisting of 3–5 lanceolate, entire segments, lamina and stipule segments similar. Corolla pink to magenta. Spur 12–30 mm, down-curved. Style clavate, glabrous, unmargined, with a simple, wide stigmatic opening. Cleistogamous flowers not produced. Allotetraploid (CHAM + MELVIO). Secondary base chromosome number *x’* = 10. ITS sequence of MELVIO type.

*Diagnostic characters*.—Corolla pink to magenta AND spur 12–30 mm, down-curved.

*Ploidy and accepted chromosome counts*.—4*x*; 2*n* = 20.

*Age*.—Crown node age unknown, stem node age 14.7 (6.3–18.6) Ma [28].

*Included species*.—3. *Viola**cazorlensis* Gand., *V. delphinantha* Boiss., *V. kosaninii* (Degen) Hayek.

*Distribution*.—Disjunct in southern Europe: southern Spain (*V. cazorlensis*) and the Balkans (*V. delphinantha*, *V. kosaninii*) (Figure 16).

*Discussion*.—Section *Delphiniopsis* is highly distinct, phylogenetically, karyologically (*x’* = 10), and morphologically. The species are specialised to be pollinated by a single species of day-flying hawk-moth (*Macroglossum stellatarum*, Sphingidae, Lepidoptera) [175,176]. The disjunct distribution of sect. *Delphiniopsis*, between *V. delphinantha* and *V. kosaninii* in the Balkans and *V. cazorlensis* in southern Spain, has been suggested to result from vicariance and to date from the Early Pliocene, 3.6–5.3 Ma [244]. The crown age of the section have so far not been phylogenetically dated, but the idea that the species are young is further supported by their morphological similarity and reports of their being able to hybridise in culture (plants of *V. cazorlensis* × *V. delphinantha* were displayed at the Midland AGS SHOW 2012). Phylogenetic analysis confirms that sect. *Delphiniopsis* is an isolated and highly specialised lineage on a rather long branch, indicating rapid evolution [28]. The characters once considered ”primitive” [244,245], such as woodiness, entire leaves and stipules, small and uniform chromosomes and rather common chromosome number, relict distribution, and lack of cleistogamy, should rather be interpreted as secondary specialisations, just like the highly specialised pollination. The low and apparently young diversity of the section may be explained by its high level of specialisation in both pollination syndrome and choice of habitat: the species inhabit limestone crevices, a rare habitat that minimises competition but at the same time limits the population size and dispersal, thereby increasing the risk of extinction.

 


**[2.6] *Viola* sect. *Erpetion* (**
Figure 3
**r and **
Figure 9
**bg)**


*Viola* sect. *Erpetion* (DC. ex Sweet) W. Becker in Nat. Pflanzenfam., ed. 2 [Engler & Prantl], 21: 376. 1925 ≡ *Erpetion* DC. ex Sweet, Brit. Fl. Gard. 2: nr. 170. 1826 ≡ *Viola* subg. *Erpetion* (DC. ex Sweet) Y. S. Chen in Raven & Hong, Fl. China 13: 111. 2007—Type: *Viola hederacea* Labill.

*Description*.—Perennial herbs. Axes seemingly morphologically differentiated in a perennial stem with lateral sympodial stolons. Perennating stem densely or occasionally remotely noded. Sympodial stolons with a pair of bracts between each cluster of leaves. Stipules small, lanceolate. Lamina ovate-rhomboid to broadly reniform, margin crenate, long-petiolate. Corolla white to dark violet, often with a darker throat; corolla sometimes highly reduced. Spur reduced to a gibba and a green blotch on the inside of the bottom petal. Lateral petals with a broad dense pad of papillae. Style filiform, beardless. Cleistogamous flowers not produced. Allo-octoploid. Secondary base chromosome number *x’* = 25.

*Diagnostic characters*.—Sympodial stolons present. Spur reduced to a gibba. Lateral petals with a broad dense pad of papillae. 

*Ploidy and accepted chromosome counts*.—8*x*, 16*x*, 24*x*; 2*n* = 50 (*V. banksii*).

*Age*.—Crown node age 3.7 (3.2–3.9) Ma, stem node age 7.4 (6.5–7.7) Ma [28].

*Included species*.—11. *Viola**banksii* K. R. Thiele & Prober, *V. cleistogamoides* (L. G. Adams) Seppelt, *V. curtisiae* (L. G. Adams) K. R. Thiele, *V. eminens* K. R. Thiele & Prober, *V. fuscoviolacea* (L. G. Adams) T. A. James, *V. hederacea* Labill., *V. improcera* L. G. Adams, *V. perreniformis* (L. G. Adams) R. J. Little & Leiper, *V. serpentinicola* de Salas, *V. sieberiana* Spreng., *V. silicestris* K. R. Thiele & Prober

*Distribution*.—Southern and eastern Australia; Tasmania (Figure 17).

*Discussion*.—Phylogenetically, sect. *Erpetion* is an allo-octoploid lineage with two CHAM genomes and another two genomes in common with sect. *Chilenium*, indicating that sect. *Erpetion* experienced a second genome duplication after the two sections diverged. There is no indication that this ancestral tetraploid *Erpetion* still exists. Section *Erpetion* is characterised karyologically by the secondary base chromosome number *x’* = 25 [98]. The estimate of 10*x* for sect. *Erpetion* by Marcussen et al. [28] was based on unconfirmed (and probably erroneous) counts of 2*n* = 60 and 2*n* = 120 on “representatives of the *Viola hederacea* complex in the Kosciusko area” by Moore in [246].

Members of sect. *Erpetion* can be recognised immediately by two unique synapomorphies, i.e., the presence of sympodial stolons, which differ from true stolons by their clustered leaves and bibracteolate stem segments, and the pad of papillae on the lateral petals in place of the beard of trichomes some members of other lineages exhibit. Anatomically, the sympodial stolon consists of a potentially infinite chain of bibracteolate stem segments each ending in a leaf rosette, which in turn produces a new segment from the axil of its lowermost leaf. Adventitious roots are produced at the base of each rosette only. In *Fragaria* (Rosaceae), both sympodial and monopodial stolons can be found among closely related species (e.g., *F. viridis* vs. *F. vesca*, respectively), suggesting that the underlying genetics can be quite simple. 

We follow the original delimitation of Becker [1] for the section. At the time only one variable species was recognised, *Viola hederacea*, but c. 11 species are now recognised [160,226]. Genome size data (2C DNA) indicate that sect. *Erpetion* forms a polyploid series based on 8*x*, i.e., with *V. banksii* at the 8*x* level (two accessions with 1.26 and 1.27 pg), *V. fuscoviolacea* at the 16*x* level (2.57 pg), and *V. hederacea* at the 24*x* level (3.45 pg; T.M., unpublished data, and [98]). Indeed, the occurrence of autogamous taxa with very small corollas, i.e., *V. cleistogamoides* and *V. fuscoviolacea*, agree with the observation of high ploidy in this section. A cultivar attributed to *V. banksii* is frequently grown as an ornamental, but appears to be a hybrid, based on having low pollen fertility [98]. 

The sister lineage of sect. *Erpetion* is the South American sect. *Chilenium* [28], from which it may have diverged c. 7.4 Ma ago [28]. This relationship is surprisingly from a morphological perspective, as the two taxa are rather dissimilar and lack obvious synapomorphies.

 


**[2.7] *Viola* sect. *Himalayum* (**
Figure 3
**s and **
Figure 9
**bh)**


*Viola* sect. *Himalayum* Marcussen, sect. nov.—Type: *Viola kunawurensis* Royle

*Description*.—Dwarf perennial herb. Axes not morphologically differentiated. Stems subterranean from deeply buried pleiocorm, appearing aboveground as proximal or tufted rosettes. Stipules adnate to ¾ of their length. Lamina subentire with 0–2 shallow crenulae, spathulate, gradually tapering in a long petiole. Corolla c. 10 mm, light violet with dark striations. Lateral petals beardless. Spur as long as tall, saccate, 1–1.5 mm, obtuse. Style clavate, unmargined. Cleistogamous flowers produced; cleistogamy seasonal. Allo-octoploid (CHAM + MELVIO). Secondary base chromosome number *x’* = 10 (needs confirmation). ITS sequence of MELVIO type. 

*Diagnostic characters*.—Stipules adnate AND lamina spathulate and subentire AND spur as long as tall, 1–1.5 mm AND cleistogamous flowers produced.

*Ploidy and accepted chromosome counts*.—8*x*; 2*n* = 20?

*Age*.—Crown node age not applicable (monotypic section), stem node age probably 17.8–19.3 Ma.

*Included species*.—1. *Viola kunawurensis* Royle

*Distribution*.—High mountains surrounding the Tibetan Plateau: Tian Shan, Pamir, the Himalayas, Hengduan Shan, and Qilian Shan (Figure 18).

*Etymology*.—The name *Himalayum* refers to the distribution in the Himalayas and adjacent mountain ranges.

*Discussion*.—Section *Himalayum* comprises a single species, *V. kunawurensis* (=*V.* “*kunawarensis*”, *V. thianschanica* Maxim.), occurring at high elevations (3000–5000 m) in the Central Asian high mountains surrounding the Tibetan plateau (Figure 18). *Viola kunawurensis* differs from similar species of sect. *Plagiostigma* subsect. *Patellares* in having a very short spur and frequently elongated internodes arising from the deep-buried pleiocorm, as well as in chromosome number, and from sect. *Spathulidium* in style shape and in producing cleistogamous flowers. 

Mining *GPI* sequences from the sequence reads archive of the reference sequence genome of *V. kunawurensis* (as *V.* “*kunawarensis*”; NCBI accession PRJNA805692) strongly indicates the presence of four homoeologs, confirming that sect. *Himalayum* is an independent CHAM + MELVIO allotetraploid lineage and further suggesting that the extant species is octoploid as a result of a secondary autopolyploidisation (Figure 6). The single chromosome count of 2*n* = 20 [247] is doubtful as this number reflects 4*x* in sect. *Viola* and sect. *Delphiniopsis* and therefore seems at odds with the octoploid condition of *V. kunawurensis*.

Becker originally placed *Viola kunawurensis* in grex *Gmelinianae* [248], later in grex *Adnatae* [1]; see note under sect. *Plagiostigma* subsect. *Patellares*. Sun and coworkers placed *V. kunawurensis* in sect. *Viola* subsect. *Rostratae* based on the (allegedly) shared chromosome number 2*n* = 20 [247] and numerical taxonomy of 58 traits [77].

 


**[2.8] *Viola* sect. *Leptidium* (**
Figure 3
**t and **
Figure 9
**bi)**


*Viola* sect. *Leptidium* Ging. in Mém. Soc. Phys. Genève 2(1): 28. 1823 ≡ *Viola* subg. *Leptidium* (Ging.) Peterm., Deutschl. Fl.: 66. 1846—Type: *Viola stipularis* Sw.

*Description*.—Subshrubs or perennial herbs. Axes not morphologically differentiated. Stems reclining to erect, sometimes branched (in *Viola scandens* and *V. stipularis* at least 1 m long). Stipules lanceolate to ovate, laciniate, partially sheathing the stem. Lamina linear-lanceolate to reniform, margin crenate, short- to long-petiolate. Corolla whitish to violet with a white throat (corolla entirely red in *V. arguta*). Spur short and saccate (spur thick and bulbous in *V. arguta*). Bottom pair of stamens with apical “u”-shaped connective appendage. Style filiform, straight undifferentiated, with a simple stigmatic opening. Cleistogamous flowers produced, sometimes subterranean in *V. arguta* and possibly other species; cleistogamy facultative. Allotetraploid. Inferred secondary base chromosome number [*x’* = 13.5].

*Diagnostic characters*.—Aerial stems AND laciniate sheathing stipules AND short saccate or thick bulbous spur AND “u”-shaped connective appendage on bottom pair of stamens AND filiform style.

*Ploidy and accepted chromosome counts*.—4*x*, 8*x*; 2*n* = 54 (*V. dombeyana*).

*Age*.—Crown node age 8.7 (3–16) Ma [28].

*Included species*.—18. *Viola**arguta* Humb. & Bonpl. ex Schult., *V. atroseminalis* H. E. Ballard, ined., *V. boliviana* Britton, *V. bridgesii* Britton, *V. cerasifolia* A. St.-Hil., *V. dombeyana* DC. ex Ging., *V. fuscifolia* W. Becker, *V. gracillima* A. St.-Hil., *V. lehmannii* W. Becker ex H. E. Ballard & P. Jørg., *V. mandonii* W. Becker, *V. saccata* Melch., *V. scandens* Humb. & Bonpl. ex Schult., *V. steinbachii* W. Becker, *V. stipularis* Sw., *V. subdimidiata* A. St.-Hil., *V. thymifolia* Britton, *V. uleana* W. Becker, *V. veronicifolia* Planch. & Linden

*Distribution*.—Southeastern Mexico to Bolivia; northwestern Venezuela; southeastern Brazil (Figure 19).

*Discussion*.—Section *Leptidium* is an allotetraploid (4*x*) lineage, derived from ancient hybridisation and chromosome doubling of the common ancestor of subgenus *Viola* and the most recent common ancestor of sect. *Leptidium* and sect. *Tridens*; this allopolyploidisation may have happened c. 15 Ma ago [28]. A comprehensive phylogeny of sect. *Leptidium* has not been published. While *V. arguta* appears to be 4*x*, further allopolyploidisation has occurred in *V. stipularis* (8*x*). The count of *n* = 27 in *V. dombeyana* (as *V. humboldtii* Tr. & Pl. [53]), presumably referring to the 8*x* level as well, is the only count for the section and needs confirmation.

This widely distributed Latin American lineage encompasses 17 species and possibly the mysterious *V. producta* W. Becker. *Viola scandens* and *V. stipularis* account for the Mesoamerican and Antillean portions of the range of the section, with four species in southeastern Brazil and 13 (14?) occupying middle and higher elevations of the northern and central Andean Mountains in South America. All species have petals glabrous within, and all share a peculiar synapomorphy of prolonged “u”-shaped anther connective appendages on the bottom pair of stamens, first documented in two Brazilian species by Freitas and Sazima [171]. The Mesoamerican and southeastern Brazilian lineages may have diverged 8.7 (3–16) Ma ago [28].

A transition from nectar to pollen flowers and “buzz” pollination has been suggested for the majority of the species within sect. *Leptidium*; the unique “u” shape of the connective stamen appendages appears to be an adaptation to this [171]. The flowers are pollinated by *Anthrenoides* bees (Andrenidae) that hold onto the connective appendages while harvesting pollen by vibration (“buzz-pollination”) [171], and may thus be analogous to similar structures in unrelated genera (e.g., *Arbutus*, Ericaceae). Curiously, a secondary transition to hummingbird pollination, unique in the genus, seems to have occurred in *V. arguta*; this species produces copious amounts of nectar, 4 µL per 24 h [174] but it has also preserved the “u”-shaped connective appendages indicative of ancestal “buzz”-pollination.

 

**[2.9] *Viola* sect. *Melanium* (**Figure 3**u**–**y and **Figure 9**bj–bm)**

*Viola* sect. *Melanium* Ging. in Mém. Soc. Phys. Genève 2(1): 28. 1823 ≡ *Viola* subg. *Melanium* (Ging.) Peterm., Deutschl. Fl.: 65. 1846; (Ging.) Kupffer in Kusnezow et al., Fl. Caucas. Crit. 3(9): 221. 1909 (isonym)—Type: *Viola tricolor* L.

≡*Mnemion* Spach, Hist. Nat. Vég. [Spach] 5: 510. 1836—Lectotype (Nieuwland & Kaczmarek 1914 [249], page 210): *Viola tricolor* L.

≡*Viola* sect. *Pogonostylos* Godron, Fl. Lorraine, ed. 2, 1: 90. 1857, nom. illeg. superfl. (Szhenzhen Code Art. 52.1; *Viola tricolor* L.)

≡*Viola* sect. *Novercula* Kupffer in Kusnezow et al., Fl. Caucas. Crit. 3(9): 225. 1909, nom illeg. superfl. (Szhenzhen Code Art. 52.1; *Viola tricolor* L.)

=*Jacea* Opiz in Bercht. & Opiz, Oekon.-Techn. Fl. Böhm. [Berchtold & al.] 2(2): 8. 1839, nom. illeg., non Mill., Gard. Dict. Abr., ed. 4: [not paginated]. 1754 (=*Centaurea*)

*Description*.—Annual to perennial herbs. Taproot preserved, in perennials often deeply buried and thickened. Axes not morphologically differentiated. All stems more or less aerial; in perennials proximal portion rhizome-like. Stipules usually foliaceous, pinnately or palmately lobed with leaflike segments. Lamina entire, crenulate or crenate, petiolate. Corolla small or large (bottom petal 2–34 mm), often varicoloured and/or variegated, nearly always with a yellow throat. Spur short or long and slender (0.9–16 mm). Calycine appendages short or long (0.5–5.3 mm). Style capitate, bearded. Cleistogamous flowers usually not produced; if produced, then cleistogamy seasonal (*V. rafinesquei*). Allotetraploid (CHAM + MELVIO). ITS sequence of MELVIO type. Aneuploid.

*Diagnostic characters*.—All stems more or less aerial AND stipules usually foliaceous AND corolla small to very large, nearly always with a yellow throat. 

*Ploidy and accepted chromosome counts*.—4*x*, 8*x*, 12*x*, 16*x*, 20*x*, >20*x*; 2*n* = 4, 8, 10, 12, 14, 16, 18, 20, 22, 24, 26, 28, 34, 36, 40, 48, 52, c.64, c.96, 120, c. 128.

*Age*.—Crown node age 12.5 (11.8–12.8) Ma [28].

Included species.—112.

*Distribution*.—Western Eurasia; one species in eastern North America (Figure 20). Mainly in mountainous areas, with a centre of diversity in the mountains of the Balkans, Apennine Peninsula and Sicily, seven species in northwestern Africa (three of them endemic) and one species in eastern North America (*Viola rafinesquei*). A few species are widespread in the lowlands, nearly all annuals or biennials (e.g., *V. arvensis*, *V. tricolor*, and *V. rafinesquei*).

*Discussion*.—Section *Melanium* is phylogenetically an allotetraploid CHAM + MELVIO lineage having retained the MELVIO homoeolog for *ITS* (Figure 6). Morphologically the section is highly characteristic, primarily by the annual or perennial habit with undifferentiated stems, the often large and leaf-like, usually deeply divided stipules, the orbicular, ovate, lanceolate or linear, crenate (or entire) laminas, often also by pronounced heterophylly, the small to very large, often multicoloured, corolla with a yellow throat (cream throat in, e.g., *V. argenteria*, *V. cornuta*, and *V. orthoceras*), the unique capitate-bearded style, and the absence of cleistogamous flowers (present in *V. rafinesquei*). Section *Melanium* is morphologically distinct and has by numerous authors been ranked as subgenus or even genus (*Mnemion* Spach and *Jacea* Opiz non Mill.). However, molecular data place it firmly among the other north hemisphere allotetraploid lineages, albeit with very long branches, suggesting that its morphological differentiation is a result of rapid evolution rather than deep divergence. The subtending branch of *Melanium* is also long and its diversification did not start until 12–13 Ma ago, corresponding with the onset of a global climatic cooling trend from c. 14 Ma ago [31]. Subsection *Bracteolatae*, which comprises most of the species, started diversifying 9–10 Ma ago [28]. In line with a relatively recent origin, the detailed evolutionary relationships within sect. *Melanium* remain elusive when based on markers such as *ITS*, chloroplast loci, and ISSRs [94,217]. The low-copy genes used by Marcussen et al. [28] revealed high ploidy levels for the three species sampled. These findings are corroborated also by the occurrence of numerous loci for ribosomal DNA in the section [250,251,252]. At present (2022), transcriptome data exist for *Viola tricolor* only [236].

Section *Melanium* is characterised by an extraordinarily high karyological diversity and plasticity which imply that ploidy can not be inferred from chromosome numbers alone. Here, we estimate the ploidy of 12 taxa in sect. *Melanium* (Table 2) from the number of homoeologs of the low-copy nuclear gene *GPI* [28] and genome size estimated from flow-cytometry [218,219,220,221,253] and for the Earth Biogenome Project (EBP [233]). These data in combination indicate that subsect. *Ebracteatae* is 4*x* (*V. dirimliensis*), subsect. *Cleistogamae* is 8*x* (*V. rafinesquei*), and that subsect. *Bracteolatae* comprises several ploidy levels, at least 8*x* (*V. kitaibeliana*, *V. beckiana*, *V. elegantula*, *V. cornuta*), 12*x* (*V. tricolor*), 16*x* (*V. arvensis*, *V. calcarata*), and 20*x* (*V. lutea*, *V. bubanii*). Given that the 1Cx-values within subsect. *Bracteolatae* are stable around 0.27–0.30 pg, changes in chromosome number in these taxa are due to chromosome fusions rather than to loss or deletions. Still, karyotype homology seems preserved at least in some allopolyploids containing the homologous ancestral 4x genomes, because considerably good chromosome pairing during the meiosis and fertility occurs in some heteroploid hybrids (e.g., [37,66,254,255]). Such chromosome fusions appear to occur throughout sect. *Melanium*, and are at the most extreme in subsect. *Ebracteatae*: in the presumably tetraploid *V. modesta* (2*n* = 4) the ancestral four monoploid genomes have probably fused to just four chromosomes. The highest widespread chromosome number is 2*n* = 52 (20*x*), found in six species of subsect. *Bracteolatae*, while chromosome numbers above this value (e.g., 2*n* = 64, 96, 120, and 128, the last two in *V. bubanii*) may have been counted in hybrids or aberrant individuals and require confirmation. It seems futile to try estimating the base chromosome number *x* for sect. *Melanium*, knowing that the nascent *Melanium* allotetraploid likely started out with *n* = 10 to 12 chromosomes just like the other CHAM + MELVIO tetraploids, and knowing that reductions in chromosome number have occurred independently in different sublineages of this section. Not surprisingly, each attempt until now has produced a different *x*, i.e., *x* = 5 [217], *x* = 6 [56], *x* = 7 [217], *x* = 10 [56], and *x* = 11 [255].

Although a detailed revision of sect. *Melanium* must await comprehensive phylogenomic study of the lineage, the already available data from phylogeny [28,94,217], morphology, and genome size and pollen aperture number which reflects ploidy [157], yield sufficient resolution to delimit five sublineages within sect. *Melanium* (Table 3). These lineages form morphologically and biogeographically recognisable units but do not conform with previous classifications (e.g., [1,21,72,133]). Below we formally introduce them as subsections, i.e., (1) subsect. *Bracteolatae*, (2) subsect. *Cleistogamae*, (3) subsect. *Dispares*, (4) subsect. *Ebracteatae*, and (5) subsect. *Pseudorupestres*. The vast majority of species belong in subsect. *Bracteolatae* and only a dozen in the other four subsections which may well be considered relictual and phylogenetically isolated.

**Key to the subsections of sect. *Melanium***:1a.Cleistogamous flowers produced. Annual or biennial. (eastern North America) ..................................................................................................................................................................................................................... **subsect. *Cleistogamae*** (*V. rafinesquei*)1b.Cleistogamous flowers not produced. Annual to perennial. (Palaearctic, elsewhere alien) .................................................................................................................................................................................................................................................................................. 2.2a.Corolla violet, with a cream-coloured throat. Stipules ovate-lanceolate, dentate. Bottom petal 9.5–10.5 mm. Low, high-Alpine perennial. (western Alps and Corsica) ....................................................................................................................................................................................................................... **subsect. *Pseudorupestres*** (*V. nummulariifolia*)2b.Corolla colour various, with a bright yellow throat (if throat cream or white and lateral petals directed horizontally or downwards: *V. cornuta* and *V. orthoceras*). Stipules variable, often foliaceous, rarely dentate. Bottom petal 2–34 mm. Annual or perennial. ..................................................................................................................................................................................................................................................................... 3.3a.Annual. Basal leaves entire or indistinctly crenulate. Bottom petal 2–11.5 mm. Spur 0.9–3 mm. ......................................................................................................................................................................................................................................................... **subsect. *Ebracteatae***3b.Annual to perennial. Lamina crenate or entire, but if annual then lamina of basal leaves crenate. Bottom petal 5–34 mm. Spur 1–16 mm. .................................................................................................................................................................................................................................................................................... 4.4a.Calycine appendages 0.3–1.0 mm long. Bottom petal 5–13 mm. Spur 1–3.5 mm. .......................................................................................................................................................................................................................................................... **subsect. *Dispares***4b.Calycine appendages 0.9–4.7 mm long. Bottom petal 5.4–34 mm. Spur 1.8–16 mm. ........................................................................................................................................................................................................................................................ **subsect. *Bracteolatae***

 


**[2.9.1] Viola sect. *Melanium* subsect. *Bracteolatae* (**
Figure 3
**u and **
Figure 9
**bj–bk)**


*Viola* subsect. *Bracteolatae* Kupffer in Kusnezow et al., Fl. Caucas. Crit. 3(9): 228. 1909—Lectotype (designated here): *Viola tricolor* L.

≡*Viola* subsect. *Melanium* (Ging.) Vl. V. Nikitin in Bot. Zhurn. (Moscow & Leningrad) 83(3): 135. 1998—Type: *Viola tricolor* L.

=*Viola* sect. *Pseudonovercula* Kupffer in Kusnezow et al., Fl. Caucas. Crit. 3: 222. 1909—Type: *Viola cornuta* L.

*Description*.—Annual to perennial. Lamina of basal leaves entire or crenate, but if plants annual then lamina crenate. Calycine appendages 0.9–4.7 mm. Corolla with bright yellow, rarely pale yellow throat. Bottom petal (spur included) 5.5–34 mm. Spur 1.8–16 mm. Cleistogamous flowers not produced. Pollen apertures 4 or 5 heteromorphic. Ploidy 8*x*, 12*x*, 16*x*, 20*x*, >20*x*.

*Diagnostic characters*.—See Table 3 and key.

Ploidy and accepted chromosome counts.—8*x*, 12*x*, 16*x*, 20*x*, >20*x*; 2*n* = 16, 18, 20, 22, 24, 26, 28, 34, 36, 40, 48, 52, c. 64, c. 96, 120, c. 128. 

*Age*.—Crown node c. 4 Ma (Figure 6), probably an underestimate; stem node age 9.8 (9.1–10.0) Ma [28].

*Included species*.—98 (in addition the two ornamental hybrids *Viola* ×*williamsii* Wittr. and *V*. ×*wittrockiana* Gams). *Viola*
*acrocerauniensis* Erben, *V. aethnensis* (Ging.) Strobl, *V. aetolica* Boiss. & Heldr., *V. albanica* Halácsy, *V. allchariensis* Beck, *V. alpina* Jacq., *V. altaica* Ker Gawl., *V. arsenica* Beck, *V. arvensis* Murray, *V. athois* W. Becker, *V. babunensis* Erben, *V. beckiana* Fiala ex Beck, *V. bertolonii* Pio, *V. bornmuelleri* Erben, *V. brachyphylla* W. Becker, *V. bubanii* Timb.-Lagr., *V. calcarata* L., *V. cenisia* L., *V. cephalonica* Bornm., *V. cheiranthifolia* Bonpl., *V. comollia* Massara, *V. cornuta* L., *V. corsica* Nyman, *V. crassifolia* Fenzl, *V. crassiuscula* Bory, *V. cryana* Gillot, *V. culminis* F. Fen. & Moraldo, *V. dacica* Borbás, *V. declinata* Waldst. & Kit., *V. dichroa* Boiss., *V. diversifolia* (Ging.) W. Becker, *V. doerfleri* Degen, *V. dubyana* Burnat ex Gremli, *V. dukadjinica* W. Becker & Košanin, *V. elegantula* Schott, *V. epirota* (Halácsy) Raus, *V. etrusca* Erben, *V. euboea* Halácsy, *V. eugeniae* Parl., *V. eximia* Formánek, *V. ferrarinii* Moraldo & Ricceri, *V. fragrans* Sieber, *V. frondosa* (Velen.) Velen., *V. ganiatsasii* Erben, *V. gostivariensis* Bornm., *V. gracilis* Sm., *V. graeca* (W. Becker) Halácsy, *V. grisebachiana* Vis., *V. guaxarensis* M. Marrero, Docoito Díaz & Martín Esquivel, *V. heldreichiana* Boiss., *V. henriquesii* (Willk. ex Cout.) W. Becker, *V. herzogii* (W. Becker) Bornm., *V. hispida* Lam., *V. hymettia* Boiss. & Heldr., *V. ivonis* Erben, *V. kitaibeliana* Schult., *V. kopaonikensis* Pančić ex Tomović & Niketić, *V. langeana* Valentine, *V. lutea* Huds., *V. magellensis* Porta & Rigo ex Strobl, *V. merxmuelleri* Erben, *V. minuta* M. Bieb., *V. montcaunica* Pau, *V. munbyana* Boiss. & Reut., *V. nana* (DC. ex Ging.) Le Jol., *V. nebrodensis* C. Presl, *V. odontocalycina* Boiss., *V. orbelica* Pančić, *V. oreades* M. Bieb., *V. orphanidis* Boiss., *V. orthoceras* Ledeb., *V. palmensis* (Webb & Berthel.) Sauer, *V. paradoxa* Lowe, *V. parnonia* Kit Tan, Sfikas & Vold, *V. perinensis* W. Becker, *V. phitosiana* Erben, *V. pseudaetolica* Tomović, Melovski & Niketić, *V. pseudogracilis* (A. Terracc.) Strobl ex Degen & Dörfl., *V. pseudograeca* Erben, *V. raunsiensis* W. Becker & Košanin, *V. rausii* Erben, *V. rhodopeia* W. Becker, *V. roccabrunensis* Espeut, *V. samothracica* (Degen) Raus, *V. schariensis* Erben, *V. serresiana* Erben, *V. sfikasiana* Erben, *V. slavikii* Formánek, *V. stojanowii* W. Becker, *V. striis-notata* (J. Wagner) Merxm. & W. Lippert, *V. subatlantica* (Maire) Ibn Tattou, *V. tineorum* Erben & Raimondo, *V. tricolor* L., *V. ucriana* Erben & Raimondo, *V. valderia* All., *V. velutina* Formánek, *V. voliotisii* Erben, *V. vourinensis* Erben

*Distribution*.—Western Eurasia.

*Discussion*.—Sect. *Melanium* subsect. *Bracteolatae* comprises the vast majority of the species in the section and is difficult to describe (Table 3). The lineage is phylogenetically characterised by being at least 8-ploid (Table 2), karyologically by a variety of chromosome numbers, and morphologically by the sometimes very large corollas with bottom petal up to 32 mm long, and usually heteromorphic mostly 4-colporate, rarely 5-colporate pollen [157]. The diversification in subsect. *Bracteolatae* is evidently recent and may at least partly have been driven by geographic isolation in combination with homoploid and heteroploid hybrid speciation [254], as indicated from chromosome counts, crossing experiments [254], genome size variation (Table 2), and subcloning of nuclear genes and ribotypes [28,250,252]. Not surprisingly, the two phylogenies of sect. *Melanium* [94,217], both using ITS, showed little variation among species. The evolutionary relationships within subsect. *Bracteolatae* are poorly understood. However, our preliminary interpretation based on all available lines of evidence is that the subsection comprises at least 3–4 independent homoeologous genome lineages that occur in different variants, numbers and combinations among the different species. In some cases the shared subgenomes are similar enough to allow for gene flow among different species despite differences in ploidy, such as between *V. tricolor* (2*n* = 26; 12*x*) and *V. arvensis* (2*n* = 34; 16*x*), whereas in other species pairs the subgenomes are too dissimilar to allow for gene flow or even hybrid formation, such as between *V. tricolor* and *V. cornuta* (2*n* = 22; 8*x*) [254]. The available morphological, genetical [254] and molecular evidence from ITS [94] and 5S-IGS [252] suggest that, for instance, *V. heldreichiana*, *V. kitaibeliana*, *V. hymettia* (all 2*n* = 16; 8*x*), *V. tricolor* (12*x*) and *V. arvensis* (16*x*) form a polyploid series. Furthermore, the species with 2*n* = 20 (8*x*) and 2*n* = 40 (16*x*) of the Alps, Dinarids, Apeninnes and Sicily, traditionally referred to as the *V. calcarata* group [94], are probably closely related. The Pyrenean *V. cornuta* and the Caucasian *V. orthoceras* (both with several shared, rather unique character states; 2*n* = 22) are probably geographic isolates. *Viola tricolor* and *V. arvensis* are cosmopolitan weeds. *Viola* ×*williamsii* and *V*. ×*wittrockiana* are grown as ornamentals.

 


**[2.9.2] *Viola* sect. *Melanium* subsect. *Cleistogamae* (**
Figure 3
**v and **
Figure 9
**bl)**


*Viola* subsect. *Cleistogamae* Marcussen & Danihelka, subsect. nov.—Type: *Viola rafinesquei* Greene (=*V. bicolor* Pursh non Hoffm.)

*Description*.—Annual to biennial. Lamina of basal leaves crenate. Calycine appendages 0.5–2 mm. Corolla with bright yellow throat. Bottom petal (spur included) 8–10 mm. Spur 1–1.5 mm. Cleistogamous flowers produced. Pollen apertures 4. Ploidy 8*x*. Chromosome number *n* = 17.

*Diagnostic characters*.—Cleistogamous flowers produced.

*Ploidy and accepted chromosome counts*.—8*x*; 2*n* = 34.

*Age*.—Crown node age not applicable (monotypic subsection), stem node age 9.8 (9.1–10.0) Ma [28].

*Included species*.—1. *Viola rafinesquei* Greene

*Distribution*.—Eastern North America.

*Etymology*.—The name *Cleistogamae* refers to the occurrence of seasonal cleistogamy in the type species.

*Distribution*.—Section *Melanium* subsect. *Cleistogamae* comprises *Viola rafinesquei* (=*V. bicolor* Pursh non Hoffm.) only, which differs from all other pansies in two key respects: it has seasonal cleistogamy, i.e., produces chasmogamous flowers in spring (after vernalisation) and cleistogamous ones later in summer, and its native range is in eastern North America while all the other *Melanium* species have their native ranges in the western Palearctic. Cleistogamy in *V. rafinesquei* involves reduction of the four lower anthers, unlike in other *Viola* where the three upper anthers are reduced [256], suggesting cleistogamy in these lineages is not entirely homologous. *Viola rafinesquei* has the chromosome number 2*n* = 34 and 4-colporate pollen, and is an octoploid [28]. Different views on the nomenclature of *V. rafinesquei* have been proposed [257,258] but we hold that of Shinners [257] to be correct. For general taxonomy, see Clausen et al. [256].

The subsections *Cleistogamae* and *Bracteolatae* appear to be monophyletic at the octoploid level and may have split 9–10 Ma ago [28]. The two are, however, genetically distant and cannot be crossed successfully [256].

 


**[2.9.3] *Viola* sect. *Melanium* subsect. *Dispares* (**
Figure 3
**w)**


*Viola* subsect. *Dispares* Marcussen & Danihelka, subsect. nov.—Type: *Viola dyris* Maire

*Description*.—Ephemeral annual or dwarf perennial. Lamina of basal leaves entire or crenate. Calycine appendages 0.3–1 mm. Corolla with bright yellow throat. Bottom petal (spur included) 5–13 mm. Spur 1–3.5 mm. Cleistogamous flowers not produced. Pollen apertures 3 or 4. Ploidy probably 4*x*, 8*x*.

*Diagnostic characters*.—See Table 3 and key.

*Ploidy and accepted chromosome counts*.—Probably 4*x*, 8*x*; 2*n* = 12 (*Viola poetica*), 20, 22 (*V. dyris*), 24 (*V. demetria*).

*Age*.—Crown node c. 2.5 Ma (Figure 6), stem node age probably 11.8–12.8 Ma [28].

*Included species*.—3. *Viola**demetria* Prolongo ex Boiss., *V. dyris* Maire, *V. poetica* Boiss. & Spruner

*Distribution*.—Disjunctly distributed in the Mediterranean area of southern Europe and northern Africa: *Viola dyris* in Morocco (High Atlas), *V. demetria* in southernmost Spain (Andalusia), and *V. poetica* in central Greece (Parnassos).

*Etymology*.—The name *Dispares* refers to the strikingly different general habits and life histories, and few apomorphic characters for this subsection.

*Discussion*.—Section *Melanium* subsect. *Dispares* is the third and last lineage nested within the basal polytomy of sect. *Melanium* (Figure 6). We infer that the subsection comprises three species, *V. demetria*, *V. dyris*, and *V. poetica*. The last species has not been investigated phylogenetically, but monophyly is strongly supported for the other two species using both ITS and chloroplast sequence data [94]. The very short calycine appendages (<1 mm) are an apomorphy for the subsection. Furthermore, all three species have stipules with the main segment resembling the lamina (crenulate in *V. demetria* with 0–3 narrow basal segments [i.e., palmate], entire and undivided in the other two) and small corollas (c. 5 mm in *V. dyris*, up to 13 mm in the other two). In both *V. demetria* and *V. poetica* the spur is intensively violet, and thicker and almost saccate in *V. demetria*. In other respects the three species are morphologically disparate, which probably reflects their adaptations to different extreme environments, i.e., to high-Alpine habitats in the perennials *V. dyris* (scree) and *V. poetica* (rock crevices and screes) as opposed to summer-dry habitats with a short growing season in the ephemeral annual *V. demetria*. The three species are also highly disjunct. *Viola poetica* (2*n* = 12) has 3-colporate pollen and is probably 4*x*, while *V. dyris* (2*n* = 20, 22) and *V. demetria* (2*n* = 24) both have 4-colporate pollen [157] and are probably 8*x*. The chromosome numbers 2*n* = 12 (*V. poetica*) and 2*n* = 24 (*V. demetria*) form a polyploid series; the former is unique and the latter extremely rare among pansies [65,66]. The divergence of *V. demetria* and *V. dyris* may have been relatively recent, only c. 2.7 Ma (Figure 2). 

 


**[2.9.4] *Viola* sect. *Melanium* subsect. *Ebracteatae* (**
Figure 3
**x and **
Figure 9
**bm)**


*Viola* subsect. *Ebracteatae* Kupffer in Kusnezow et al., Fl. Caucas. Crit. 3: 225. 1909 ≡ *Viola* ser. *Ebracteatae* (Kupffer) Vl. V. Nikitin in Bot. Zhurn. (Moscow & Leningrad) 83(3): 135. 1998—Lectotype (Nikitin 1998 [72], page 135): *Viola modesta* Fenzl

*Description*.—Ephemeral annuals. Lamina of basal leaves entire or subcrenate. Calycine appendages 0.5–5.3 mm. Corolla with bright yellow throat. Bottom petal (spur included) 2–11.5 mm. Spur 0.9–3 mm. Cleistogamous flowers not produced. Pollen apertures 3 or heteromorphic 4. Ploidy 4*x*, 8*x*, >8*x*.

*Diagnostic characters*.—Annuals AND lamina of basal leaves entire or subcrenate.

*Ploidy and accepted chromosome counts*.—4*x*, 8*x*, >8*x*; 2*n* = 4, 8, 10, 20, 36.

*Age*.—Crown node c. 8.5 Ma (Figure 6), stem node age 12.5 (11.8–12.8) Ma [28].

*Included species*.—9. *Viola**denizliensis* O. D. Düsen, Göktürk, U. Sarpkaya & B. Gürcan, *V. dirimliensis* Blaxland, *V. ermenekensis* Yıld. & Dinç, *V. mercurii* Orph. ex Halácsy, *V. modesta* Fenzl, *V. occulta* Lehm., *V. parvula* Tineo, *V. pentadactyla* Fenzl, *V. rauliniana* Erben

*Distribution*.—Western Eurasia. Diversity centre in the eastern Mediterranean area.

*Discussion*.—Section *Melanium* subsect. *Ebracteatae* is the second lineage nested within the basal polytomy of sect. *Melanium* (Figure 6). This lineage is characterised phylogenetically by being partly tetraploid [28], karyologically by having very low chromosome numbers (2*n* = 4, 8, 10; polyploid 2*n* = 20, 36), and morphologically by being small-flowered ephemeral annuals (bottom petal 2–11.5 mm) with entire or subcrenate basal leaves. In most species the appendages of the two lower sepals are conspicuously longer than those of the other sepals (not in *V. denizliensis* and *V. dirimliensis*). The tetraploids have small, monomorphic 3-colporate pollen [157].

 


**[2.9.5] *Viola* sect. *Melanium* subsect. *Pseudorupestres* (**
Figure 3
**y)**


*Viola* subsect. *Pseudorupestres* (W. Becker) Marcussen & Danihelka, comb. et stat. nov.—Basionym: *Viola* [sect. *Melanium*; unranked] “γ” *Pseudorupestres* W. Becker in Nat. Pflanzenfam., ed. 2 [Engler & Prantl], 21: 372. 1925 (“*Pseudo-rupestres*”).—Type: *Viola nummulariifolia* Vill.

*Description*.—Dwarf perennial. Stipules dentate, not foliaceous. Lamina of basal leaves entire. Calycine appendages 1.2–1.5 mm. Corolla violet with cream throat. Bottom petal (spur included) 9.5–11.5 mm. Spur 2.3–2.5 mm. Cleistogamous flowers not produced. Pollen apertures 3. Ploidy probably 4*x*. Chromosome number *n* = 7.

*Diagnostic characters*.—Dwarf perennial AND stipules dentate, not foliaceous AND corolla violet with cream throat.

*Ploidy and accepted chromosome counts*.—Probably 4*x*; 2*n* = 14.

*Age*.—Crown node age not applicable (monotypic subsection), stem node age c. 7.2 Ma (Figure 6).

*Included species*.—1. *Viola nummulariifolia* Vill.

*Distribution*.—Southern Europe: Maritime Alps and Corsica.

*Discussion*.—Section *Melanium* subsect. *Pseudorupestres* comprises a single species, *Viola nummulariifolia* Vill. (=*V*. *argenteria* Moraldo & Forneris). Chloroplast and ITS data place it in (or as sister to) the basal polytomy within sect. Melanium [94] (Figure 2), and in a PCO of genomic ISSR data the species ends up ‘close’ to the outgroup [217]. Furthermore, other attributes seem to suggest an isolated phylogenetic position. The low chromosome number of 2*n* = 14 and the 3-colporate pollen [157] indicate low ploidy level, presumably 4*x*, the ancestral condition in sect. *Melanium* [28]. Morphologically, *V. nummulariifolia* has a suite of character states that might be interpreted as plesiomorphic, e.g., perenniality, the flower having a cream throat (not bright yellow as in most other pansies) and simple, entire to dentate stipules, reminiscent of *V. rupestris* (sect. *Viola*), and not large and foliaceous as in many other pansies. *Viola nummulariifolia* has a relictual distribution at high elevations (1800–2900 m) on crystalline rocks in the Maritime Alps and in Corsica [259].

 


**[2.10] *Viola* sect. *Melvio* (**
Figure 3
**z and **
Figure 9
**bn)**


*Viola* sect. *Melvio* Marcussen, sect. nov.—Type: *Viola decumbens* L. f.

*Description*.—Perennial subshrubs. Axes not morphologically differentiated. All stems aerial. Stipules somewhat adnate, green, linear, with 1–2 basal teeth. Lamina entire, linear, subapiculate and somewhat succulent. Bracteoles persistent, 1–3 mm. Corolla violet with a white throat. Spur slender, yellow or orange. Style dorsiventrally flattened and tapering towards the tip, in lateral view filiform and sigmoid. Cleistogamous flowers not produced. Allopolyploid (MELVIO).

*Diagnostic characters*.—Style dorsiventrally flattened and tapering towards the tip, in lateral view filiform and sigmoid. Allopolyploid (MELVIO).

*Ploidy and accepted chromosome counts*.—Allopolyploid, possibly 6*x*; chromosome number unknown.

*Age*.—Crown node age not applicable (monotypic section), stem node age 20.5–22.6 Ma [28].

*Included species*.—1. *Viola decumbens* L. f.

*Distribution*.—South Africa: Cape region (Figure 21).

*Etymology*.—Section *Melvio* is named after the lineage to which it belongs, the diploid MELVIO lineage, for which *Viola decumbens* is the only extant species. The name was originally applied by T.M. [88] to delimit a clade in the *ITS* phylogeny of Ballard et al. [2] which comprised sect. *Melanium* (“MEL”) and sect. *Viola* (“VIO”) only, as a result of *ITS* homoeolog loss and limited sampling.

*Discussion*.—Section *Melvio* comprises a single species, *Viola decumbens* (Figure 4), a shrublet with an isolated distribution in the fynbos of the southern Cape of South Africa [260]. It is the sole member of the otherwise extinct Eurasian MELVIO clade and sister to the taxa involved in the dozen of allopolyploidisations that occurred in Eurasia 15–19 Ma ago. *Viola decumbens* may have been isolated in South Africa since Early Miocene, 20–25 Ma ago [28]. *Viola decumbens* is allopolyploid, possibly paleohexaploid, based on gene copy number for two nuclear genes [28]. The species was previously included in sect. *Xylinosium* [1,28], to which it is superficially similar in shrubby habit. It differs, however, from sect. *Xylinosium* in several key traits. These include the style, which in *V. decumbens* is characteristically dorsiventrally flattened and tapering towards the tip, in lateral view filiform and sigmoid, vs. clavate in sect. *Xylinosium*; the leaves which in *V. decumbens* are entire, linear, subapiculate and somewhat succulent vs. lanceolate and usually crenate in sect. *Xylinosium*; the bracteoles which are 3–5 mm and persistent in *V. decumbens* vs. 1–2 mm or caducous in sect. *Xylinosium*; and the indument of stems and leaves, minutely papillate in *V. decumbens* and distinctly longer in sect. *Xylinosium* (sometimes glabrous or ciliate). The inclusion of *V. decumbens* in sect. *Xylinosium* by Marcussen et al. [28] was a mistake relating to a chloroplast sequence of *V. arborescens* (sect. *Xylinosium*) that had been erroneously assigned to *V. decumbens* (*trnL-trnF*; KJ138159). Indeed, another chloroplast sequence (*rbcL*; AM235165) places this species in agreement with the nuclear homoeologs, of which none are shared between these two taxa.

 


**[2.11] *Viola* sect. *Nematocaulon* (**
Figure 3
**aa and **
Figure 9
**bo)**


*Viola* sect. *Nematocaulon* Marcussen, Nicola, J. M. Watson, A. R. Flores, H. E. Ballard, sect. nov.—Type: *Viola filicaulis* Hook. f.

*Description*.—Perennial herbs. Axes not morphologically differentiated: all stems creeping, branched and remotely noded. Stipules ovate, free, remotely long-fimbriated. Lamina reniform to ovate, few-crenate, long-petiolate. Corolla small, white with violet striations, with a golden yellow throat. Spur short, yellow. Style filiform, terminated in a quadrangular stigmatic opening. Cleistogamous flowers produced; cleistogamy facultative. Chromosome number *n* = 36.

*Diagnostic characters*.—Corolla with a yellow throat AND style filiform.

*Ploidy and accepted chromosome counts*.—Ploidy unknown; 2*n* = 72.

*Age*.—Unknown.

*Distribution*.—New Zealand (Figure 22).

*Included species*.—1. *Viola filicaulis* Hook. f.

*Etymology*.—The name *Nematocaulon* is a Greek translation of the species epithet of the type species, *Viola filicaulis*, which refers to the creeping stems of that species.

*Discussion*.—*Viola filicaulis* is distinct from all other groups and species of violets, as noted already by Hooker [261] in the protologue. Becker [1] noted in the introduction to his treatment of *Viola* that *V. filicaulis* was sufficiently distinct to be placed in a section of its own, although he did not erect one. DNA samples of *V. filicaulis* have not been available for phylogenetic analysis. However, its morphological affinities are clearly with the other southern hemisphere sections of subg. *Viola*. In having a filiform style it is most similar to the species of sect. *Tridens*, sect. *Erpetion*, and sect. *Leptidium*. In the violet-striate pigmentation and shape of the corolla it approaches sect. Tridens (which, however, lacks the yellow throat) and in expressing facultative cleistogamy it is similar to sect. *Chilenium* and sect. *Leptidium*. The high chromosome number of *V. filicaulis* (2*n* = 72 [262]) also agrees with polyploidy in all of these sections. At the same time, style shape, stem not differentiated in a rhizome and lateral stems, and facultative cleistogamy effectively exclude an affinity of *V. filicaulis* to the morphologically superficially similar sections in the northern hemisphere (i.e., *Chamaemelanium*, *Nosphinium*, *Plagiostigma*, and *Viola*). 

*Viola filicaulis* produces cleistogamous flowers in abundance, both seasonally (during summer) and facultatively under unfavourable conditions. These are, however, more morphologically variable and appear less specialised (petals reduced but not absent, number of fertile stamens variable) than in the sympatric *V. cunninghamii* which belongs in sect. *Plagiostigma* subsect. *Bilobatae* and which has a north-temperate origin [26,263].

 

**[2.12] *Viola* sect. *Nosphinium* (**Figure 3**ab**–**ag and **Figure 9**bp–cl)**

*Viola* sect. *Nosphinium* W. Becker in Engler, Nat. Pflanzenfam., ed. 2 [Engler & Prantl], 21: 374. 1925 ≡ *Viola* subg. *Nosphinium* (W. Becker) Espeut in Botanica Pacifica 9(1): 34. 2020.—Lectotype (Espeut 2020 [61], page 34): *Viola chamissoniana* Ging.

*Description*.—Perennial herbs (subshrubs or treelets in most species of subsect. *Nosphinium*). Axes in some species morphologically differentiated into a perennating stem and annual aerial stems (subsect. *Langsdorffianae*, modified in subsect. *Nosphinium*) or stolons (some species of subsect. *Mexicanae*). Perennating stem usually a rhizome, deep or shallow, vertical or horizontal, terminating in an apical rosette. Stipules membranous or partially herbaceous, free (basally or strongly adnate in a few species of subsect. Mexicanae), glandular-fimbriate to glandular-laciniate. Lamina various, long-petiolate. Corolla violet (white in a few species of subsect. *Mexicanae* and subsect. *Nosphinium*), throat white. Calycine appendages short and rounded or elongate, quadrate and often dentate, often elongated somewhat in cleistogamous fruit. Petals large, lateral petals glabrous or sparsely to densely bearded within (spurred petal bearded in some species of subsect. *Borealiamericanae*). Spur thick, as long as tall (sometimes nearly twice as long as tall in *V. langsdorffii*). Style cylindrical with slight subapical swelling (subsect. *Langsdorffianae*), or clavate with apex flanked by a dorsolateral sharp edge or protruding thickened apically oriented or spreading rim, stigma on a pronounced rostellum. Cleistogamous flowers produced, mostly seasonal (cleistogamy absent in subsect. *Pedatae* and in most species of subsect. *Nosphinium*). Allodecaploid with one 2*x* genome from sect. *Chamaemelanium*, one or more 4*x* genomes from sect. *Plagiostigma*, and one 4x genome from sect. *Viola*. *ITS* sequence of MELVIO (sect. *Viola*) type. Inferred secondary base chromosome number [*x’* = 28].

*Diagnostic characters*.—Rosulate habit (rarely stoloniferous or with aerial stems) AND rhizome thick AND stipules free (rarely adnate) AND corolla violet (rarely white) AND petals large AND spur thick and short AND style clavate with dorsolateral edge or thickened rim and pronounced rostellum (rarely with merely a weak dorsolateral swelling) AND allodecaploid with one 2*x* genome from sect. *Chamaemelanium*, one or more 4*x* genomes from sect. *Plagiostigma*, and one 4*x* genome from sect. *Viola*.

*Ploidy and accepted chromosome counts*.—10*x*, 14*x*, 18*x*, ~22*x*; 2*n* = c. 44, 54, c. 76, 80, c. 80, c. 85, c. 86, c. 96, 102, c. 120.

*Age*.—Crown node 8.4 (7.5–8.8) Ma [28].

*Included species*.—62.

*Distribution*.—North America, Hawaiian Islands, Mexico and Central America, a few species in northern South America, one species in northeastern Asia (Figure 23).

*Discussion*.—Sect. *Nosphinium* is a young and nearly exclusively North American radiation. The lineage is allodecaploid and derived from successive hybridisation between North American members of sect. *Chamaemelanium* grex *Nudicaules* (2*x*), sect. *Plagiostigma* subsect. *Stolonosae* (4*x*), and sect. *Viola* (4*x*; Figure 24); it has retained the *ITS* homoeolog of the sect. *Viola* parent (Figure 6) and the chloroplast of the sect. *Plagiostigma* parent [2,45,81]. Section *Nosphinium* comprises five of Becker’s [1] infrageneric taxa, i.e., sect. *Nosphinium* in the strict sense (the Hawaiian violets sensu Becker) and sect. *Nomimium* greges *Borealiamericanae*, *Pedatae*, *Mexicanae*, and *Langsdorffianae* (excluding *V. moupinensis*). These five taxa, in addition to *V. clauseniana*, represent different lineages that we recognise at the subsection level. Section *Nosphinium* consists of two principal lineages, a western, Pacific lineage and an eastern lineage. The western lineage gave rise to subsects. *Nosphinium* and *Langsdorffianae* by independent allopolyploidisations with various sect. *Plagiostigma* taxa, bringing the ploidy of these lineages to 14*x* and 18*x*, respectively (Figure 24). The eastern lineage gave rise to subsects. *Borealiamericanae*, *Clausenianae*, and *Pedatae* without change in ploidy (10*x*) and subsect. *Mexicanae* (14*x*) by yet another allopolyploidisation with another sect. *Plagiostigma* taxon (Figure 24).

Morphologically, the members of sect. *Nosphinium* are a “compromise” among the three parental sections, except for their larger stature which probably reflects higher ploidy. The predominantly violet corolla is shared with sect. *Viola* and the short spur with the other two parents. The style shape is intermediate between sects. *Plagiostigma* and *Viola*. The ability of forming lobed or dissected leaves is shared with sect. *Chamaelenanium*. The lanceolate sepals are more similar to sects. *Plagiostigma* and *Viola* than to sect. *Chamaemelanium* which generally has narrow-lanceolate sepals. 

Apart from the unique decaploidisation that gave rise to sect. *Nosphinium*, the section is difficult to characterise with unique morphological synapomorphies, given that some lineages were produced by secondary allopolyploidisations involving ancestors of diverse subsections in sect. *Plagiostigma*. It is much easier to distinguish the subsections recognised here. Generally, the section is distinguished by a rather thick rhizome, typically large stature, near absence of stolons (present only in some *Mexicanae*), and a short thick spur. Caulescent subsections *Langsdorffianae* and *Nosphinium* (woody except *V. kauaensis*) have broad semi-sheathing stipules and a broadly rounded or convex style apex bent into a slender or thick rostellum but lacking a distinct thickened dorsolateral rim; acaulescent subsections *Clausenianae* and *Pedatae* have partially to almost completely adnate stipules, the former with a style strongly protruded and conspicuously thickened dorsally and a short strongly incurved ventral rostellum, the latter with a style lacking dorsal elongation and merely with a thin dorsolateral margin surrounding a concavity which hides the ventral stigmatic orifice. Subsection *Borealiamericanae* lacks stolons, has lateral petals always densely bearded and bottom petal bearded in some species, and a style with a well developed spreading conspicuously thickened dorsolateral rim, while subsect. *Mexicanae* often produces stolons, has lateral petals beardless or sparsely bearded, bottom petal beardless, and a style with a weak dorsolateral rim oriented forward.


**Key to the subsections of sect. *Nosphinium***


1a.Plant caulescent with aerial stems. Lower stipules broad, triangular or ovate, sheathing the stem. Style apex bent to form a short slender or broad blunt rostellum or tip, the rostellum flanked by a weak dorsolateral swelling. ............................................................................................................................................................................................................................................................................. 21b.Plant acaulescent, stolons present or absent. Stipules narrow, linear-lanceolate, not sheathing the stem. Style apex with a pronounced thickened dorsolateral margin flanking the prominent rostellum (dorsolateral margin thin and rounded with stigmatic orifice hidden in the cavity in subsect. *Pedatae*). ............................................................................................................................................................................................................................................................................... 32a.Flowers white or nearly so on the inside, lacking violet striation. Shrubs or subshrubs, rarely rhizomatous herb with reclining to ascending aerial stems (*V. kauaensis*). Lower stipules triangular, acute to acuminate at apex, margins glandular-lacerate. Flowers often 1–4 together on lateral stems with reduced or absent leaves. Cleistogamous flowers not produced (present in *V. kauaensis*). (Hawaii) ........................................................................................................................................................................................................................................... **subsect. *Nosphinium***2b.Flowers violet, with dark violet striation. Herbs. Lower stipules broadly ovate, obtuse at apex, margins shortly glandular-fimbriate. Flowers solitary with well developed subtending leaves. Cleistogamous flowers produced. (northern Pacific region) ..................................................................................................................................................................................................................................................... **subsect. *Langsdorffianae***3a.At least the outer stipules adnate in basal 1/3 or nearly entirely to petiole. Petals beardless. ...................................................................................................................................................................................................................................................................................... 43b.Stipules free (outer mostly adnate to petiole in *V. humilis*). Lateral petals commonly bearded (bottom petal also bearded in some *Borealiamericanae*). ...................................................................................................................................................................................................................................................................................... 54a.Rhizome stout, vertical, and barrel-like. Stipules adnate for most of their length. Lamina typically deeply pedately divided. Style apex narrowly rounded from above, with dorsolateral margin as a narrowly rounded rim continuing to the ventral surface, the stigma hidden in the narrow triangular cavity created by the rim. Cleistogamous flowers not produced. ....................................................................................................................................................................... **subsect. *Pedatae*** (*V. pedata*)4b.Rhizome relatively slender, oblique or somewhat horizontal, not barrel-like. Stipules adnate for up to 1/3 of their length. Lamina not divided, margins merely crenate. Style apex obtriangular from above, with dorsolateral margin protruding as a thickened broadly truncate or slightly emarginate rim continuing to the rostellum on the ventral surface, the rostellum apically oriented or incurved. Cleistogamous flowers produced. ............................................................................................................................................................................................................................................... **subsect. *Clausenianae*** (*V. clauseniana*)5a.Stolons absent. Stipules free. Laminas in some species lobed or dissected. Calycine appendages elongate and dentate in some species. Corollas violet to dark violet. Lateral petals densely bearded, bottom petal bearded in some species. Style apex with pronounced thickened spreading broadly rounded, sometimes weakly trilobate dorsolateral rim, sides or lateral lobes continuing to the rostellum. (North America, *V. nuevoleonensis* in northern Mexico) ........................................................................................................................................................................................................................................ **subsect. *Borealiamericanae***5b.Stolons usually present (absent in *V. beamanii*, *V. cuicochensis*, *V. hemsleyana*, *V. hookeriana*, and *V. humilis*). Stipules free, or basally or mostly adnate. Laminas undivided. Calycine appendages short and entire. Corollas white or violet. Lateral petals beardless or sparsely bearded (sometimes densely in *V. nubicola* with violet corollas, and in *V. grahamii* and *V. oxydontis* with white corollas), bottom petal beardless. Style apex with weakly thickened apically oriented dorsolateral rim (somewhat prolonged and somewhat thickened dorsally in *V. hookeriana*) continuing partly or completely to rostellum. (Mexico to northern South America) ..................................................................................................................................................................................................................................... **subsect. *Mexicanae***

 


**[2.12.1]. *Viola* sect. *Nosphinium* subsect. *Borealiamericanae* (**
Figure 3
**b and **
Figure 9
**bp–by)**


*Viola* subsect. *Borealiamericanae* (W. Becker) Gil-ad in Bossiera 53: 42. 1997 (“*Boreali-Americanae*”; Shenzhen Code Art. 41.8) ≡ *Viola* [sect. *Nomimium*; unranked] *Borealiamericanae* W. Becker in Repert. Spec. Nov. Regni Veg. 19: 396. 1923 (“*Boreali-Americanae*”) ≡ *Viola* [sect. *Plagiostigma*] subsect. *Borealiamericanae* (W. Becker) Brizicky in J. Arnold Arb. 42: 327. 1961, nom. inval. (Shenzhen Code Art. 41.5; “*Boreali-Americanae*”) ≡ *Viola* [sect. *Plagiostigma*] subsect. *Borealiamericanae* (W. Becker) Val. Tikhom. in Bot. Zhurn. 100: 497. 2015, nom. inval. (Shenzhen Code Art. 41.8; “*Boreali-Americanae*”) ≡ *Viola* sect. *Borealiamericanae* (W. Becker) Espeut in Botanica Pacifica 9(1): 35. 2020 (“*Boreali-Americanae*”)—Type (only species cited): *Viola nuevoleonensis* W. Becker

=*Viola* subg. *Hesperion* Nieuwl. & Kaczm. in Amer. Midl. Naturalist 3: 211. 1914—Type: *Viola palmata* L.

*Description*.—Perennial herbs. Axes not morphologically differentiated; stem a perennial rhizome terminating in an apical rosette. Stipules narrow, free, glandular-lacerate. Laminas in some species lobed or dissected. Calycine appendages various. Petals violet (rarely whitish), lateral and often the spurred petal densely bearded. Style clavate with a pronounced thickened spreading broadly rounded sometimes weakly trilobate dorsolateral rim with sides or lateral lobes continuing to the ventrally oriented rostellum. Cleistogamous flowers produced, seasonal (in temperate species) or simultaneous (in subtropical species). Base chromosome number *x* = 27.

*Diagnostic characters*.—Habit strictly rosulate AND stipules free AND petals violet AND lateral (sometimes spurred) densely bearded AND style with pronounced thickened spreading broadly rounded sometimes weakly trilobate dorsolateral rim and ventrally oriented rostellum AND cleistogamy present AND base chromosome number *x* = 27.

*Ploidy and accepted chromosome counts*.—10*x*; 2*n* = 54.

*Age*.—Crown node at least 2.6 (0.7–5.0) Ma (Figure 6), stem node age 3.2–5.4 Ma [45].

*Included species*.—38. *Viola**affinis* Leconte, *V. baxteri* House, *V. brittoniana* Pollard, *V. calcicola* R. A. McCauley & H. E. Ballard, *V. chalcosperma* Brainerd, *V. communis* Pollard, *V. cucullata* Aiton, *V. edulis* Spach, *V. egglestonii* Brainerd, *V. emarginata* (Nutt.) Leconte, *V. fimbriatula* Sm., *V. floridana* Brainerd, *V. hirsutula* Brainerd, *V. impostor* R. Burwell & H. E. Ballard, ined. [H. E. Ballard 18-002], *V. langloisii* Greene, *V. latiuscula* Greene, *V. lovelliana* Brainerd, *V. missouriensis* Greene, *V. monacanora* J. L. Hastings & H. E. Ballard, ined. [H. E. Ballard 21-015], *V. nephrophylla* Greene, *V. novae-angliae* House, *V. nuevoleonensis* W. Becker, *V. palmata* L., *V. pectinata* E. P. Bicknell, *V. pedatifida* G. Don, *V. pedatiloba* (Brainerd) Burwell & H. E. Ballard, ined., *V. pratincola* Greene, *V. retusa* Greene, *V. rosacea* Brainerd, *V. sagittata* Aiton, *V. septemloba* Leconte, *V. septentrionalis* Greene, *V. sororia* Willd., *V. stoneana* House, *V. subsinuata* (Greene) Greene, *V. tenuisecta* Zumwalde & H. E. Ballard, ined. [Ballard 21-017], *V. viarum* Pollard, *V. villosa* Walter

*Distribution*.—North America.

*Discussion*.—This endemic North American lineage retains the initial allodecaploid genome constitution of the ancestor to sect. *Nosphinium*. A suite of traits delimits the subsection, including a thickish rhizome, strictly rosulate habit, free stipules, undivided or lobed to dissected leaf laminas, large violet to dark violet, rarely whitish corolla, densely bearded lateral petals and often bearded bottom petal, and a style with a spreading conspicuously thickened dorsolateral rim and distinct rostellum. Species express a wide range of diagnostic features in cleistogamous capsule and seed morphology. The centre of diversity is in the Appalachian Mountain range and adjacent uplands. Ezra Brainerd and others conducted many studies of interspecific hybridisation in the subsection, including long-term garden observations and cultivation of F_3_ and F_4_ generations (summarised in Brainerd [264]). Hybridisation is extensive among locally co-occurring species, with hybrids, typically vigorous, failing in chasmogamous reproduction, commonly producing either underdeveloped capsules or capsules with a reduced proportion of viable seeds relative to parental species, and progeny of hybrids express recombinant phenotypic traits of the parental taxa in the plants derived from seeds of the cleistogamous capsules. All species but one occur north of Mexico, whereas *V. nuevoleonensis* is confined to northeastern Mexico.

Despite gradually increasing synonymy by specialists since Brainerd [69], recent studies by HEB and collaborators are revealing many overlooked new species (including some local and regional endemics) and resurrecting previously synonymised species, making it is one of the more diverse subsectional lineages in the genus, and the second largest in the Western Hemisphere (minimum 38 species, possibly as many as 60). *Viola communis* Pollard thrives in lawns and fencerows, and a few species have been inadvertently introduced into Europe [73,265,266,267,268].

 


**[2.12.2] *Viola* sect. *Nosphinium* subsect. *Clausenianae* (**
Figure 3
**a**
**c and Figure 9bz)**


*Viola* subsect. *Clausenianae* H. E. Ballard, subsect. nov.—Type: *Viola clauseniana* M. S. Baker

*Description*.—Perennial herbs. Axes not morphologically differentiated; stem a perennial rhizome terminating in an apical rosette. Stipules narrow, adnate in lowest 1/3. Laminas undivided. Calycine appendages short and truncate to rounded. Petals violet, beardless. Style clavate, the apex triangular from above, the pronounced thickened dorsolateral rim protruding apically as a broadly truncate or weakly emarginate margin continuing down to the rostellum, the rostellum oriented apically or incurved. Cleistogamous flowers produced, seasonal.

*Diagnostic characters*.—Habit strictly rosulate AND stipules basally adnate AND petals violet AND all petals beardless AND style with apically protruding broadly truncate dorsolateral rim and forward-pointing to incurved rostellum AND cleistogamy present.

*Ploidy and accepted chromosome counts*.—10*x*; 2*n* = c. 44 (needs confirmation).

*Age*.—Crown node not applicable (monotypic subsection), stem node age 5.0–11.5 Ma [45].

*Included species*.—1. *Viola clauseniana* M. S. Baker

*Distribution*.—USA (Utah).

*Discussion*.—A monotypic subsection for the anomalous Utah endemic *Viola clauseniana*. Phylogenetic analyses place it firmly among other *Nosphinium* groups but indicate only ambiguous placement otherwise. Genetically it appears to retain the initial allodecaploid constitution of the ancestor of the section [45], which puts into question the count of *n* = c. 22 reported by Clausen [59] from the type locality; we would rather expect *n* = 27 as in the subsections *Borealiamericanae* and *Pedatae*. While most similar morphologically to the *Borealiamericanae*, the absence of petal beards, basally adnate stipules, and style with dorsally protruding and very thickened dorsolateral rim and a forward-pointing to incurved rostellum delimit it uniquely in the section. T.M. has observed unusual morphology in the cleistogamous flowers. The species is known from a single area, Zion National Park, and occurs in isolated “hanging gardens”, seasonally moist to wet cliff ledges.

 


**[2.12.3] *Viola* sect. *Nosphinium* subsect. *Langsdorffianae* (**
Figure 3
**ad and **
Figure 9
**ca–cb)**


*Viola* subsect. *Langsdorffianae* (W. Becker) W. Becker in Acta Horti Gothob. 2: 286. 1926 ≡ *Viola* [sect. *Nomimium*; unranked] *Langsdorffianae* W. Becker in Nat. Pflanzenfam., ed. 2 [Engler & Prantl], 21: 368. 1925 (excl. *V. moupinensis*) ≡ *Viola* sect. *Langsdorffianae* (W. Becker) Espeut in Botanica Pacifica 9(1): 35. 2020—Type (Shenzhen Code Art. 10.8): *Viola langsdorffii* Fisch. ex Ging.

=*Viola* sect. *Arction* Juz. in Schischk. & Bobrov, Fl. URSS 15: 437, 1949, nom. inval. (Shenzhen Code Art. 39.1, descr. rossica); *Viola* sect. *Arction* Juz. ex Zuev in Peschkova, Fl. Sibiri 10: 96. 1996, nom. inval. (Shenzhen Code Art. 40.1, without type)

*Description*.—Perennial, herbs. Axes morphologically differentiated into a perennial rhizome with or without a terminating apical rosette and lateral, annual floriferous stems. Stipules ovate, free, sheathing the stem, shortly glandular-fimbriate. Laminas undivided. Calycine appendages short and truncate to rounded. Petals violet, lateral bearded. Style cylindrical or slightly clavate with a weak dorsolateral swelling and ventrally oriented rostellum. Cleistogamous flowers produced, seasonal. Allo-14-ploid or allo-18-ploid (10x with additional 4*x* genomes from sect. *Plagiostigma*). Secondary base chromosome number *x’* = 40.

*Diagnostic characters*.—Herbaceous AND aerial stems AND stipules ovate, obtuse, shortly glandular-fimbriate and sheathing the stem AND cleistogamy present.

*Ploidy and accepted chromosome counts*.—14*x*, 18*x*, ~22*x*; 2*n* = c. 80 (*V. howellii*), c. 96, 102 (*V. langsdorffii*), c. 120 (*V. “langsdorffii”* sensu Taylor & Mulligan [269]).

*Age*.—Crown node not known, stem node age 1.3–8.8 Ma [45].

*Included species*.—3. *Viola**(Langsdorffianae)* sp., ined. [J. A. Calder & Roy L. Taylor 36425; or: J. A. Calder, Roy L. Taylor & L. C. Sherk 34963], *V. howellii* A. Gray, *V. langsdorffii* Fisch. ex Ging.

*Distribution*.—Western North America and northeastern Asia.

*Discussion*.—Subsect. *Langsdorffianae* is a young high-polyploid lineage that has diversified in response to climate cooling in the Pleistocene [45]. The patterns of variation within subsect. *Langsdorffianae* are poorly understood, but available information from phylogenetics and chromosome counts indicate that the subsection comprises three ploidy levels, each of which we tentatively refer to as species. *Viola howellii* (14*x*) occurs in the North American Pacific Northwest, *V. langsdorffii* (18*x*) occurs on both sides of Beringia from California to Japan, and a yet undescribed taxon (20*x*) occurs at least in the Queen Charlotte Islands of British Columbia [269].

The only phylogenetically investigated species, *V. langsdorffii* (18*x*), arose from successive allopolyploidisations involving the allodecaploid ancestor common to all *Nosphinium* and *V. suecica* (4*x*) of the *Stolonosae* and an unknown member of the *Bilobatae* (4*x*) [45]. The expected chromosome number for *V. langsdorffii* is [2*n* = 104], and the closest actual count is of 2*n* = 102 chromosomes in plants from Hokkaido [270]. As pointed out by Marcussen et al. [45], the numerous counts of 2*n* = (ca.) 96 chromosomes [44,271,272,273,274] “presumably reflect partly the great difficulty in counting many small chromosomes, and partly the wish to align counts with multiples of *x* = 12, the base number attributed to *Langsdorffianae* by early authors” [44,59]. Reports of lower chromosome counts for *V. langsdorffii* (2*n* = c. 60, c. 64, c. 72) must be rejected on the basis of being incompatible with the phylogenetic history of this species and section [45].

The lower chromosome number of *Viola howellii* (2*n* = 14*x* = 80) suggests that it lacks either the *Stolonosae* or the *Bilobatae* genome present in *V. langsdorffii*. Clausen [59] reported “tetraploid” (*n* = 20) and “octoploid” (*n* = 40) counts from Oregon, but whether these refer to the same taxon has not been confirmed; we think the counts of *n* = 20 may rather refer to the sympatric *V*. (subsect. *Rostratae*) *aduncoides*.

The counts of *n* = 60 and 2*n* = c. 120 chromosomes in plants of “*V. langsdorffii*” from the Queen Charlotte Islands, British Columbia [269], most likely represent the 22*x* level and a yet undescribed species. Presumably this taxon has acquired yet another 4*x* genome from sect. *Plagiostigma*.

Several taxa have been distinguished from *Viola langsdorffii* in foliage and style traits, including *V. superba* [275] and the acaulescent *V. simulata* [276] in western North America and *V. kamtschadalorum* in eastern Asia [61,277], but no studies have confirmed the distinctness of these taxa.

 


**[2.12.4] *Viola* sect. *Nosphinium* subsect. *Mexicanae* (**
Figure 3
**ae and **
Figure 9
**cc–ci)**


*Viola* subsect. *Mexicanae* (W. Becker) Marcussen & H. E. Ballard, stat. nov. ≡ Basionym: *Viola* [sect. *Nomimium*; unranked] *Mexicanae* W. Becker in Repert. Spec. Nov. Regni Veg. 19: 396. 1923 ≡ *Viola* sect. *Mexicanae* (W. Becker) Espeut in Botanica Pacifica 9(1): 35. 2020.—Lectotype (designated here): *Viola humilis* Kunth

*Description*.—Perennial herbs. Axes usually morphologically differentiated into a perennial rhizome terminating in an apical rosette and lateral stolons which are often absent. Stipules narrow, free or basally to mostly adnate, glandular-lacerate. Laminas undivided. Calycine appendages mostly short and rounded. Petals violet or whitish, lateral glabrous or sparsely bearded (sometimes densely bearded in *V. grahamii*, *V. nubicola*, and *V. oxyodontis*). Style clavate with a sharp-edged or sometimes weakly thickened apically oriented or slightly incurved dorsolateral rim (somewhat prolonged on the upper side in *V. hookeriana*) continuing partly or fully to the ventrally oriented rostellum. Cleistogamous flowers produced, simultaneous. Allo-14-ploid (10*x* with an additional 4*x* genome from sect. *Plagiostigma*). Secondary base chromosome number *x’* = 40.

*Diagnostic characters*.—Habit rosulate or stoloniferous AND stipules free or adnate AND petals violet or whitish AND lateral petals glabrous or sparsely (rarely densely) bearded AND style with weakly thickened apically oriented (rarely prolonged) dorsolateral rim and ventrally oriented rostellum AND cleistogamy present.

*Ploidy and accepted chromosome counts*.—14*x*; 2*n* = 80 (*Viola nannei*).

*Age*.—Crown node 5.1 (2.6–7.8) Ma (Figure 6), stem node age 3.2–8.8 Ma [45].

*Included species*.—10. *Viola**beamanii* Calderón, *V. cuicochensis* Hieron., *V. grahamii* Benth., *V. guatemalensis* W. Becker, *V. hemsleyana* Calderón, *V. hookeriana* Kunth, *V. humilis* Kunth, *V. nannei* Pol., *V. nubicola* H. E. Ballard, ined. [J. H. Beaman 2976], *V. oxyodontis* H. E. Ballard

*Distribution*.—Mexico to Ecuador.

*Discussion*.—This subsection currently comprises 10 species expressing diverse morphologies but which appear to belong to a single lineage (an unpublished *ITS* phylogeny by HEB including most species was monophyletic with strong support among other lineages of the genus). It arose from a secondary allopolyploidisation event from the ancestor of the *Nosphinium* lineage and an early sister lineage to the North American sublineage of *Stolonosae* [45]. Slightly more than half produce above-ground stolons, two non-stoloniferous species often produce adventitious shoots on roots (*V. beamanii* and *V. hookeriana*), a few species have white flowers (*V. grahamii*, *V. oxyodontis*, and central Mexican populations of *V. hookeriana*), and most species have lateral petals beardless or with sparse beards. One species has strongly adnate outer stipules (*V. humilis*) while two others have basally adnate stipules (*V. grahamii*, *V. oxyodontis*). The style apex has the thin short dorsolateral rim erect (rather than spreading as in the *Borealiamericanae*). Most species are restricted to Mexico, a few extend into Central America, and two are found in northern South America.

 


**[2.12.5] *Viola* sect. *Nosphinium* subsect. *Nosphinium* (**
Figure 3
**af and **
Figure 9
**cj–ck)**


*Viola* subsect. *Nosphinium* (W. Becker) Marcussen & H. E. Ballard, stat. nov. ≡ Basionym: *Viola* sect. *Nosphinium* W. Becker in Nat. Pflanzenfam., ed. 2 [Engler & Prantl], 21: 374. 1925—Lectotype (Espeut 2020 [61], page 34): Viola chamissoniana Ging.

≡*Viola* [unranked] (”Gruppe”) *Sandvicenses* W. Becker in Beih. Bot. Centralbl., Abt. 2, 34: 209. 1917.—Lectotype (designated here): *Viola chamissoniana* Ging.

*Description*.—Branching or non-branching shrubs or treelets, rarely perennial herbs (*Viola kauaensis*). Axes morphologically differentiated into erect stems, rarely rhizomes (*V. kauaensis*), and lateral floriferous stems or branches (very rarely absent). Leaves of floriferous stems in most species reduced to a pair of stipules, giving the floriferous stem the appearance of a leafless, bracteose, 1–4-flowered inflorescence; rarely floriferous stems with normal-sized leaf laminas (*V. chamissoniana* and *V. kauaensis*) or reduced leaf laminas (*V. tracheliifolia*). Stipules triangular, free, sheathing the stem, glandular-lacerate. Laminas crenulate, undivided. Calycine appendages short and truncate to rounded. Petals on the inside violet or whitish, concolourous and lacking violet striation, lateral sometimes bearded; petals often violet on the back side. Style cylindrical or slightly clavate with a weak dorsolateral swelling and thick blunt or short conic rostellum. Cleistogamous flowers produced in *V. kauaensis* only. Allo-14-ploid (10*x* with one additional 4x genome from sect. *Plagiostigma*). Inferred secondary base chromosome number [*x’* = 40].

*Diagnostic characters*.—Woody (rarely herbaceous) AND aerial stems AND stipules triangular, acute or acuminate, glandular-lacerate and sheathing the stem AND cleistogamy absent (rarely present).

*Ploidy and accepted chromosome counts*.—14*x*; 2*n* = c. 76, c. 85, c. 86.

*Age*.—Crown node 5.0 (3.4–6.5) Ma (Figure 25), stem node age 3.9–7.2 Ma [45].

*Included species*.—9. *Viola chamissoniana* Ging., *V. helena* C. N. Forbes & Lydgate, *V. kauaensis* A. Gray, *V. lanaiensis* W. Becker, *V. maviensis* H. Mann, *V. oahuensis* C. N. Forbes, *V. robusta* Hillebr., *V. tracheliifolia* Ging., *V. wailenalenae* (Rock) Skottsb.

*Distribution*.—Hawaiian Islands.

*Discussion*.—This endemic Hawaiian Island subsection arose from a secondary allopolyploidisation including genomes of the allodecaploid ancestor of the *Nosphinium* lineage and a Pacific sublineage of allotetraploid subsect. *Stolonosae* (different from that leading to the *Mexicanae*) [45]. Subsection *Nosphinium* is represented by nine species, most of which are woody and produce lateral 1-4-flowered leafless inflorescences. These species have entirely rayless wood, which agrees with the phylogenetic inference that woodiness is secondary [45,81,278]. *Viola tracheliifolia*, the largest species, is a branched shrub or treelet with lateral inflorescences with reduced (but not absent) leaf laminas. Only *V. kauaensis* has retained the presumably ancestral, herbaceous habit and lateral floriferous stems with solitary flowers in the axil of normal leaves (i.e., peduncles not clustered together on leafless lateral axes) and is the only species producing cleistogamous flowers. The predominantly woody habit and racemose inflorescence, broad semi-sheathing stipules, style with apex bent into a tall short rostellum, and near-absence of cleistogamy define the subsection. An initial phylogenetic study using *ITS* [81] indicated *V. langsdorffii* erroneously as a direct sister taxon to subsect. *Nosphinium*, but the relationships were later shown to be more complex due to separate allopolyploid origins in the *Langsdorffianae* and *Nosphinium* lineages [45]. 

Ballard et al. [81] indicated that the initial diversification occurred on the oldest island of Kauai, with speciation occurring along ecological gradients, and later dispersal and further speciation to younger islands eastward. Havran et al. [85] reanalysed biogeography of subsect. *Nosphinium* with more sophisticated models and arrived at a scenario involving initial dispersal to Maui Nui. A reanalysis of the molecular data set by T.M. arrived at the original finding of colonisation beginning on Kauai (Figure 25), as supported by both ancestral state reconstruction and inferred node ages, and subsequent dispersal and diversification proceeding eastward per the Progression Rule, i.e., hypotheses of phylogeographic congruence among codistributed taxa that track the ages of the islands [279]. This scenario receives further support from the facts that Kauai is home to the only species that has retained the ancestral herbaceous morphology (*V. kauaensis*) and that the average branch length is higher for taxa on Kauai than for taxa on any other Hawaiian island.

 


**[2.12.6] *Viola* sect. *Nosphinium* subsect. *Pedatae* (**
Figure 3
**ag and **
Figure 9
**cl)**


*Viola* subsect. *Pedatae* (Pollard ex W. Becker) Brizicky ex Marcussen & H. E. Ballard, stat. nov. ≡ Basionym: *Viola* [unranked] *Pedatae* Pollard ex W. Becker in Nat. Pflanzenfam., ed. 2 [Engler & Prantl], 21: 369. 1925 ≡ *Viola* “class” *Pedatae* Pollard in Bot. Gaz. 26: 237. 1898, nomen inval. (Shenzen Code Art. 37.6) ≡ *Viola* subsect. *Pedatae* “(Pollard) Brizicky” in J. Arnold Arb. 42: 327. 1961, nom. inval. (Shenzhen Code Art. 41.5) ≡ *Viola* sect. *Pedatae* (Pollard ex W. Becker) Espeut in Botanica Pacifica 9(1): 35. 2020—Type (Shenzhen Code Art. 10.8): *Viola pedata* L.

≡*Oionychion* Nieuwl. & Kaczm. in Amer. Midl. Naturalist 3: 210. 1914.—Type: *Viola pedata* L.

*Description*.—Perennial herbs. Axes not morphologically differentiated; stem a rhizome terminating in an apical rosette. Rhizome thick, vertical and barrel-like. Stipules narrow, long-adnate to petiole. Laminas deeply pedately divided (rare variations with triternate or merely apically incised laminas). Calycine appendages prominent, truncate or dentate. Petals violet, beardless. Style clavate, apex narrowly rounded from above, with dorsolateral margin as a narrowly rounded rim continuing to the ventral surface, the stigma hidden in the narrow triangular cavity created by the rim. Cleistogamous flowers not produced. Secondary base chromosome number *x’* = 27.

*Diagnostic characters*.—Rosulate acaulescent AND stipules long-adnate AND laminas deeply pedately divided (rarely otherwise) AND petals violet AND all petals beardless AND style with narrowly rounded dorsolateral rim and hidden stigma AND cleistogamous flowers not produced. Allodecaploid. *n* = 27.

*Ploidy and accepted chromosome counts*.—10*x*; 2*n* = 54.

*Age*.—Crown node not applicable (monotypic subsection), stem node age 5.0–6.0 Ma [45].

*Included species*.—1. *Viola pedata* L.

*Distribution*.—Eastern North America.

*Discussion*.—A monotypic subsection for *Viola pedata*, a widely distributed eastern North American species of dry oak woodlands, oak savannas and dry prairies. The subsection (and species) are unusual in having a short vertical barrel-like rhizome pulled below the soil surface by contractile roots, long-adnate stipules, a clavate or narrowly ellipsoid style lacking a noticeable to prominent thickened dorsolateral rim (this simply a thin non-spreading margin), and absence of cleistogamy. The type variety produces deeply pedately divided laminas with linear segments; populations with narrowly flabellate laminas in the Sandhills region of the southeastern U.S. and populations with triternately divided laminas in the east-central Piedmont, are treated as varieties. This species is unusual also in maintaining a presumably balanced polymorphism in corolla colour pattern, with individuals with dark violet-black upper petals increasingly common further south in the range, and individuals with all petals light violet increasingly common to the north. Finally, *V. pedata* is the only member of the genus reported to be self-incompatible [280]. Phylogenetic studies involving all North American lineages have shown that, like *V. clauseniana*, *V. pedata* does belong to the *Nosphinium* lineage but has ambiguous relationships among the other species. It has retained the initial allodecaploid constitution of the ancestor of the *Nosphinium* lineage [45] but has obviously diverged considerably from the other subsections.

 

**[2.13] *Viola* sect. *Plagiostigma* (**Figure 3**ah**–**an and **Figure 9**cm–dx)**

*Viola* sect. *Plagiostigma* Godr., Fl. Lorraine, ed. 2, 1: 90. 1857 ≡ *Viola* [unranked] (“Gruppe”) *Plagiostigma* (Godr.) Kupffer in Oesterr. Bot. Z. 53: 329. 1903 ≡ *Viola* [sect. *Nomimium*] subsect. *Plagiostigma* (Godr.) J. C. Clausen in Ann. Bot. (Oxford) 43: 751. 1929; (Godr.) P. Y. Fu, Fl. Pl. Herb. Chin. Bor.-Or. 6: 89. 1977 (isonym)—Type: *Viola palustris* L.

*Description*.—Perennial herbs, very rarely annuals. Axes morphologically differentiated in a rhizome and lateral stems; sometimes only a rhizome present. Rhizome densely or sometimes remotely noded, with an apical leaf rosette. Lateral stems annual aerial stems or stolons. Stipules free or adnate to petiole. Lamina extremely variable, entire or deeply divided, petiolate. Corolla violet, pink or white, with violet striations, and a white or yellow-green throat. Spur short and saccate to very long and slender. Style clavate, at apex flattened above, laterally and distally margined, or bilobate, beardless. Cleistogamous flowers produced; cleistogamy seasonal. Allotetraploid (CHAM + MELVIO). ITS sequence of CHAM type. Secondary base chromosome number x’ = 12.

*Diagnostic characters*.—Corolla violet, pink or white with a white or yellow-green throat AND style clavate, at tip flattened above, laterally and distally margined, or bilobate AND base chromosome number *x’* = 12.

*Ploidy and accepted chromosome counts*.—4*x*, 8*x*, 12*x*; 2*n* = 20, 22, 24, 26, 44, 48, 72, 74.

*Age*.—Crown node age 16.6 (15.4–17.0) Ma [28].

*Included species*.—142.

*Distribution*.—Throughout the northern temperate region, with a few species south of the equator in Australasia and South America; centre of diversity in eastern Asia (Figure 26).

*Discussion*.—Sect. *Plagiostigma* is phylogenetically an allotetraploid CHAM + MELVIO lineage; unlike all other sections except sect. *Danxiaviola* it has retained the CHAM homoeolog for *ITS* (Figure 6). It is karyologically characterised by the secondary base chromosome number *x* = 12, and morphologically by the clavate, margined or bilobate, beardless style, and the occurrence of seasonal cleistogamy. Here, we recognise a narrowly circumscribed sect. *Plagiostigma* that comprises the six Beckerian greges [1] having a ‘plagiostigmate’ style shape and a secondary base chromosome number *x’* = 12, i.e., sect. *Nomimium* greges *Adnatae* p.p., *Bilobatae*, *Diffusae*, *Serpentes* p.p., *Stolonosae*, and *Vaginatae*. In this respect our classification approaches Clausen’s [29,59] but we further exclude the North American allodecaploid lineage, herein transferred to sect. *Nosphinium* [28,45,61].

With its 139 known species and a crown node of 16.6 (15.4–17.0) Ma, sect. *Plagiostigma* is both the oldest and the most species-rich of all *Viola* sections. It could be justified to treat subsect. *Diffusae* and subsect. *Patellares* as separate sections. We keep them within sect. *Plagiostigma* because of at least two synapomorphies, the style shape and the base chromosome number *x* = 12. We recognise seven subsections within sect. *Plagiostigma* (Figure 27), each monophyletic and morphologically characterised, i.e., subsect. *Australasiaticae*, subsect. *Bilobatae*, subsect. *Bulbosae*, subsect. *Diffusae*, subsect. *Patellares*, and subsect. *Stolonosae*. *Diffusae* and *Patellares* are sisters (or sister) to the lineage comprising *Bilobatae*, *Bulbosae*, and *Stolonosae*. The phylogenetic placement of subsect. *Australasiaticae* within the section is unknown, as this taxon is represented by *ITS* sequences only and this marker (Figure 6) poorly reflects the genome phylogeny (Figure 27).

While 2*n* = 24 is retained in most of the subsections, 2*n* = 22 is apomorphic in subsect. *Formosanae* and, possibly, 2*n* = 46 in subsect. *Austalasiaticae*.

There is little agreement between Becker’s [1] greges and the subsections proposed herein. This is discussed briefly under each subsection. 

 


**Key to the subsections of *Viola* sect. *Plagiostigma***


1a.Rhizome annual, vertical, growing from deep-buried bulbils, aerial part more remotely noded. Lateral stolons present, underground, leafless. Outer stipules adnate, inner stipules free. ................................................................................................................................................................................................................................. **subsect. *Bulbosae***1b.Rhizome perennial or very rarely plant annual (in *V. diffusa*), horizontal or vertical, bulbils absent. Lateral stolons absent or when present not underground and leafless. Stipules free or adnate. ..................................................................................................................................................................................................................................................................... 2.2a.Rhizomatous herbs lacking lateral stolons and aerial stems but sometimes with adventitious buds on roots. Stipules adnate to petiole in the lower 1/3–1/4. Leaf margin crenulate to deeply divided. Spur 1–10 mm, usually slender and longer than the calycine appendages. Corolla white to deep (bluish or reddish) violet. ...................................................................................................................................................................................................................................................... **subsect. *Patellares***2b.Rhizomatous herbs, usually with lateral stolons and/or aerial stems. Stipules free or up to 1/3 adnate to petiole. Leaf margin crenulate or crenate, never deeply divided. Spur usually short and saccate, 1–4 mm, rarely 5–7 mm (in *V. formosana*). Corolla usually white or pale violet ......................................................................................................................................................................................................................................................... 3.3a.Bottom petal longer than the other petals, deeply emarginate or cleft. Spur longer than tall, 1.5–7 mm. .................................................................................................................................................................................................................................................... **subsect. *Formosanae***3b.Bottom petal shorter or subequal to the other petals, acute, obtuse or rarely emarginate. Spur as long as tall, 1–4 mm. ................................................................................................................................................................................................................................................................................ 4. 4a.Lateral stems aerial, decumbent or erect, rarely short or absent (in *V. cunninghamii*). Stipules foliaceous, free or adnate at base only, dentate or entire. Lamina semilunate to triangular or hastate. Style margined and bilobate at apex. ................................................................................................................................................................................................................................................. **subsect. *Bilobatae***4b.Lateral stems stolons, rarely aerial stems or absent. Stipules pale or purple-brown, rarely greenish, free or partially adnate, fimbriate or entire. Lamina reniform to narrowly lanceolate. Style apex bilobate or flattened, distinctly margined. ...................................................................................................................................................................................................................................................................................... 5.5a.Sepals usually broadly lanceolate to broadly ovate, rarely lanceolate (but then with long denticulate sepal appendages: *V. thomsonii*). Sepal appendages up to 2 mm, sometimes denticulate. Bottom petal 7–25 mm, usually not conspicuously shorter than the other petals, sometimes longer, truncate or emarginate, rarely acute, violet-striate. Style apex flattened above and margined, rarely bilobate. Stipules lanceolate to ovate, entire or remotely denticulate to fimbriate-dentate, free or adnate at base only. Corolla commonly white with a yellow-green throat, rarely violet or pink. ............................................................................................................................................................................................................................................... **subsect. *Stolonosae***5b.Sepals linear-lanceolate to lanceolate, rarely ovate-lanceolate (in *V. (Diffusae) guangzhouensis*), with short appendages, 0.4–1 mm, rounded or slightly denticulate (absent in *V. kwangtungensis*). Bottom petal 5–12 mm, shorter and narrower than the others, usually acute, with conspicuous violet striation or reticulation. Style apex bilobate. Stipules linear to broadly lanceolate, densely or remotely fimbriate, free or adnate in the lower 1/3. Corolla pale pink or pale violet, rarely white. ....................................................................................................................................................................................................................................................... 66a.Lateral petals not bearded. Peduncles glabrous; plant usually glabrous or nearly so. Rhizome long and remotely noded or short and densely noded. Stolons present or rarely absent, with (many) scattered leaves. Stipules free or adnate at base only, often brownish, long-fimbriate to laciniate. Corolla usually pale violet to whitish, without a greenish throat. Lamina margin crenate, occasionally with conspicuous mucronules. Perennials. ....................................................................................... ......................................................................................................................................................... **subsect. *Australasiaticae***6b.Lateral petals usually bearded. Peduncles with patent hairs, rarely glabrous (in *V. nanlingensis*); plant usually hairy. Rhizome short, densely noded. Stolons with 1–2 (smaller) leaves and a leaf rosette at apex. Stipules adnate in the lower 1/3 (stipules on aerial stems free in *V. guangzhouensis*), remotely or rarely densely fimbriate. Corolla usually pale pink to pale violet, with a greenish throat. Lamina margin crenulate, never with mucronules. Perennials or rarely annuals (*V. diffusa*). ................................................................................................................................................................................................................................................... **subsect. *Diffusae***

 


**[2.13.1] *Viola* sect. *Plagiostigma* subsect. *Australasiaticae* (**
Figure 3
**ah and **
Figure 9
**cm–cp)**


*Viola* subsect. *Australasiaticae* (M. Okamoto) Marcussen, comb. et stat. nov.—Basionym: *Viola* ser. *Australasiaticae* M. Okamoto in Taxon 42(4): 784. 1993.—Type: *Viola sumatrana* Miq. 

*Description*.—Rhizome perennial; bulbils absent. Lateral stems usually present: aboveground stolons, most leaves scattered. Stipules free or adnate at base only, brown, linear-lanceolate to broadly lanceolate, long-fimbriate. Lamina triangular-ovate to reniform, base cuneate to deeply cordate, apex obtuse to acuminate, margin crenate or mucronulate. Corolla white or pale violet. Sepals linear-lanceolate to lanceolate; appendages short or absent (0–1.4 mm), rounded or slightly denticulate. Lateral petals not bearded; bottom petal shorter than the other petals (5–12 mm), acute to obtuse; spur short (1–2.5 mm) and saccate. Style at apex margined and bilobate.

*Diagnostic characters*.—Plants usually stoloniferous AND stolons with most leaves scattered AND sepals linear-lanceolate to lanceolate AND lateral petals glabrous AND bottom petal shorter than the others AND style margined and bilobate at apex.

*Ploidy and accepted chromosome counts*.—4*x*? 8*x*; 2*n* = 46.

*Age*.—Crown node age c. 12.0 Ma; stem node c. 16.3 Ma (Figure 6).

*Included species*.—10. *Viola**annamensis* Baker f., *V. austrosinensis* Y. S. Chen & Q. E. Yang, *V. balansae* Gagnep., *V. duclouxii* W. Becker, *V. hossei* W. Becker, *V. kwangtungensis* Melch., *V. mucronulifera* Hand.-Mazz., *V. shiweii* Xiao Chen Li & Z. W. Wang, *V. sikkimensis* W. Becker, *V. sumatrana* Miq.

*Distribution*.—Southeastern Asia and Malesia.

*Discussion*.—Becker [1] erected (sect. *Nomimium*) grex *Serpentes* as a catch-all taxon for stoloniferous species from subtropical Asia. This group was highly heterogeneous and the constituent species have later been redistributed among sect. *Viola* subsects. *Rostratae* and *Viola*, and sect. *Plagiostigma* subsects. *Australasiaticae*, *Diffusae*, *Patellares*, and *Stolonosae* [86,126,229,231]. Wang [76] expanded Becker’s greges, as sect. *Serpentes*, to include numerous *Stolonosae* species. Okamoto et al. [229] showed that the type species of grex *Serpentes* (*V. serpens* Blume, a synonym of *V. pilosa*) belongs in subsect. *Viola* and they therefore designated ser. *Australasiaticae* (type: *V. sumatrana*) as a replacement name for the remaining species not belonging in subsect. *Viola*. However, also Okamoto’s [229] *Australasiaticae* proved heterogeneous and including taxa from different sections and subsections. The type, *V. sumatrana*, was however not analysed phylogenetically before the recent study by C. Li et al. [231] which clearly identified the *Australasiaticae* in the strict sense as a separate lineage within sect. *Plagiostigma* (Figure 6). We here define subsect. *Australasiaticae* narrowly as comprising all known *Plagiostigma* species having stolons with scattered leaves, linear-lanceolate or lanceolate sepals, unbearded lateral petals, and a bilobate style.

The only chromosome counts for subsect. *Australasiaticae* are of 2*n* = 46 in *V. sumatrana* and *V. annamensis* (as *V. rheophila* Okamoto) and were reported without metadata by H. Okada in Okamoto et al. [229] and are therefore in need of confirmation. If proved correct, they presumably reflect the 8*x* level and present a unique number in the genus and a possible apomorphy for subsect. *Australasiaticae*. It is not known whether this chromosome number and ploidy are shared by all the members of the subsection.

Spinulose or mucronulate leaf margins (as an adaptation to guttation?) occur only in this subsection within *Viola* but have apparently originated twice. In *Viola balansae* and *V. kwangtungensis* the mucronules are extensions of the apex of each leaf tooth and are in the plane of the leaf. In *V. mucronulifera* the mucronules are adaxial extensions of the invagination between leaf teeth and are perpendicular to the plane of the leaf [231].

 


**[2.13.2] *Viola* sect. *Plagiostigma* subsect. *Bilobatae* (**
Figure 3
**ai and **
Figure 9
**cq–ct)**


*Viola* subsect. *Bilobatae* (W. Becker) W. Becker in Acta Horti Gothob. 2: 288. 1926 ≡ *Viola* [sect. *Nomimium*; unranked] (“Gruppe”) *Bilobatae* W. Becker in Beih. Bot. Centralbl., Abt. 2, 34: 226. 1917 ≡ *Viola* ser. *Bilobatae* (W. Becker) Steenis in Bull. Jard. Bot. Buitenzorg, ser. 3, 13 (1933–1936): 260. 1934 ≡ *Viola* sect. *Bilobatae* (W. Becker) Juz. in Schischk. & Bobrov, Fl. URSS 15: 439. 1949—Lectotype (Espeut 2020 [61], page 33): *Viola arcuata* Blume

*Description*.—Rhizome perennial; bulbils absent. Lateral stems present or rarely absent: aerial stems, decumbent or erect, leaves scattered. Stipules free or adnate at base only, green and foliaceous, up to 40 mm, linear-lanceolate to ovate, obtuse to acuminate, entire, remotely denticulate or lobed. Lamina ovate-triangular to narrowly triangular or nearly hastate, base truncate to broadly cordate, often with a lunate sinus, apex more or less acute, margin crenulate. Corolla white. Sepals linear to ovate-lanceolate; appendages short (c. 0.5 mm), rounded or slightly denticulate. Lateral petals bearded or not; bottom petal shorter than the other petals (6–8 mm), apex rounded; spur short (1–2 mm) and saccate. Style at apex margined and bilobate.

*Diagnostic characters*.—Stipules foliaceous AND style margined and distinctly bilobate at apex.

*Ploidy and accepted chromosome counts*.—4*x*, 8*x*; 2*n* = 24, 44, 48.

*Age*.—Crown node age c. 4.7 Ma (Figure 6), stem node age 13.5 (12.2–14.0) Ma [28].

*Included species*.—8. *Viola**amurica* W. Becker, *V. caleyana* G. Don, *V. cunninghamii* Hook. f., *V. hamiltoniana* D. Don, *V. lyallii* Hook. f., *V. merrilliana* W. Becker, *V. raddeana* Regel, *V. triangulifolia* W. Becker

*Distribution*.—Eastern Asia, Malesia, Australia, New Zealand.

*Discussion*.—The overall morphology of sect. *Plagiostigma* subsect. *Bilobatae* is superficially similar to that of the unrelated sect. *Viola* subsect. *Rostratae*, and conspicuously so in species such as *V. raddeana* (*Bilobatae*) and *V. stagnina* (*Rostratae*), which both are adapted to floodplain habitats. Reported chromosome counts of 2*n* = 20 in subsect. *Bilobatae* (cf. [61]) are likely errors.

 


**[2.13.3] *Viola* sect. *Plagiostigma* subsect. *Bulbosae* (**
Figure 3
**aj and **
Figure 9
**cu–cv)**


*Viola* subsect. *Bulbosae* Marcussen, subsect. nov.—Type: *Viola bulbosa* Maxim.

*Description*.—Rhizome annual, growing from underground bulbil. Lateral stems present: underground stolons, usually leafless but with scattered nodes. Stipules outer stipules adnate, inner stipules free, pale, linear-lanceolate, remotely fimbriate. Lamina oblong-ovate, suborbicular or reniform, base cuneate or narrowly cordate, apex rounded or acute, margin crenulate. Corolla white. Sepals lanceolate to broadly lanceolate; appendages short (c. 0.8 mm), rounded. Lateral petals bearded or not; bottom petal shorter than the other petals (7–8 mm), apex rounded; spur short (1.2–1.7 mm) and saccate. Style at apex margined and bilobate. 

*Diagnostic characters*.—Rhizome vertical, growing from deep-buried bulbils.

*Ploidy and accepted chromosome counts*.—4*x*; 2*n* = 24.

*Age*.—Crown node age unknown, stem node age 13.5 (12.2–14.0) Ma [28].

*Included species*.—2. *Viola bulbosa* Maxim., *V. tuberifera* Franch.

*Distribution*.—Eastern Himalaya and central China.

*Discussion*.—Section *Plagiostigma* subsect. *Bulbosae* comprises two species, *Viola bulbosa* and *V. tuberifera* [77,281]. The species are characterised by having small underground bulbs, a unique feature in *Viola*. The bulb is composed of 4–8 fleshy petiole bases along a condensed axial portion which apically elongates into the annual aerial stem and laterally produces underground, leafless stolons with cleistogamous flowers. The species were included in subsect. *Patellares* by both Becker [1] and Wang [76], as grex *Adnatae* and sect. *Adnatae*, respectively.

 


**[2.13.4] *Viola* sect. *Plagiostigma* subsect. *Diffusae* (**
Figure 3
**ak and **
Figure 9
**cw–cz)**


*Viola* subsect. *Diffusae* (W. Becker) Chang in Bull. Fan Mem. Inst. Biol., ser. n., 1(3): 249, 1949 [non vidimus] ≡ *Viola* [unranked] (“Gruppe”) *Diffusae* W. Becker in Beih. Bot. Centralbl., Abt. 2, 40: 113. 1924 ≡ *Viola* (sect. *Nomimium*) ser. *Diffusae* (W. Becker) Steenis in Bull. Jard. Bot. Buitenzorg, ser. 3, 13 (1933–1936): 260. 1934 ≡ *Viola* sect. *Diffusae* (W. Becker) Ching J.Wang, Fl. Reipubl. Popularis Sin. 51: 100. 1991.—Type (Shenzhen Code Art. 10.8): *Viola diffusa* Ging.

*Description*.—Rhizome perennial or rarely plant annual; bulbils absent. Lateral stems present: aboveground stolons, most leaves in apical rosette; rarely also aerial stems with scattered leaves. Stipules usually adnate in the lower ⅓, pale, greenish, or brown, subulate to lanceolate, acuminate, remotely long-fimbriate. Lamina ovate, ovate-oblong or elliptic, base cuneate to shallow-cordate, often decurrent, apex usually obtuse, margin crenate. Corolla usually pale pink or pale violet, with a greenish throat. Sepals linear to ovate-lanceolate; appendages short (0.3–0.8 mm), rounded or slightly denticulate. Lateral petals bearded or not; bottom petal shorter than the other petals (5–12 mm), apex acute; spur short (1–2.5 mm) and saccate. Style at apex margined and bilobate.

*Diagnostic characters*.—Stolons long with 1–2 leaves and a leaf rosette at apex AND stipules 1/3 adnate to petiole AND corolla mostly pale pink to pale violet, with a greenish throat AND style margined and bilobate at apex. 

*Ploidy and accepted chromosome counts*.—4*x*, 8*x*, 12*x*; 2*n* = 24, 26, 48, 74.

*Age*.—Crown node age 8.5 Ma (Figure 6), stem node age 16.6 (15.4–17.0) Ma [28].

*Included species*.—17. *Viola**(Diffusae)* sp. 1, ined., *V. (Diffusae)* sp. 2, ined., *V. (Diffusae)* sp. 3, ined., *V. amamiana* Hatus., *V. apoensis* Elmer, *V. changii* J. S. Zhou & F. W. Xing, *V. diffusa* Ging., *V. guangzhouensis* A. Q. Dong, J. S. Zhou & F. W. Xing, *V. huizhouensis* Y. S. Huang & Q. Fan, *V. jinggangshanensis* Z. L. Ning & J. P. Liao, *V. lucens* W. Becker, *V. nagasawae* Makino & Hayata, *V. nanlingensis* J. S. Zhou & F. W. Xing, *V. pricei* W. Becker, *V. tenuis* Benth., *V. wilsonii* W. Becker, *V. yunnanensis* W. Becker & H. Boissieu

*Distribution*.—Southeastern Asia.

*Discussion*.—Section *Plagiostigma* subsect. *Diffusae* comprises a handful of southeast Asian species, characterisable by stolons with few internodes and a terminal leaf rosette, stipules adnate to the petiole in the lower third, and more or less lanceolate laminas with a narrow and shallow sinus. Some species have aerial stems. Most species are distinctly stiffly hairy and have pale violet or pink petals, often yellowish green at the base, with a short and narrow, pointed bottom petal and a very short spur. This subsection, although easily recognisable in most cases, is poorly understood owing to taxonomic confusion with the other stolonose subsections *Australasiaticae* and *Stolonosae*.

More than half of the species placed in subsect. *Diffusae* are narrow endemics native to southern China that have been discovered and described within the last 15 years [127,128,282,283,284]. These species grow on rock surfaces, often in inaccessible places, and numerous species are still undescribed in addition to the 17 cited here (Y.-S. Huang, pers. comm. [285]).

 


**[2.13.5] *Viola* sect. *Plagiostigma* subsect. *Formosanae* (**
Figure 3
**al and **
Figure 9
**da–db)**


*Viola* subsect. *Formosanae* (J.-C. Wang & T.-C. Huang) Marcussen, comb. et stat. nov.—Basionym: *Viola* grex *Formosanae* J.-C. Wang & T.-C. Huang in Taiwania 35(1): 14. 1990.—Type (only species listed): *Viola formosana* Hayata

*Description*.—Rhizome perennial; bulbils absent. Lateral stems present: aboveground stolons, most leaves in apical rosette. Stipules free or adnate at base, purplish-brown, lanceolate or narrowly ovate, long fimbriate-laciniate. Lamina broadly triangular-ovate or oblong-orbicular, base deeply cordate to rounded, apex acute to rounded or obtuse, margin crenate. Corolla white or pale violet. Sepals narrowly lanceolate to oblong; appendages short (0.5–1 mm), rounded. Lateral petals not bearded; bottom petal longer than the other petals (8–15 mm), apex deeply emarginate or shallowly cleft; spur long and slender (1.5–7 mm). Style at apex margined and flattened, not bilobate. Secondary base chromosome number *x’* = 11.

*Diagnostic characters*.—Bottom petal longer than the other petals AND stolons with most leaves in apical rosette AND chromosome number 2*n* = 22.

*Ploidy and accepted chromosome counts*.—4*x*; 2*n* = 22.

*Age*.—unknown.

*Included species*.—2. *Viola formosana* Hayata, *V. stoloniflora* Yokota & Higa

*Distribution*.—Southeastern Asia: the islands of Taiwan (*V. formosana*) and Okinawa (*V. stoloniflora*).

*Discussion*.—Becker was familiar with *Viola formosana* ([286], page 167), the only of the two species known at the time, but he did not mention it or place it systematically in his revision of the genus [1]. The second species, *V. stoloniflora*, has been placed in subsect. *Australasiaticae* [229] or in its predecessor, subsect. *Serpentes*, “on account of its procumbent stolons, almost free fimbriate stipules, and deplanate obtriangular-dilatate styles” [97]. In their revision of the violets of Taiwan, Wang & Huang (1990 [75]) recognised the distinctness of *V. formosana* and placed it in a provisional group of its own, *Formosanae*, one of eight unranked greges; their delimitation of greges is reconcilable with our classification.

The phylogenetic placement of subsect. *Formosanae* is unresolved, but published chloroplast DNA sequences of *Viola formosana* place it among the other stoloniferous subsections [287].

The two species *Viola formosana* and *V. stoloniflora* have never been grouped together, despite their close geographical proximity and several synapomorphies that set them apart from all other subsections of sect. *Plagiostigma*, including the long and emarginate bottom petal, the shape of the stolons (reminescent of subsect. *Diffusae*), and the rare chromosome number 2*n* = 22 [75,97]. 

*Viola stoloniflora* is extinct in the wild; its only known locality in Okinawa Island was destroyed by the construction of the Benoki Dam, which was completed in 1987 [97].

 


**[2.13.6] *Viola* sect. *Plagiostigma* subsect. *Patellares* (**
Figure 3
**am and **
Figure 9
**dc–dh)**


*Viola* subsect. *Patellares* (Boiss.) Rouy & Foucaud, Fl. France [Rouy & Foucaud] 3: 35. 1896 ≡ *Viola* [sect. *Nomimium*; unranked] §.3. *Patellares* Boiss., Fl. Orient. 1: 451. 1867, p.p. (excl. *Viola uliginosa*).—Lectotype (designated here): *Viola kamtschatica* Ging. (=*V. selkirkii* Pursh ex Goldie)

=*Viola* [sect. *Nomimium*; unranked] b. *Patellariae* Nyman, Consp. Fl. Eur. 1: 79. 1878, p.p.—Lectotype (designated here): *Viola umbrosa* Fr. (=*Viola selkirkii* Pursh ex Goldie)

=*Viola* subg. *Violidium* K. Koch in Linnaea 15: 251. 1841. ≡ *Viola* sect. *Violidium* (K. Koch) Juz. in Schischk. & Bobrov, Flora URSS 15: 408. 1949 ≡ *Viola* subsect. *Violidium* (K. Koch) P. Y. Fu in Fl. Pl. Herb. Chin. Bor.-Or. 6: 93. 1977.—Type: *Viola somchetica* K. Koch

=*Viola* [unranked] (“Gruppe”) *Estolonosae* Kupffer in Oesterr. Bot. Z. 53: 329. 1903 ≡ *Viola* subsect. *Estolonosae* (Kupffer) Kupffer in Kusnezow et al., Fl. Caucas. Crit. 3(9): 217. 1909 ≡ *Viola* sect. *Estolonosae* (Kupffer) Vl. V. Nikitin in Bot. Zhurn. (Moscow & Leningrad) 83(3): 132. 1998.—Lectotype (Nikitin 1998 [72], page 133): *Viola purpurea* Stev. (=*V. somchetica* K. Koch)

=*Viola* [sect. *Nomimium*; unranked] *Adnatae* W. Becker in Nat. Pflanzenfam., ed. 2 [Engler & Prantl], 21: 368. 1925 ≡ *Viola* subsect. *Adnatae* (W. Becker) W. Becker in Acta Horti Gothob. 2: 285. 1926 ≡ *Viola* ser. *Adnatae* (W. Becker) Steenis in Bull. Jard. Bot. Buitenzorg, ser. 3, 13 (1933–1936): 258. 1934 ≡ *Viola* sect. *Adnatae* (W. Becker) Ching J. Wang, Fl. Reipubl. Popularis Sin. 51: 41. 1991; (W. Becker) Vl. V. Nikitin in Bot. Zhurn. (Moscow & Leningrad) 83(3): 132. 1998 (isonym); (W. Becker) Vl. V. Nikitin in Novosti Sist. Vyssh. Rast. 31: 222. 1998 (isonym).—Lectotype (Nikitin 1998 [72], page 132): *Viola selkirkii* Pursh ex Goldie

=*Viola* [unranked] “Gruppe” *Pinnatae* W. Becker, Beih. Bot. Centralbl., Abt. 2. 40(2): 119. 1924 ≡ *Viola* sect. *Pinnatae* (W. Becker) Ching J. Wang, Fl. Reipubl. Popularis Sin. 51: 76. 1991 ≡ *Viola* subsect. *Pinnatae* (W. Becker) Vl. V. Nikitin, Novosti Sist. Vyssh. Rast. 34: 125. 2002.—Type (Shenzhen Code Art. 10.8): *Viola pinnata* L.

=*Viola* sect. *Brachycerae* Espeut in Botanica Pacifica 9(1): 32. 2020.—Type: *Viola brachyceras* Turcz.

*Description*.—Rhizome perennial; bulbils absent. Lateral stems absent. Stipules adnate in the lower 1/3 to 3/4, pale, greenish, or purple-brown, linear to ovate-lanceolate, acute or acuminate, entire or remotely denticulate-fimbriate. Lamina lanceolate to orbicular or triangular, sometimes 3–5-sect, base cuneate to deeply cordate, sometimes decurrent, apex obtuse to acuminate, margin subentire, crenulate, dentate, or deeply incised. Corolla white to deep violet. Sepals lanceolate to ovate; appendages short to very long (0.4–6 mm), rounded to 2–3-dentate. Lateral petals usually bearded; bottom petal usually longer than the other petals ((5–)10–23(–25) mm), apex rounded to emarginate; spur long (3–10 mm) and slender, rarely short (1–2 mm) and saccate. Style at apex margined and flattened, not bilobate.

*Diagnostic characters*.—All stems rhizomatous AND stipules 1/3 adnate to petiole AND spur slender, up to 10 mm AND cleistogamous flowers produced.

*Ploidy and accepted chromosome counts*.—4*x*, 8*x*, 12*x*; 2*n* = 22, 24, 48, 72.

*Age*.—Crown node age c. 8.3 Ma (Figure 6), stem node age 16.6 (15.4–17.0) Ma [28].

*Included species*.—62. *Viola**alaica* Vved., *V. albida* Palib., *V. alexandrowiana* (W. Becker) Juz., *V. alexejana* Kamelin & Junussov, *V. bambusetorum* Hand.-Mazz., *V. baoshanensis* W. S. Shu, W. Liu & C. Y. Lan, *V. belophylla* Boissieu, *V. betonicifolia* Sm., *V. bhutanica* H. Hara, *V. boissieuana* Makino, *V. breviflora* Jungsim Lee & M. Kim, *V. cuspidifolia* W. Becker, *V. dactyloides* Schult., *V. forrestiana* W. Becker, *V. gmeliniana* Schult., *V. hancockii* W. Becker, *V. hirtipes* S. Moore, *V. inconspicua* Blume, *V. ingolensis* Elisafenko, *V. iwagawae* Makino, *V. japonica* Langsd. ex Ging., *V. jooi* Janka, *V. keiskei* Miq., *V. lactiflora* Nakai, *V. macroceras* Bunge, *V. magnifica* C. J. Wang & X. D. Wang, *V. mandshurica* W. Becker, *V. maximowicziana* Makino, *V. miaolingensis* Y. S. Chen, *V. microcentra* W. Becker, *V. mongolica* Franch., *V. multifida* Willd. ex Schult., *V. nujiangensis* Y. S. Chen & X. H. Jin, *V. pacifica* Juz., *V. patrinii* Ging., *V. pekinensis* (Regel) W. Becker, *V. perpusilla* Boissieu, *V. phalacrocarpa* Maxim., *V. philippica* Cav., *V. pinnata* L., *V. prionantha* Bunge, *V. rupicola* Elmer, *V. selkirkii* Pursh ex Goldie, *V. senzanensis* Hayata, *V. seoulensis* Nakai, *V. sieboldii* Maxim., *V. somchetica* K. Koch, *V. sphaerocarpa* W. Becker, *V. tashiroi* Makino, *V. tenuicornis* W. Becker, *V. tienschiensis* W. Becker, *V. tokaiensis* Sugim., nom. nud., *V. tokubuchiana* Makino, *V. trichopetala* C. C. Chang, *V. turkestanica* Regel & Schmalh., *V. ulleungdoensis* M. Kim & J. Lee, *V. umphangensis* S. Nansai, Srisanga & Suwanph., *V. variegata* Fisch. ex Link, *V. violacea* Makino, *V. yezoensis* Maxim., *V. yunnanfuensis* W. Becker, *V. yuzufeliensis* A. P. Khokhr.

*Distribution*.—North-temperate, with a diversity centre in northeastern Asia; only four species in Europe and one in North America, the scattered circumboreal *V. selkirkii*.

*Discussion*.—Section *Plagiostigma* subsect. *Patellares* is species-rich and easily characterised by the absence of stolons, and stipules adnate to the petiole in the lower third. The corolla can be of a deep lilac tone, sometimes fragrant but with a fragrance somewhat different from that of sect. *Viola* (e.g., *V. odorata*), and the spur of the bottom petal is usually relatively longer than in the other subsections of *Plagiostigma*. The lamina shape is extremely variable, from spathulate to cordate in outline, and with margins subentire to crenate or variously deeply divided. Some species form adventitious shoots from roots and have the ability to regenerate from cut roots (e.g., *V. prionantha*). Many species of the subsection have seeds that germinate directly without stratification.

Phylogenetic relationships within subsect. *Patellares* are contradictory. There is poor correspondence in patterns obtained from *ITS* sequences, cpDNA sequences, and morphology [77], but also among studies [82,86,87,89]. This may on one side indicate the presence of real genealogical conflicts resulting from incomplete lineage sorting, allopolyploidisation, and chloroplast introgression, but also taxonomic confusion and misidentifications.

Nested within subsect. *Patellares* is a pair of dwarf species from the Ryukyus Archipelago (Japan) with dwarf habit and 2*n* = 22, *Viola tashiroi* and *V. iwagawae*. These species form adventitious shoots from roots that superficially look like stolons [288].

Becker [248] erected grex *Gmelinianae* for a heterogeneous group of Central Asian rosette plants with cuneate or spathulate leaves and adnate stipules, which he later incorporated in grex *Adnatae* [1]. The *Gmelinianae* is, however, polyphyletic and here we redistribute its members among three sections: sect. *Plagiostigma* subsect. *Patellares* with *V. gmeliniana*, *V. perpusilla*, and *V. turkestanica*; sect. *Spathulidium* with *V. spathulata*; and sect. *Himalayum* with *V. kunawurensis*. The group consisting of *V. perpusilla*, *V. turkestanica*, and the similar *V. alata*, are atypical within subsect. *Patellares* in having subentire leaves and unmargined style; they however have the characteristic long spurs of that subsection while sect. *Himalayum* has a short spur.

 


**[2.13.7] *Viola* sect. *Plagiostigma* subsect. *Stolonosae* (**
Figure 3
**an and **
Figure 9
**di–dx)**


*Viola* subsect. *Stolonosae* (Kupffer) Kupffer in Kusnezow et al., Fl. Caucas. Crit. 3(9): 217. 1909 ≡ *Viola* [unranked; “Gruppe”] *Stolonosae* Kupffer in Oesterr. Bot. Z. 53: 329. 1903.—Lectotype (designated here): *Viola palustris* L.

=*Viola* subg. *Verbasculum* Nieuwl. & Kaczm. in Amer. Midl. Naturalist 3: 213. 1914.—Type: *Viola primulifolia* L.

=*Viola* [unranked] (”Gruppe”) *Vaginatae* W. Becker in Beih. Bot. Centralbl., Abt. 2, 36: 29. 1918 ≡ *Viola* ser. *Vaginatae* Taken in J. Sci. N.-E. Norm. Univ., Biol. 1: 86. 1955 ≡ *Viola* subsect. *Vaginatae* (W. Becker) P.Y.Fu, Fl. Pl. Herb. Chin. Bor.-Or. 6: 91. 1977 ≡ *Viola* sect. *Vaginatae* (W. Becker) Ching J.Wang, Fl. Reipubl. Popularis Sin. 51: 85. 1991.—Type (Shenzhen Code Art. 10.8): *Viola vaginata* Maxim.

*Description*.—Rhizome perennial; bulbils absent. Lateral stems present or absent: aboveground stolons, most leaves scattered; or rarely aerial stems with leaves in apical rosette (in *V. moupinensis*). Stipules free or occasionally up to 1/2 adnate (in *V. brachyceras*), pale, greenish, or brown, (linear-lanceolate to) lanceolate to ovate, acuminate, entire or remotely denticulate-fimbriate. Lamina lanceolate to reniform, base cuneate to deeply cordate, apex rounded to acuminate, margin subentire to crenate. Corolla white or pale violet. Sepals lanceolate to ovate; appendages short or long (0.5–2 mm), rounded or dentate. Lateral petals bearded or not; bottom petal shorter than, or subequal to, the other petals (6–20 mm), apex acute to emarginate; spur short (1–5 mm) and saccate. Style at apex margined and flattened, rarely bilobate.

*Diagnostic characters*.—Stolons (if present) with most leaves scattered AND sepals lanceolate to ovate AND stipules usually lanceolate to ovate AND style apex margined and flattened, rarely bilobate.

*Ploidy and accepted chromosome counts*.—4*x*, 8*x*; 2*n* = 20, 24, 44, 48.

*Age*.—Crown node age c. 12.7 Ma [45]; stem node age 13.5 (12.2–14.0) Ma [28].

*Included species*.—41. *Viola**adenothrix* Hayata, *V. binayensis* Okamoto & K. Ueda, *V. bissetii* Maxim., *V. blanda* Willd., *V. brachyceras* Turcz., *V. brevipes* (M. S. Baker) Marcussen, ined., *V. cochranei* H. E. Ballard, *V. davidii* Franch., *V. diamantiaca* Nakai, *V. epipsila* Ledeb., *V. fargesii* H. Boissieu, *V. glaucescens* Oudem., *V. grandisepala* W. Becker, *V. hultenii* W. Becker, *V. incognita* Brainerd, *V. jalapaensis* W. Becker, *V. javanica* W. Becker, *V. kjellbergii* Melch., *V. lanceolata* L., *V. macloskeyi* F. E. Lloyd, *V. maoershanensis* Y. S. Chen & Q. E. Yang, *V. mearnsii* Merr., *V. minuscula* Greene, *V. moupinensis* Franch., *V. nitida* Y. S. Chen & Q. E. Yang, *V. nuda* W. Becker, *V. occidentalis* (A. Gray) Howell, *V. palustris* L., *V. petelotii* W. Becker ex Gagnep., *V. pluviae* Marcussen, H. E. Ballard & Blaxland, *V. primulifolia* L., *V. principis* Boissieu, *V. renifolia* A. Gray, *V. rossii* Hemsl., *V. shikokiana* Makino, *V. striatella* H. Boissieu, *V. suecica* Fr., *V. thomsonii* Oudem., *V. vaginata* Maxim., *V. vittata* Greene, *V. yazawana* Makino

*Distribution*.—North-temperate; one species (*Viola lanceolata*) in northern South America. *Viola suecica* (=*V. achyrophora* Greene, *V. epipsiloides* Á. Löve & D. Löve, *V. epipsila* subsp. *repens* W. Becker) is circumboreal.

*Discussion*.—The delimitation of this subsection is “locked” by the existence of allopolyploids between distantly related internal lineages, one of which happens to be the type of the subsection (*Viola palustris*). The polyploids include the North American *V. blanda* and *V. incognita* (8*x*) which are allopolyploids of *V. renifolia* or perhaps more likely *V. brachyceras* (4*x*) and a taxon within the *V. primulifolia* group (4*x*); the Amphiatlantic *V. palustris* (8*x*) which is the alloploid of *V. minuscula* (=*V. pallens* auct., non (Banks) Brainerd; 4*x*) and *V. epipsila* (4*x*); the Pacific American *V. pluviae* (8*x*) which is the alloploid of *V. macloskeyi/occidentalis* (4*x*) and *V. suecica* (4*x*); and presumably also the North American *V. brevipes* [45,93]. These five allo-octoploids are no older than 2.5–5 Ma, and their marked boreal distributions suggest they originated in response to the climate cooling and repeated glaciations in the Pleistocene [93].

Disregarding allopolyploidy, at least four informal species groups are nevertheless recognisable at the 4*x* level based on published phylogenetic studies (Figure 8; [45,82,86,87,287]). These include (1) a clade comprising the Chinese species *V. davidii* and *V. grandisepala*; (2) a clade of mostly hairy species occurring in eastern Asia and northern North America comprising *V. principis*, *V. renifolia*, *V. yazawana*, and presumably also *V. adenothrix* and *V. brachyceras*; (3) a clade of mostly large species with acuminate laminas and larger pale violet to pink corollas and broad somewhat sheathing denticulate stipules comprising the circumboreal *V. epipsila-suecica* complex, *V. moupinensis*, and most of Becker’s [1] grex *Vaginatae*, i.e., *V. bissetii*, *V. diamantiaca*, *V. vaginata*, etc.; and, finally, (4) the North American stoloniferous species comprising *V. primulifolia*, *V. lanceolata*, *V. macloskeyi*, *V. minuscula*, etc., by Marcussen et al. [45] referred to as “grex *Primulifoliae*”. 

The group of species having a creeping, remotely noded rhizome and which was previously informally designated as the *Palustres* grex comprises a subset of the species in clade 3, i.e., *V. epipsila* and *V. suecica*, and their allopolyploids, i.e., *V. palustris*, *V. pluviae*, and *V. brevipes*, formed with species in clade 4.

Phylogenetic studies of the north-temperate species of subsect. *Stolonosae* [45,93] indicate that a narrow species concept coinciding with morphological-geographic units best applies to these taxa. This concept challenges in particular the traditional classification of the North American taxa into a few, broadly defined species based on lamina shape [289,290,291]. Hence, we consider *V. minuscula* distinct from *V. macloskeyi*, *V. occidentalis* distinct from *V. primulifolia*, *V. vittata* distinct from *V. lanceolata*, and *V. suecica* distinct from *V. epipsila*. Among the octoploids *V. brevipes* and *V. pluviae* are distinct from *V. palustris*, and *V. incognita* is distinct from *V. blanda*. Taxonomy, variation, and phylogeography in this circumboreal complex are poorly understood and require further study throughout its range. Certain characters traditionally considered diagnostic, such as leaf shape and pubescence in *V. palustris*, have proven variable even within single specimens [292,293].

The chromosome number 2*n* = 20, apparently at odds with the predominance of 2*n* = 24 in this subsection, has been reported several times in *Viola brachyceras* and also in the closely related *V. yazawana*, for which also 2*n* = 40 has been reported (cf. [61] and references therein); this number could also explain 2*n* = 44 (not 48) in the octoploids *V. blanda* and *V. incognita*, and possibly also in *V. maoershanensis* [294]. Counts of 2*n* = 20 outside of this species group within subsect. *Stolonosae* are probably errors.

 


**[2.14] *Viola* sect. *Rubellium* (**
Figure 3
**ao and **
Figure 9
**dy–ea)**


*Viola* sect. *Rubellium* W. Becker in Nat. Pflanzenfam., ed. 2 [Engler & Prantl], 21: 374. 1925.—Type (Shenzhen Code Art. 10.8): *Viola rubella* Cav.

=*Viola* [unranked] § II. Tri(-Pluri-)Caules Reiche in Fl. Chile [Reiche] 1: 140. 1896, nom. inval. (Shenzhen Code Art. 21.2)

*Description*.—Perennial subshrubs. Axes morphologically (weakly) differentiated in a perennial monopodial aerial stem and lateral monopodial aerial elongated stems bearing flowers; lateral stems with distichous phyllotaxy in Viola portalesia. Stipules small, bract-shaped, fimbriate. Lamina oblong to lanceolate, base cuneate, margin crenate, short-petiolate. Peduncle long. Corolla violet to whitish inward with a greenish throat, or magenta to pink throughout (*V. rubella*). Spur short. Style clavate, at apex neither margined nor bearded, bent into a simple, ventrad rostellum, or apex rounded with the rostellum on the ventral surface. Cleistogamous flowers not produced. Diploid. Base chromosome number *x* = 6.

*Diagnostic characters*.—Subshrubs AND corolla magenta or violet AND style apex strongly bent ventrad or with stigma on ventral side AND diploid with 2n = 12.

*Ploidy and accepted chromosome counts*.—2*x*; 2*n* = 12 (*V. rubella*).

*Age*.—Crown node age 1.6 (0.4–2.2) Ma; stem node 26.5 (25.7–26.8) Ma [28].

*Included species*.—3. *Viola**capillaris* Pers., *V. portalesia* Gay, *V. rubella* Cav.

*Distribution*.—Central Chile (Figure 28).

*Discussion*.—Section *Rubellium* is phylogenetically isolated and the only subshrubby diploid lineage within subg. *Viola* [60]. The original delimitation was established by Becker (1925). Previously, Reiche [122,130] circumscribed the group under an invalid taxonomic rank (i.e., the unranked *Tri(-Pluri-)Caules* within the invalid Division *Sparsifoliae*). Sparre [63] included in sect. *Rubellium* also the herbaceous *V. huidobrii*, by us reclassified in sect. *Viola* subsect. *Rostratae*.

 


**[2.15] *Viola* sect. *Sclerosium* (**
Figure 3
**ap and **
Figure 9
**eb–ed)**


*Viola* sect. *Sclerosium* W. Becker in Nat. Pflanzenfam., ed. 2 [Engler & Prantl], 21: 374. 1925.—Lectotype (designated here): *Viola cinerea* Boiss.

=*Viola* [sect. *Nomimium*; unranked] §.2. *Cinereae* Boiss., Fl. Orient. 1: 451. 1867, p. p. (excl. *V. spathulata*) ≡ *Viola* [unranked] (”Gruppe”) *Cinereae* Boiss. em. W. Becker in Beih. Bot. Centralbl., Abt. 2, 36: 36. 1918

*Description*.—Annual herbs or perennial subshrubs, glabrous or densely short-pubescent. Axes morphologically differentiated in aerial stems and short axillary branches bearing cleistogamous flowers. Stipules small, lanceolate. Lamina ovate to lanceolate, remotely denticulate, petiolate. Corolla pink with a green throat. Spur short and thick. Style slender and cylindrical or slightly clavate, crested; crest a pair of apical or subapical lateral ear-like processes. Simultaneous production of chasmogamous in upper leaf axils and cleistogamous flowers on short branches in lower leaf axils. Allotetraploid (CHAM + MELVIO). Secondary base chromosome number *x*’ = 11.

*Diagnostic characters*.—Style with a pair of apical or subapical lateral ear-like processes. Base chromosome number *x* = 11.

*Ploidy and accepted chromosome counts*.—4*x*, 8*x*; 2*n* = 22 (*V. stocksii*). 

*Age*.—Crown node 3.5–10 Ma [150].

*Included species*.—7. *Viola**behboudiana* Rech. f. & Esfand., *V. cinerea* Boiss., *V. erythraea* (Fiori) Chiov., *V. etbaica* Schweinf., *V. kouliana* Bhellum & Magotra, *V. somalensis* Engl., *V. stocksii* Boiss.

*Distribution*.—Northeastern Africa to southwestern Asia (Figure 29). Disjunctly distributed in the monsoon region on both sides of the Red Sea, Sokotra and the Arabic coast of the Indian Ocean, southern Iran, most of Pakistan, and northwestern India.

*Discussion*.—Variation patterns within sect. *Sclerosium* are poorly understood. It contains closely related races that are difficult to delimit but differ in distribution, life history traits (annual or perennial), pubescence, and style shape. Nine allopatric taxa have been described [1,150,295,296] but most authorities have interpreted the variation as more or less continuous and have retained only one or two variable species [79,297]. However, a detailed study of the Iranian taxa [91,150] revealed three morphologically discrete species and allopolyploid relationships among them (*V. stocksii* 4*x*; *V. cinerea* 8*x*; *V. behboudiana* 8*x*), which may suggest more taxa warrant recognition within the section. Section *Sclerosium* may have started to diversify in Late Miocene 3.5–10 Ma ago [150]. The young age corroborates the low morphological differentiation among taxa. The crown group age coincides with the initiation (or intensification) of the Indian monsoon system, caused by the uplift of the Himalayas and the East African mountain plateaus [298,299]. The precipitation brought by the monsoon plays an important role for the flora in this otherwise arid region. 

Section *Sclerosium* is vegetatively somewhat similar to sect. *Xylinosium* (especially *Viola scorpiuroides*) but the sections are distantly related, allopatric, they differ in several important characters, and any similarity must be interpreted as parallel adaptation to arid environments.

 


**[2.16] *Viola* sect. *Spathulidium* (**
Figure 3
**aq and **
Figure 9
**ee)**


*Viola* sect. *Spathulidium* Marcussen, sect. nov.—Type: *Viola spathulata* Willd.

*Description*.—Perennial herbs. Axes not morphologically differentiated. All stems rhizomatous, forming cushions. Stipules ¾ adnate to petiole. Lamina spathulate to lanceolate, subentire, tapering into short and indistinct petiole. Corolla pale violet, pink or whitish. Spur 1.5–4 mm, longer than tall. Style clavate, geniculate at base, at apex 2-lobed, with a distinct dorsolateral margin and ventral rostrum. Cleistogamous flowers not produced. Allo-octoploid (CHAM + MELVIO). ITS sequence of MELVIO type.

*Diagnostic characters*.—Lamina spathulate to lanceolate, subentire, tapering into short and indistinct petiole AND style clavate, at apex 2-lobed, with a distinct dorsolateral margin AND cleistogamous flowers not produced.

*Ploidy and accepted chromosome counts*.—[Section by origin 8*x*], 16*x* (*V. spathulata*). Chromosome number unknown.

*Age*.—Crown node c. 1 Ma; stem node 5.0 (4.2–5.3) Ma [28].

*Included species*.—3. *Viola**maymanica* Grey-Wilson, *V. pachyrrhiza* Boiss. & Hohen., *V. spathulata* Willd. ex Schult.

*Distribution*.—Disjunctly distributed in the high mountains of southwestern Asia (Figure 30): *Viola pachyrrhiza* in northeastern Iraq and southern Iran; *V. spathulata* in northern Iran (Elburs mountains); and *V. maymanica* in northwestern Afghanistan.

*Etymology*.—The name *Spathulidium* refers to the distinctive spathulate leaves.

*Discussion*.—Section *Spathulidium* is an allooctoploid CHAM + MELVIO lineage and has retained the MELVIO homoeolog for ITS (Figure 2). The lineage is morphologically recognisable on being cushion plants, inhabiting rock fissures, with spathulate short-petiolate leaves, a somewhat bilobed style, and the absence of cleistogamous flowers. The *Spathulidium* lineage is inferred to be the alloploid of two unknown tetraploid lineages; further allopolyploidy based on 8*x* may have happened in *V. spathulata* (16*x*) [28]. The three species of sect. *Spathulidium* have traditionally been grouped within sect. *Plagiostigma* subsect. *Patellares* based on being violet-flowered rosette plants with narrow leaves and adnate stipules [1,248]. However, sect. *Spathulidium* differs from subsect. *Patellares* in being cushion plants, having leaves with entire or subcrenate margins, in lacking cleistogamy, and in ploidy. Section *Spathulidium* differs from sect. *Himalayum* in being cushion plants, in having a margined style apex and a much longer spur, and in lacking cleistogamous flowers. Both sections are 8*x* but have different allopolyploid origins.

Section *Spathulidium* is most closely related to the African sect. *Abyssinium* (see note under the latter).

 


**[2.17] *Viola* sect. *Tridens* (**
Figure 3
**ar and **
Figure 9
**ef)**


*Viola* sect. *Tridens* W. Becker in Nat. Pflanzenfam., ed. 2 [Engler & Prantl], 21: 376. 1925.—Type (Shenzhen Code Art. 10.8): *Viola tridentata* Sm.

*Description*.—Perennial procumbent herb, forming perennial herbaceous mats with branched stems. Axes morphologically differentiated in elongated rhizome and lateral, short floriferous stems with distichous phyllotaxy. Stipules completely adnate to the pseudopetiole or only with the free end forming a short tooth. Leaves tridentate on floriferous shoots, bilobate or entire on sterile shoots, small, imbricated, fleshy. Corolla small, white with violet striation. Spur short. Anthers with scattered hairs. Style cylindrical, at base curved, slightly tapering towards apex, filiform. Cleistogamous flowers not produced. Allohexaploid. Secondary base chromosome number *x’* = 20.

*Diagnostic characters*.—Leaves tridentate, distichous, imbricate.

*Ploidy and accepted chromosome counts*.—6*x*; 2*n* = 40.

*Age*.—Crown node age not applicable (monotypic lineage); stem node 9.2 (1.0–14.7) Ma [28].

*Included species*.—1. *Viola tridentata* Sm.

*Distribution*.—Southernmost South America: Argentina, Chile, Falkland/Malvinas Islands (Figure 31).

*Discussion*.—Section *Tridens* is immediately recognisable by the tridentate, distichous, and imbricate leaves. Phylogenetically, sect. *Tridens* is allohexaploid and two of its diploid genomes are shared with other polyploid southern hemisphere lineages, i.e., *Leptidium* on the one side, and *Chilenium/Erpetion* on the other (Figure 5). The original inference by Marcussen et al. [28] that *Tridens* is 12*x* was based on incorrect counts for sect. *Erpetion* and sect. *Tridens* which overestimated the ploidy.

The delimitation of sect. *Tridens* is the same as Becker’s [1] except for the inclusion of *V. muscoides* Phil. as a synonym of *V. tridentata* based on shared diagnostic characters. *Viola muscoides* was erroneously synonymised with *Myrteola nummularia* (Poir.) O. Berg (Myrtaceae) by Kausel [300].

 

**[2.18] *Viola* sect. *Viola* (**Figure 3**as**–**at and **Figure 9**eg–fa)**

≡*Viola* sect. *Nomimium* Ging., p.p. in Mém. Soc. Phys. Genève 2(1): 28. 1823, nom. inval. (Szhenzhen Code Art. 22.2; *Viola odorata* L.) ≡ *Viola* subg. *Nomimium* (Ging.) Peterm., Deutschl. Fl.: 64. 1846, nom. inval. (Szhenzhen Code Art. 22.2)

≡*Viola* [sect. *Nomimium*; unranked] §.4. *Rostellatae* Boiss., Fl. Orient. 1: 451. 1867, nom. inval. (Szhenzhen Code Art. 22.2; *Viola odorata* L.) ≡ *Viola* subsect. *Rostellatae* (Boiss.) Rouy & Foucaud, Fl. France [Rouy & Foucaud] 3: 3. 1896, nom. inval. (Szhenzhen Code Art. 22.2). ≡ *Viola* sect. *Rostellatae* (Boiss.) J. C. Clausen in Madroño 17: 196. 1964, nom. inval. (Shenzhen Code Art. 22.2)

≡*Viola* [sect. *Nomimium*; unranked] a. *Rostellata* Nyman, Consp. Fl. Eur. 1: 76. 1878, nom. inval. (Szhenzhen Code Art. 22.2; *Viola odorata* L.)

*Description*.—Perennial herbs. Axes morphologically differentiated in a perennial rhizome with lateral stems; sometimes only one type of stem produced. Rhizome creeping or vertical, branched or not, with apical rosette of leaves. Lateral stems annual aerial stems, stolons, or absent. Stipules usually free, entire, dentate, laciniate or fimbriate, sometimes large and foliaceous. Lamina reniform to rhomboid, crenulate, petiolate. Flowers scented or scentless. Corolla violet to white, with a white throat. Spur (much) longer than tall, up to 16 mm. Style clavate or rarely filiform, at apex not margined, bearded or not. Capsule trigonous and explosive or globose and non-explosive. Cleistogamous flowers usually produced; cleistogamy seasonal, rarely facultative. Allotetraploid (CHAM + MELVIO). Secondary base chromosome number *x’* = 10. *ITS* sequence of MELVIO type.

*Diagnostic characters*.—Perennial herbs AND corolla with a white throat AND style clavate, unmargined AND base chromosome number *x* = 10.

*Ploidy and accepted chromosome counts*.—4*x*, 8*x*; 12*x*; 2*n* = 20, 40, 58, 60.

*Age*.—Crown node 11.8 (10.1–12.4) Ma [28].

*Included species*.—75.

*Distribution*.—Throughout the temperate zone of the northern hemisphere; one species in southern South America (Figure 32). Diversity centre in western Eurasia.

*Discussion*.—Section *Viola* is phylogenetically an allotetraploid CHAM + MELVIO lineage and has retained the MELVIO homoeolog for ITS (Figure 6). Karyologically it is characterised by the secondary base chromosome number *x’* = 10, and morphologically by the clavate unmargined style. Section *Viola* is one of three species-rich segregates of Becker’s widely delimited sect. *Nomimium*, which comprised nearly all the temperate herbaceous, violet- or white-flowered taxa with seasonal cleistogamy. Section *Viola* differs from both sect. *Plagiostigma* and sect. *Nosphinium* in having the base chromosome number *x* = 10 and a unmargined style, sometimes bearded above.

Section *Viola* is phylogenetically subdivided into two morphologically well-defined groups (Figure 6 and Figure 33), here treated as subsect. *Rostratae* and subsect. *Viola*.

 


**Key to the subsections of sect. *Viola***


1a.Capsules globose, often hairy, non-dehiscent, on decumbent peduncles. Seeds large, with conspicuous elaiosome more than half the length of the seed (myrmecochory). Lateral stems stolons or absent. Style glabrous. ........................................................................................................................................................................................................................................................................ **subsect. *Viola***1b.Capsule elongate, trigonous, glabrous, forcibly ejecting seeds after dehiscence, on erect peduncles at maturity. The elaiosome much less than half the length of the seed (diplochory). Lateral stems usually aerial (occasionally stolons or absent). Style bearded above or beardless. ............................................................................................................................................................................................................................................................... **subsect. *Rostratae***

 


**[2.18.1] *Viola* sect. *Viola* subsect. *Rostratae* (**
Figure 3
**as and **
Figure 9
**eg–ev)**


*Viola* subsect. *Rostratae* (Kupffer) W. Becker in Acta Horti Gothob. 2: 285. 1926 ≡ *Viola* [unranked] (“Gruppe”) *Rostratae* Kupffer in Oesterr. Bot. Z. 53: 328. 1903 ≡ *Viola* sect. *Rostratae* (Kupffer) Kupffer in Kusnezow et al., Fl. Caucas. Crit. 3(9): 193. 1909 ≡ *Viola* [sect. *Nomimium*) [unranked] *Rostratae* (Kupffer) Becker in Nat. Pflanzenfam., ed. 2 [Engler & Prantl], 21: 365. 1925.—Lectotype (designated here): *Viola riviniana* Rchb.

≡*Viola* sect. *Trigonocarpea* Godron, Fl. Lorraine, ed. 2, 1: 88. 1857 ≡ *Viola* subsect. *Trigonocarpea* (Godr.) P. Y. Fu, Fl. Pl. Herb. Chin. Bor.-Or. 6: 82. 1977; Vl. V. Nikitin in Novosti Sist. Vyssh. Rast. 33: 178. 2001 (isonym).—Lectotype (Nikitin 1996 [301], page 189): *Viola riviniana* Rchb.

=*Viola* [unranked] *Rosulantes* Borbás in Hallier & Wohlfarth, Syn. Deutsch. Schweiz. Fl., ed. 3, 1: 196. 1892 ≡ *Viola* subsect. *Rosulantes* (Borbás) J. C. Clausen in Madroño 17: 196. 1964, nom. inval. (Shenzhen Code Art. 41.5)

=*Lophion* subg. *Eucentrion* Nieuwl. & Kaczm. in Amer. Midl. Naturalist 3: 216. 1914.—Type: *Viola rostrata* Pursh

=*Lophion* subg. *Rhabdotion* Nieuwl. & Kaczm. in Amer. Midl. Naturalist 3: 216. 1914.—Type: *Viola striata* Aiton

=*Viola* [sect. *Nomimium*; unranked] *Umbraticolae* W. Becker in Repert. Spec. Nov. Regni Veg. 19: 396. 1923.—Type (Shenzhen Code Art. 10.8): *Viola umbraticola* Kunth

=*Viola* [unranked] *Repentes* Kupffer in Oesterr. Bot. Z. 53: 329. 1903 ≡ *Viola* subsect. *Repentes* (Kupffer) Juz. in Schisk. & Bobrov, Fl. URSS 15: 401.—Type: *Viola uliginosa* Besser

≡*Viola* sect. *Icmasion* Juz. ex Tzvelev in Cvelev, Opred. Sosud. Rast. Severo-Zapadn. Rossii: 679. 2000.—Type: *Viola uliginosa* Besser

=*Viola* subsect. *Grypocerae* Espeut in Botanica Pacifica 9(1): 16. 2020.—Type: *Viola grypoceras* A. Gray

=*Viola* [unranked] *Mirabiles* Nyman Syll. Fl. Eur.: 226. 1855, nom. inval. (Shenzhen Code Art. 38.1) ≡ *Viola* [unranked] b2 *Mirabiles* Nyman ex Borbás Syn. Deutsch. Schweiz. Fl., ed. 3, 1: 195. 1890 ≡ *Viola* subsect. *Mirabiles* (Nyman ex Borbás) Juz. in Schischk. & Bobrov, Flora URSS 15: 375. 1949 ≡ *Viola* sect. *Mirabiles* (Nyman ex Borbás) Vl. V. Nikitin in Bot. Zhurn. (Moscow & Leningrad) 83(3): 130. 1998.—Type (Shenzhen Code Art. 10.8): *Viola mirabilis* L.

=*Viola* [sect. *Chilenium*] subsect. *Coeruleae* Sparre in Lilloa 17: 414. 1949.—Type: *Viola huidobrii* Gay

*Description*.—Rhizome with an apical leaf rosette and lateral aerial stems or stolons, or all stems rhizomatous, or all stems aerial. Stipules often large and foliaceous. Style bearded or not. Capsule trigonous, forcibly ejecting seeds after dehiscence. Seeds with a small elaiosome.

*Diagnostic characters*.—Capsules trigonous, erect at maturity, explosive; seeds with small elaiosome covering less than 1/2 of the raphe.

*Ploidy and accepted chromosome counts*.—4*x*, 8*x*, 12*x*; 2*n* = 20, 40, 58, 60.

*Age*.—Crown node c. 11 Ma [92]; stem node 11.8 (10.1–12.4) Ma [28].

*Included species*.—51. *Viola**acuminata* Ledeb., *V. adunca* Sm., *V. aduncoides* Á. Löve & D. Löve, *V. anagae* Gilli, *V. appalachiensis* L. K. Henry, *V. canina* L., *V. caspia* (Rupr.) Freyn, *V. dirphya* Tiniakou, *V. elatior* Fr., *V. faurieana* W. Becker, *V. ganpinensis* W. Becker, ined. [E. Bodinier 2176], *V. grayi* Franch. & Sav., *V. grypoceras* A. Gray, *V. henryi* H. Boissieu, *V. huidobrii* Gay, *V. jordanii* Hanry, *V. kosanensis* Hayata, *V. kusanoana* Makino, *V. labradorica* Schrank, *V. lactea* Sm., *V. laricicola* Marcussen, *V. mariae* W. Becker, *V. mauritii* Tepl., *V. mirabilis* L., *V. obtusa* (Makino) Makino, *V. oligyrtia* Tiniakou, *V. ovato-oblonga* (Miq.) Makino, *V. papuana* W. Becker & Pulle, *V. pendulicarpa* W. Becker, *V. percrenulata* H. E. Ballard, ined. [H. S. Gentry 7247], *V. pseudomirabilis* H. J. Coste, *V. pumila* Chaix, *V. reichenbachiana* Jord. ex Boreau, *V. riviniana* Rchb., *V. rostrata* Pursh, *V. rupestris* F. W. Schmidt, *V. sacchalinensis* H. Boissieu, *V. serrula* W. Becker, *V. shinchikuensis* Yamam., *V. sieheana* W. Becker, *V. stagnina* Kit. ex Schult., *V. stewardiana* W. Becker, *V. striata* Aiton, *V. tanaitica* Grosset, *V. thibaudieri* Franch. & Sav., *V. uliginosa* Besser, *V. umbraticola* Kunth, *V. utchinensis* Koidz., *V. walteri* House, *V. websteri* Hemsl., *V. willkommii* R. Roem. ex Willk..

*Distribution*.—North-temperate, except for *Viola huidobrii* in southern South America and *V. papuana* in New Guinea. *Viola riviniana* is naturalised in western North America, Australia, and New Zealand.

*Discussion*.—Within sect. *Viola*, this lineage is characterised by the explosive capsules, borne on erect peduncles at maturity (in fact a plesiomorphic trait within *Viola*). Subsection *Rostratae* is widely distributed in the temperate zone of Eurasia and North America; one species occurs in southern South America, and one in New Guinea. Becker [1] included in grex *Rostratae* only species with aerial floriferous stems but subsequent studies have shown that the subsection should be more inclusive. 

Subsect. *Rostratae* has often been further subdivided based on shoot system but none of the segregates delimit monophyletic units, due to extensive allopolyploidy and presumably also parallel evolution. The vast majority of the species in subsect. *Rostratae* have a basal leaf rosette and lateral floriferous stems; these have traditionally been referred to as grex *Rosulantes* Borbás. In a few species the chasmogamous flowers are produced from the leaf rosette and the aerial stems develop after chasmogamous anthesis; these have been referred to as grex *Mirabiles* Nym. (i.e., *V. mirabilis*, *V. pseudomirabilis*, and *V. willkommii*). Other species have lateral stems that are stolon-like (e.g., *V. anagae*, *V. appalachiensis*, *V. papuana*, *V. walteri*; grex *Repentes* Kupffer: *V. uliginosa*) or absent altogether (e.g., *V. ganpinensis*, *V. pendulicarpa*, and *V. shinchikuensis*; grex *Umbraticolae* W. Becker: *V. percrenulata* and *V. umbraticola*). Finally, a fourth group of species that lack the basal rosette and instead have a sympodial growth system from annual floriferous stems has been referred to as grex *Arosulatae* Borbás (i.e., *V. canina*, *V. elatior*, *V. lactea*, *V. pumila*, *V. stagnina*). The formal greges *Arosulatae*, *Mirabiles*, and *Rosulantes* are superfluous as they are *de facto* synonyms of the higher taxon subsect. *Rostratae* as a result of being interconnected by allopolyploids (Figure 33, [84,92,302]). Among these, only the group formerly referred to as “*Arosulatae*” may merit recognition on ecological grounds. These western Eurasian species are ecological specialists to floodplains [303,304] and each possesses at least one *stagnina* genome. We suggest that this group be referred to informally as the *V. stagnina* group. Becker included here *V. acuminata* and *V. jordanii* by mistake: neither has a sympodial growth system lacking a basal rosette nor possesses a *stagnina* genome (Figure 33, [84,302]).

Morphologically, the southern South American *Viola huidobrii* (including its synonym *V. brachypetala*) belongs in subsect. *Rostratae*, based on having a rhizome with a terminal leaf rosette and lateral floriferous stems, violet corolla, long spur, and the characteristic rostellate style (Figure 9el). *Viola huidobrii* was previously included in sect. *Chilenium* [1,62] or sect. *Rubellium* [63]. It is the only species of sect. *Viola* native to the southern hemisphere. The Taiwanese endemic *V. shinchikuensis* (2*n* = 20) is (erroneously?) reported to be similar to subsect. *Viola* in having globose capsules borne on prostrate peduncles when mature [75,305] but it is phylogenetically placed in subsect. *Rostratae* [86] (Figure 2) with which it also shares numerous typical traits, e.g., bearded style, acute sepals with dentate appendages, bracteoles in the uppermost part of the peduncle, and thick non-hyaline stipules. The New Guinean endemic *V. papuana* has an unusual filiform style (which puzzled Becker; Figure 9eo) and isolated distribution but is a good match for subsect. *Rostratae* in other morphological characters, including the 4–9 mm long, upcurved spur and a pale violet corolla, and lateral stems or stolons. The reported chromosome count of 2*n* = 48 [74] is dubious.

Subsect. *Rostratae*, and sect. *Viola* as a whole, appears to have originated in western Eurasia. Only one clade, grex *Rosulantes* s.str., has dispersed into eastern Asia, North America, and South America.

A read-leaved mutant of *V. riviniana*, f. *purpurea* auct., is sometimes grown as an ornamental, often under the erroneous name *V. labradorica* hort. non Schrank.

 


**[2.18.2] *Viola* subsect. *Viola* (**
Figure 3
**at and **
Figure 9
**ew–fa)**


=*Viola* sect. *Odoratae* Boiss. in Diagn. Pl. Orient. 8: 51. 1849, nom. inval. (Szhenzhen Code Art. 22.2; *Viola odorata* L.)

=*Viola* sect. *Hypocarpea* Godron, Fl. Lorraine, ed. 2, 1: 86. 1857 ≡ *Viola* subsect. *Hypocarpea* (Godron) P. Y. Fu, Fl. Pl. Herb. Chin. Bor.-Or. 6: 82. 1977, nom. inval. (Szhenzhen Code Art. 22.2; *Viola odorata* L.)

=*Viola* [unranked] (”Gruppe”) *Uncinatae* Kupffer in Oesterr. Bot. Z. 53: 328. 1903, nom inval. (Szhenzhen Code Art. 22.2; *Viola odorata* L.) ≡ *Viola* sect. *Uncinatae* (Kupffer) Kupffer in Kusnezow et al., Fl. Caucas. Crit. 3(9): 174. 1909, nom. inval. (Szhenzhen Code Art. 22.2)

=*Viola* [unranked] a) *Curvato-pedunculatae* W. Becker in Beih. Bot. Centralbl., Abt. 2, 26: 1. 1910, nom. inval. (Szhenzhen Code Art. 22.2; *Viola odorata* L.)

=*Viola* subg. *Euion* Nieuwl. & Kaczm. in Amer. Midl. Naturalist 3: 211. 1914, nom. inval. (Szhenzhen Code Art. 22.2; *Viola odorata* L.)

=*Viola* [unranked] α *Lignosae* W. Becker in Beih. Bot. Centralbl., Abt. 2, 26: 1. 1910 ≡ *Viola* [unranked] (“Gruppe”) D. *Lignosae* W. Becker in Nat. Pflanzenfam. ed. 2 [Engler & Prantl], 21: 367. 1925.—Lectotype (designated here): *Viola chelmea* Boiss.

=*Viola* [unranked] (“Gruppe”) *Serpentes* W. Becker in Beih. Bot. Centralbl., Abt. 2, 40: 102. 1924—*Viola* subsect. *Serpentes* (W. Becker) W. Becker in Acta Horti Gothoburg. 2: 287. 1926—*Viola* ser. *Serpentes* (W. Becker) Steenis in Bull. Jard. Bot. Buitenzorg, ser. 3, 13 (1933–1936): 259. 1934—*Viola* sect. *Serpentes* Ching J. Wang, Fl. Reipubl. Popularis Sin. 51: 88. 1991.—Type (Shenzhen Code Art. 10.8): *Viola serpens* Wall. ex Ging. (=*V. pilosa* Blume)

*Description*.—Rhizome with apical rosette of leaves. Lateral stolons present or absent. Stipules free, not foliaceous. Style beardless. Capsule globose, non-explosive. Seeds with large elaiosome.

*Diagnostic characters*.—Capsules globose, usually hairy, decumbent at maturity, non-dehiscent. Seeds with a large elaiosome covering 1/2–3/4 of the raphe.

*Ploidy and accepted chromosome counts*.—4*x*, 8*x*; 2*n* = 20, 40.

*Age*.—Crown node age c. 5 Ma [92]; stem node 11.8 (10.1–12.4) Ma [28].

*Included species*.—24. *Viola**alba* Besser, *V. ambigua* Waldst. & Kit., *V. barhalensis* G. Knoche & Marcussen, *V. bocquetiana* Yıld., *V. canescens* Wall., *V. chelmea* Boiss., *V. collina* Besser, *V. hirta* L., *V. hondoensis* W. Becker & H. Boissieu, *V. indica* W. Becker, *V. isaurica* Contandr. & Quézel, *V. jangiensis* W. Becker, *V. jaubertiana* Marès & Vigin., *V. kizildaghensis* Dinç & Yıld., *V. libanotica* Boiss., *V. odorata* L., *V. pilosa* Blume, *V. pyrenaica* Ramond ex DC., *V. sandrasea* Melch., *V. sintenisii* W. Becker, *V. suavis* M. Bieb., *V. thomasiana* Songeon & E. P. Perrier, *V. vilaensis* Hayek, *V. yildirimlii* Dinç & Bagci

*Distribution*.—Eurasia; diversity centre in southern Europe. *Viola odorata* is naturalised throughout the temperate zone.

*Discussion*.—The principal apomorphy of subsect. *Viola* is the globose and non-explosive capsules borne on decumbent peduncles, containing large seeds with a conspicuous elaiosome, an adaptation to obligate myrmecochory. Subsection *Viola* as circumscribed here comprises three of Becker’s [1] greges. These include grex *Uncinatae* W. Becker (*V. odorata*, etc.) with both stolonose and estolonose temperate taxa, grex *Lignosae* W. Becker (*V. chelmea*, etc.) with estolonose taxa from the northeastern Mediterranean region, and parts of grex *Serpentes* W. Becker (*V. pilosa*, etc.) with stolonose taxa from southern Asia. The presence or absence of stolons has been used to classify species within the subsection but does not delimitate monophyletic groups [116]. At least in European species, the transitions from the stolonose condition (ser. *Flagellatae* Kittel) to the estolonose condition (ser. *Eflagellatae* Kittel) seems to have occurred several times and by different genetic mechanisms, and the two morphological groups are also linked by allopolyploidy, i.e., *V. suavis* (8*x*) [116]. Grex *Serpentes* has been demonstrated to be an artificial aggregate of species [229], most of them belonging in sect. *Viola* subsect. *Viola* or in various sect. *Plagiostigma* subsections.

A few species are grown as ornamentals, primarily for their fragrant flowers, i.e., *V. odorata* and filled forms of *V. alba* subsp. *dehnhardtii* (Ten.) W. Becker referred to as ‘Parma’ violets or ‘Violette de Toulouse’ [7,20]. The former (Figure 1) has been cultivated for the production of essential oil for the perfume industry [16,17].

 


**[2.19] *Viola* sect. *Xanthidium* (**
Figure 3
**au and **
Figure 9
**fb,fc)**


*Viola* sect. *Xanthidium* Marcussen, Nicola, J. M. Watson, A. R. Flores & H. E. Ballard, sect. nov.—Type: *Viola flavicans* Wedd.

*Description*.—Perennial herbs. Axes not morphologically differentiated. All stem rhizomatous, with leaves in loose apical rosettes. Stipules partially or largely adnate to the petiole, narrow, shallowly glandular-lacerate. Lamina lanceolate, remotely crenate, petiolate. Bracteoles narrow, shallowly glandular-lacerate. Corolla yellow with brown striation. Spur short. Style clavate, geniculate at the base, when fresh ellipsoid with broadly rounded apex (in dried condition with flattened apex), the stigmatic orifice on a small rostellum on ventral surface, bearded (*Viola flavicans*; Figure 9fb) or beardless (*V. pallascaensis*; Figure 9fc). Cleistogamous flowers apparently produced; type of cleistogamy unknown.

*Diagnostic characters*.—Rosulate herbs AND bracteoles glandular-lacerate AND corolla yellow AND style ellipsoid with broadly rounded apex when fresh, flattened when dry.

*Distribution*.—Disjunct in central-western South America (northwestern Argentina and Bolivia, central-eastern Peru) (Figure 34).

*Included species*.—2. *Viola flavicans* Wedd., *V. pallascaensis* W. Becker

*Etymology*.—The name *Xanthidium* is based on the Greek translation of the species epithet of the type species, *Viola flavicans*, which refers to its yellow corolla.

*Discussion*.—Section *Xanthidium* has not yet been subject to phylogenetic analysis nor has it been characterised at the chromosomal level. Becker placed neither of these species (nor their current synonyms) in any section. He identified the taxa as related, but did not include them in his genus treatment [1]. Later, Sparre ([63], page: 348) viewed this group (as the “*V. flavescens*-group”) as “intermediary between the sections *Chilenium* and *Andinium*”. Nicola [80] placed *V. flavicans* in sect. *Nomimium* Ging., an artificial aggregate of numerous northern hemisphere lineages and sections.

 


**[2.20] *Viola* sect. *Xylinosium* (**
Figure 3
**av and **
Figure 9
**fd,fe)**


*Viola* sect. *Xylinosium* W. Becker in Nat. Pflanzenfam., ed. 2 [Engler & Prantl], 21: 373. 1925.—Lectotype (designated here): *Viola arborescens* L.

=*Viola* [sect. *Nomimium*; unranked] *Fruticulosa* Nyman, Consp. Fl. Eur. 1: 76. 1878, nom. inval. (Shenzhen Code Art. 32.1)

*Description*.—Perennial subshrubs. Axes not morphologically differentiated. All stems aerial, decumbent or ascendent. Stipules green, linear, with 0–2 basal, lateral, smaller segments. Lamina lanceolate, crenate or subentire, sessile or indistinctly petiolated. Bracteoles minute or caducous (0–2 mm). Corolla violet to whitish with a white throat or corolla bright yellow throughout. Spur stout or saccate, longer than calycine appendages. Style clavate, unmargined, beardless. Cleistogamous flowers not produced. Allopolyploid (CHAM + MELVIO). Secondary base chromosome number *x’* = 26. *ITS* sequence of MELVIO type.

*Diagnostic characters*.—Subshrubs AND lamina lanceolate, remotely crenate, indistinctly petiolated AND base chromosome number *x* = 26.

*Ploidy and accepted chromosome counts*.—≥4*x*; 2*n* = 52 (*V. arborescens*, *V. saxifraga*).

*Age*.—Crown node age 5.6 (3.9–6.2) Ma; [28].

*Included species*.—3. *Viola arborescens* L., *V. saxifraga* Maire, *V. scorpiuroides* Coss.

*Distribution*.—Three disjunct species in the Mediterranean region (Figure 35): *Viola saxifraga* in the high Atlas, *V. arborescens* in the western Mediterranean, *V. scorpiuroides* in the southeastern Mediterranean.

*Discussion*.—Section *Xylinosium* is phylogenetically an allopolyploid CHAM + MELVIO lineage and has retained the MELVIO homoeolog for *ITS* (Figure 2). Karyologically it is characterised by the secondary base chromosome number *x’* = 26, and morphologically by the subshrubby habit in combination with the minute bracteoles (caducous in *V. arborescens* and *V. saxifraga*; 1–2 mm in *V. scorpiuroides*). The exact ploidy and genomic constitution of the section are obscured by gene loss and duplication [28]. Presumably, 2*n* = 52 reflects octoploidy. Pollen in both *V. arborescens* and *V. saxifraga* is monomorphic 3-colporate, and in *V. scorpiuroides* heteromorphic 3–4-colporate which indicates secondary polyploidy in this species [157]. Both Becker [1] and Marcussen et al. [28] included in sect. *Xylinosium* also the South African *V. decumbens*. Here, we place this last taxon in the monotypic sect. *Melvio*; see there for justification. The species of sect. *Xylinosium* have sometimes been confused with those of the vegetatively similar, allopatric sect. *Sclerosium* (e.g., [295,306]), which may explain the erroneous report of cleistogamous flowers in sect. *Xylinosium* [73].

## 3. Materials and Methods

To generate a comprehensive taxonomy for *Viola*, we first compiled a list of accepted species. Morphological and chromosome count data for these species were reconciled with phylogenetic data and used to infer monophyletic groups and define apomorphies based on which, using a set of predefined criteria, the classification was based. A downloadable and printable poster with the Viola images presented in Figure 3 is available in Appendix A. The data (scripts and analysis files) presented in this study are available in Appendix A.

### 3.1. Species Checklist

To generate a global species checklist for *Viola* we first downloaded the list of accepted species names for *Viola* from the Plants of the World Online database [307] and further revised this list of species according to our expert knowledge and based on published taxonomic treatments. This included adding numerous published names, most of which we accept as species, along with some that we consider synonymous but which are accepted by other authorities. We also included the accepted fossil species. Where relevant, protologues and type specimens were inspected. We classified taxa as either “accepted”, for the species recognised by us including entities not yet published, “hybrid” for interspecific hybrids, “included” for infraspecific taxa of an accepted species such as a subspecies or variety, “synonym” for species synonyms, “unresolved” in the rare case that rank or validity of a taxon could not be determined, and “rejected” for rejected names. Subspecies and varieties are included only in the cases that they are mentioned in the synonymy. For synonyms, instead of including all names ever published for *Viola*, most of which have been out of use for a long time, we included those names cited in the updated literature covering the distribution range of *Viola*, i.e., Africa [308], Australia [226], central Asia [79,309,310], eastern Asia [61,74,78,311], Europe [22,73,312,313,314], North America [291,315], and South America [68,80,316]. The original list downloaded from POWO comprised 761 entries; the edited list, including many new species which we individually confirm as distinct and requiring publication, comprises 1792 entries, 664 of which we accept as species.

### 3.2. Morphology Data

A wide range of morphological traits of flowering and fruiting plants were examined on herbarium specimens, including online images verified as to identity, for several to many representative and morphologically diverse species in larger infrageneric groups and for most or all species in smaller groups, where available. Protologues and recently published descriptions for many species (where identifications were confirmed) were also consulted. Of particular value in delimiting or distinguishing infrageneric groups or suggesting relationships among groups were growth form; duration, habit of rhizomes, stems, or stolons; stipule size and shape, adnation, and margins; leaf lamina features; calycine appendage size, shape, and margins; corolla throat and petal colour pattern; shape of bottom (anterior) petal and its size relative to lateral and upper petals; spur size and shape; presence or absence of beards (indument within) on lateral or bottom petals; style features; capsule dehiscence behaviour; and ability/inability to produce cleistogamous flower and whether cleistogamy is seasonal or not. Previous classifications [1,29,46,47] have highlighted style morphology as particularly important in diagnosing and comparing groups, and other studies have shown details of style morphology to be effective species-diagnostic traits [317]. We made special efforts to survey styles from numerous species across all infrageneric groups (Figure 9), from specimens and from the literature, developed a rubric for interpreting and describing particular features, developed descriptions of styles for individual species, then created summary descriptions for all groups.

### 3.3. Chromosome Number Data

Base chromosome numbers within *Viola* differ among sublineages (e.g., *x* = 6 in sect. *Chamaemelanium*, *x* = 10 in sect. *Viola*, *x* = 12 in sect. *Plagiostigma*). In order to systematise this information, we first downloaded data on chromosome counts for all species from the Chromosome Counts Database (CCDB) [318] and from primary literature sources. We then evaluated the reliability of individual counts and discarded counts that did not fit other counts on the same species or lineage in terms of ploidy and base number.

### 3.4. Criteria and Principles for an Updated Infrageneric Classification of Viola

We proposed a phylogenetic classification, based on previously published data (primarily [28,45]). Criteria for defining the formal infrageneric taxa were that they are monophyletic and/or possess apomorphies (morphological or other). Taxonomic levels and taxon names were chosen to maximise taxonomic stability and continuity. Allopolyploidy is widespread in *Viola* and its phylogeny has the topology of a network rather than a tree. Such reticulate phylogenies are not always reconcilable with a hierarchical classification. To accommodate for the conflicting situations we have chosen to accept the three infrageneric segregate taxa (e.g., sections) A, B, and X even if X is the allopolyploid of A and B. This affected sect. *Chamaemelanium* (which is diploid and possibly contributed genomes to a dozen of allotetraploid sections/lineages) and sect. *Nosphinium* (which is 10*x* and combines genomes from three other sections/lineages). In the case that an infrageneric segregate taxon (e.g., section) is known to contain internal polyploids, we have chosen to delimit it so that A, B, and X, as defined above, are monophyletic. For example, subsect. *Stolonosae* is typified with *V. palustris*, and because *V. palustris* (8*x*) is the alloploid of *V. epipsila* (4*x*) and *V. minuscula* (4*x*) [45], subsect. *Stolonosae* by definition has to comprise at least these three species.

### 3.5. Generating Distributional Maps for Viola Sections

Occurrence data for each *Viola* section was downloaded from the Global Biodiversity Information Facility (GBIF) database [319] using a custom *R* [320] script using the packages *rgbif* [321], *tidyverse* [322], and *raster* [323], and cleaned using *speciesgeocodeR* [324]. All the occurrence datasets were accessed via https://GBIF.org (accessed on 11 December 2021) and have the following DOIs: subg. *Neoandinium* https://doi.org/10.15468/dl.6a3dvh, sect. *Abyssinium* https://doi.org/10.15468/dl.utndtm, sect. *Chamaemelanium* https://doi.org/10.15468/dl.wr7kd5 and https://doi.org/10.15468/dl.fg8kk8, sect. *Chilenium* https://doi.org/10.15468/dl.5ugyp9, sect. *Danxiaviola* https://doi.org/10.15468/dl.9v545h, sect. *Delphiniopsis* https://doi.org/10.15468/dl.ct87uy, sect. *Erpetion* https://doi.org/10.15468/dl.r7tjd3, sect. *Himalayum* https://doi.org/10.15468/dl.rhqf8q, sect. *Leptidium* https://doi.org/10.15468/dl.wscdns, sect. *Melanium* https://doi.org/10.15468/dl.p6ysnh, sect. *Melvio* and sect. *Nematocaulon* https://doi.org/10.15468/dl.v5nrqx, sect. *Nosphinium* https://doi.org/10.15468/dl.stx66g and https://doi.org/10.15468/dl.fhw4xu, sect. *Plagiostigma* https://doi.org/10.15468/dl.jsftmz, sect. *Rubellium* https://doi.org/10.15468/dl.a9cpek, sect. *Sclerosium* https://doi.org/10.15468/dl.mvahfp, sect. *Spathulidium* https://doi.org/10.15468/dl.x5btdu, sect. *Tridens* https://doi.org/10.15468/dl.ufbaqp, sect. *Viola* https://doi.org/10.15468/dl.efxwyy, sect. *Xanthidium* https://doi.org/10.15468/dl.vn5t5f, and sect. *Xylinosium* https://doi.org/10.15468/dl.d48ncv. To each dataset we added further records that had not been uploaded to public databases, from, e.g., literature, herbarium specimens, and field surveys. Maps were constructed using a custom *R* [320] script using the packages *maptools* [325], *rgdal* [326], and *reader* [327]. 

### 3.6. Monoploid Phylogeny of Viola (Figure 5)

We reinterpreted the phylogenetic network of Marcussen et al. [28] based on new information, i.e., new chromosome count for *Viola banksii* (2*n* = 50 not 60 [98]), correction of chromosome count for *V. tridentata* (2*n* = 40 not 80 [99]), correction of the interpretation of homoeologs in *V. decumbens*, and sequences of new taxa (e.g., [90,231]). The species checklist is available in Appendix B.

### 3.7. ITS Phylogeny for Viola (Figure 6)

In order to obtain a phylogeny with denser taxon sampling, we downloaded sequences of the ribosomal internal transcribed spacers 1 and 2 (*ITS*) for 87 representative species from GenBank, including one outgroup, and obtained another three sequences by PCR following the protocol of Ballard et al. [2] (Table 4). Sequences were combined in cases where ITS1 and ITS2 had been sequenced separately for the same species. The resulting 90 sequences were aligned in AliView [328] and terminal gaps were coded as “?”. Indels were coded by Simple Indel Coding [329] in SeqState v1.4.1 [330]. The analysis was set up in BEAUTi v1.10.4 and analysed in BEAST v1.10.4 [331] with substitution model GTR + G for the nucleotide partition and a 1-rate + G model (equivalent to JC + G) for the indel partition, a common uncorrelated lognormal clock, a Yule tree prior. The MCMC chain was run for 20 million generations with subsampling every 10,000 generations and monitored in Tracer v1.7.1 [332] to ensure all parameters reached convergence and the recommended effective sample size of at least 200. After removal of a 10% burn-in, the maximum credibility tree was calculated in TreeAnnotator v1.10.4 [331] and visualised in FigTree [333]. Normal age priors, specified as N(µ,σ), were obtained from the appendix of Marcussen et al. [28] and applied to five crown nodes, i.e., *Viola* N(30.9,0.38), the CHAM lineage N(18.98,0.35), the MELVIO lineage N(18.71,0.34), sect. *Plagiostigma* (16.62,0.45), and sect. *Melanium* N(12.51,0.25). Section *Plagiostigma* and *Viola* subg. *Viola* were each constrained as monophyletic.

### 3.8. Historical Biogeography of Viola (Figure 8)

We reconstructed the discrete historical biogeography of *Viola* (Figure 8) using a simplified approach based on stochastic character mapping [103] of four biogeographic categories, a single-rate transition model, and 50 operational taxonomic units as defined in the diploid multilabelled phylogenetic timetree [334] that is the counterpart of the phylogenetic allopolyploid network in Figure 5. Each section of the genus was given either of four biogeographic categories (Australia, northern hemisphere, South Africa, and South America) in correspondence with the area shared by 90% of its species. Stochastic character mapping was performed with 1000 simulations using the *R* [320] package *phytools* [335]. 

### 3.9. Multigene Phylogeny for Sect. Chamaemelanium (Figure 13)

The *Chamaemelanium* phylogeny was generated based on concatenated sequences of the nuclear regions *GPI*, *NRPD2a*, and *ITS*, and the chloroplast region *trnL-trnF* (Table 5). The analysis was set up in BEAUTi v1.10.4 and analysed in BEAST v1.10.4 [331] with substitution model GTR + G for each of the nucleotide partitions, a common uncorrelated lognormal clock, and a Yule tree prior. The MCMC chain was run for 10 million generations with subsampling every 10,000 generations and monitored in Tracer v1.7.1 [332] to ensure all parameters reached convergence and the recommended effective sample size of at least 200. After removal of a 10% burn-in, the maximum credibility tree was calculated in TreeAnnotator v1.10.4 [331] and visualised in FigTree [333]. The ingroup (sect. *Chamaemelanium*) was constrained as monophyletic.

### 3.10. Historical Biogeography and Age of the Hawaiian Violets, Subsect. Nosphinium (Figure 25)

The historical biogeography and age of subsect. *Nosphinium* (Figure 25) was estimated by simultaneous analysis of ITS sequence data, island biogeography, and node dating. Available sequences of the Hawaiian taxa and outgroups (Table 6) were downloaded from GenBank and aligned in AliView [328]. The dating analysis was set up in BEAUTi v1.10.4 and analysed in BEAST v1.10.4 [331] with substitution model GTR + G for the nucleotide partition with useAmbiguities set to “true”, an uncorrelated lognormal clock, and a Yule tree prior. Biogeography (i.e., island), obtained from the original publications [81,85,336,337], was added as a discrete trait and analysed under a symmetrical model and a strict clock; the biogeography of outgroup taxa was scored as missing (“?”). The MCMC chain was run for 100 million generations with subsampling every 10,000 generations and monitored in Tracer v1.7.1 [332] to ensure all parameters reached convergence and the recommended effective sample size of at least 200. After removal of a 10% burn-in, the maximum credibility tree was calculated in TreeAnnotator v1.10.4 [331] and visualised in FigTree [333]. A normal age prior, N(8.44,0.34) Ma, obtained from the appendix of Marcussen et al. [28], was applied to the crown node of sect. *Nosphinium*. Subgenus *Viola* and sect. *Nosphinium* were each constrained as monophyletic. 

### 3.11. Multigene Phylogeny for Sect. Plagiostigma and Sect. Viola (Figure 27 and Figure 33)

The multigene phylogeny for sect. *Plagiostigma* and sect. *Viola* (Figure 27 and Figure 33) was generated based on concatenated sequences of the eight nuclear regions *GPI-C* (CHAM homoeolog), *GPI-M* (MELVIO homoeolog), *NRPD2a-C* (CHAM homoeolog), *NRPD2a-M* (MELVIO homoeolog), *ITS-C* (CHAM homoeolog), *ITS-M* (MELVIO homoeolog), *SDH-C* (CHAM homoeolog), and *SDH-M* (MELVIO homoeolog; Table 7). Sequence reads for the genomic sequences (*V. acuminata*, *V. dissecta*, *V. grypoceras*, *V. raddeana*) were obtained by BLAST searching the NCBI Sequence Reads Archive (SRA) database and were assembled in BioEdit [338] by CAP alignment and in AliView [328]. The phylogenetic analysis was set up in BEAUTi v1.10.4 and analysed in BEAST v1.10.4 [331] with substitution model HKY + G for each of the four nucleotide partitions (i.e., *GPI*, *ITS*, *NRPD2a*, *SDH*), a common uncorrelated lognormal clock, and a Yule tree prior. Two MCMC chains were run for a total of 150 million generations with subsampling every 10,000 generations and monitored in Tracer v1.7.1 [332] to ensure all parameters reached convergence and the recommended effective sample size of at least 200. After removal of a 1% burn-in of each chain, the chains were merged with LogCombiner v.1.10.4 [331] and the maximum credibility tree was calculated in TreeAnnotator v1.10.4 [331] and visualised in FigTree [333]. The ingroup (sect. *Plagiostigma* + sect. *Viola*) was constrained as monophyletic.

## Figures and Tables

**Figure 1 plants-11-02224-f001:**
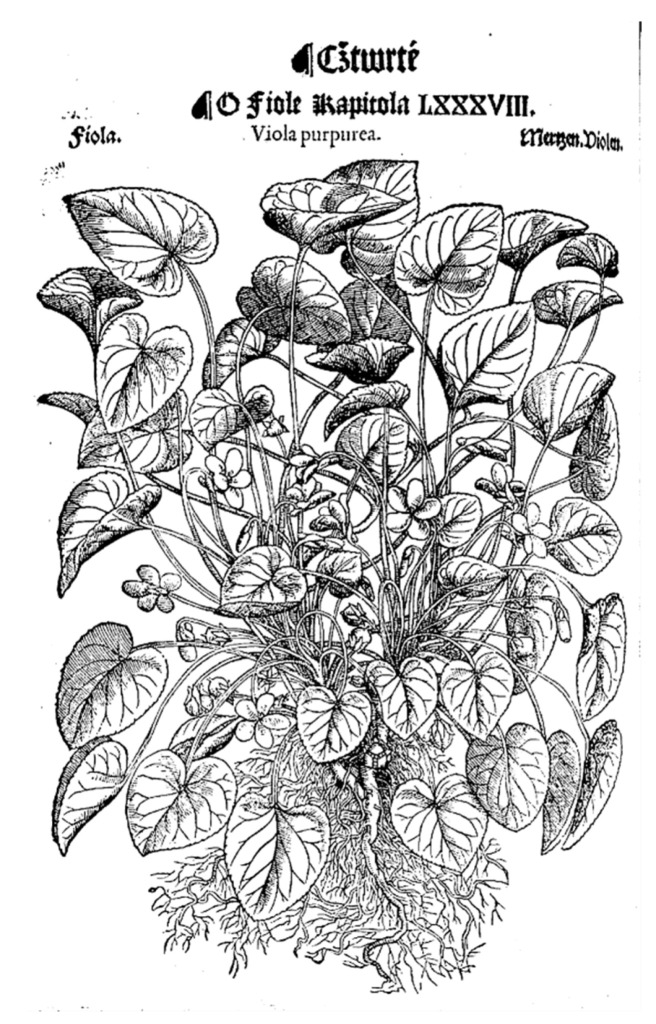
Illustration of *Viola odorata* in the herbal of Matthiolus, printed in 1562 [8]. At least partly, the foliage appears to represent the common hybrid *V. hirta* × *odorata*. In the accompanying text (fol. CCCLIIII) the rooting stolons of *V. odorata* are described and compared to those of *Fragaria* and *Pilosella officinarum*.

**Figure 2 plants-11-02224-f002:**
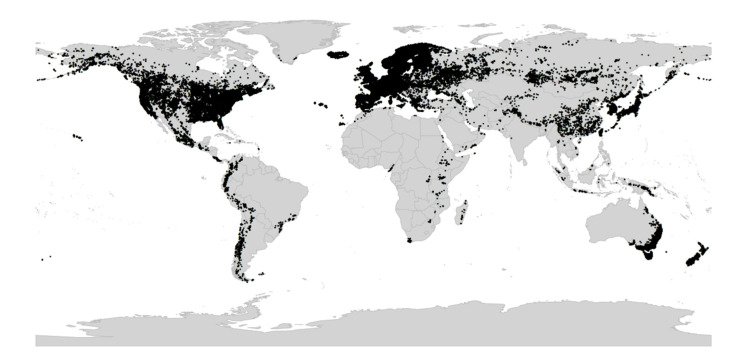
Global distribution of *Viola* L. (Violaceae), showing the predominantly temperate distribution of the genus.

**Figure 3 plants-11-02224-f003:**
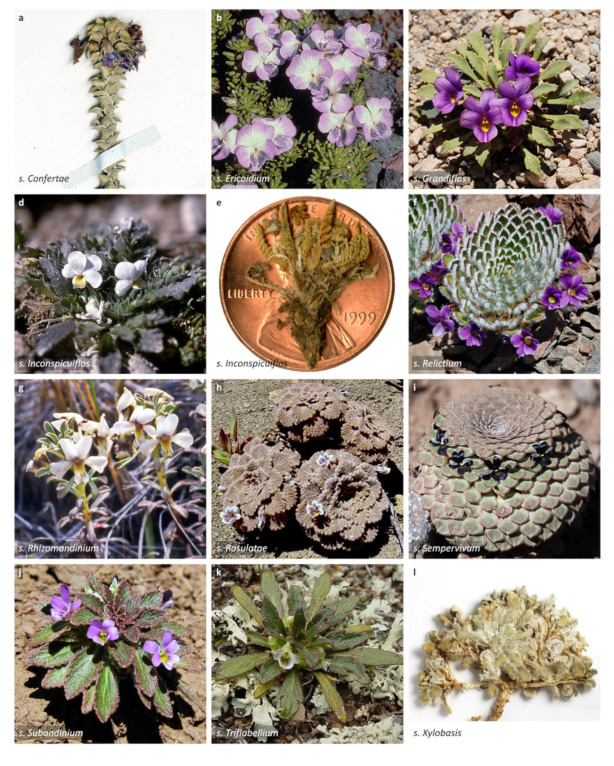
Photos of representative members of the sections and subsections of *Viola*.—(**a**–**l**) Subg. *Neoandinium*.—(**m**–**av**) Subg. *Viola*.—(**a**) **Sect. *Confertae***: *V. nassauvioides*.—(**b**) **Sect. *Ericoidium***: *V. fluehmannii* (photo © John M. Watson).—(**c**) **Sect. *Grandiflos***: *V. cheeseana* (photo © John M. Watson).—(**d**) **Sect. *Inconspicuiflos***: *V. weibelii* (photo © John M. Watson).—(**e**) **Sect. *Inconspicuiflos***: *V. lilliputana* (photo © Harvey E. Ballard).—(**f**) **Sect. *Relictium***: *V. dandoisiorum* (photo © John M. Watson).—(**g**) **Sect. *Rhizomandinium***: *V. escondidaensis* (photo © John M. Watson).—(**h**) **Sect. *Rosulatae***: *V. volcanica* (photo © Ana R. Flores).—(**i**) **Sect. *Sempervivum***: *V. atropurpurea* (photo © John M. Watson).—(**j**) **Sect. *Subandinium***: *V. subandina* (photo © Ana R. Flores).—(**k**) **Sect. *Triflabellium***: *V. triflabellata* (photo © Ana R. Flores).—(**l**) **Sect. *Xylobasis***: *V. beati* (photo © Ana R. Flores).—(**m**) **Sect. *Abyssinium***: *V. abyssinica* (photo © Robert v. Blittersdorff).—(**n**) **Sect. *Chamaemelanium***: *V. pubescens* (photo © Kim Blaxland).—(**o**) **Sect. *Chilenium***: *V. maculata* (photo © Instituto de Botánica Darwinion).—(**p**) **Sect. *Danxiaviola***: *V. hybanthoides* (photo © Qiang Fan).—(**q**) **Sect. *Delphiniopsis***: *V. cazorlensis* (photo © Santiago Martín-Bravo).—(**r**) **Sect. *Erpetion***: *V. banksii* (photo © Dewald du Plessis).—(**s**) **Sect. *Himalayum***: *V. kunawurensis* (photo © Vladimir Epiktetov).—(**t**) **Sect. *Leptidium***: *V. boliviana* (photo © Sam Wilson).—(**u**) **Sect. *Melanium*, subsect. *Bracteolatae***: *V. tricolor* and *V. arvensis* (photo © Thomas Marcussen).—(**v**) **Sect. *Melanium*, subsect. *Cleistogamae***: *V. rafinesquei* (photo © Mary Ann Yaich).—(**w**) **Sect. *Melanium*, subsect. *Dispares***: *V. dyris* (photo © Alan Keohane).—(**x**) **Sect. *Melanium*, subsect. *Ebracteatae***: *V. parvula* (photo © Albert Keshet).—(**y**) **Sect. *Melanium*, subsect. *Pseudorupestres***: *V. nummulariifolia* (photo © Sylvain Piry).—(**z**) **Sect. *Melvio***: *V. decumbens* (photo © Magriet Brink).—(**aa**) **Sect. *Nematocaulon***: *V. filicaulis* (photo © Andrew Townsend).—(**ab**)**Sect. *Nosphinium*, subsect. *Borealiamericanae***: *V. fimbriatula* (photo © Kim Blaxland).—(**ac**) **Sect. *Nosphinium*, subsect. *Clausenianae***: *V. clauseniana* (photo © Thomas Marcussen).—(**ad**) **Sect. *Nosphinium*, subsect. *Langsdorffianae***: *V. langsdorffii* (photo © Jonathan Goff).—(**ae**) **Sect. *Nosphinium*, subsect. *Mexicanae***: *V. nannei* (photo © Neptalí Ramírez Marcial).—(**af**) **Sect. *Nosphinium*, subsect. *Nosphinium***: *V. maviensis* (photo © Karl Magnacca).—(**ag**) **Sect. *Nosphinium*, subsect. *Pedatae***: *V. pedata* (photo © Kim Blaxland).—(**ah**) **Sect. *Plagiostigma*, subsect. *Australasiaticae***: *V. sumatrana* (photo © Mário Duchoň).—(**ai**) **Sect. *Plagiostigma*, subsect. *Bilobatae***: *V. hamiltoniana* (photo © Toshihiro Nagata).—(**aj**) **Sect. *Plagiostigma*, subsect. *Bulbosae***: *V. tuberifera* (photo © Masashi Igari).—(**ak**) **Sect. *Plagiostigma*, subsect. *Diffusae***: *V. huizhouensis* (photo © Yanshuang Huang).—(**al**) **Sect. *Plagiostigma*, subsect. *Formosanae***: *V. formosana* (photo © Kuan-Chieh (Chuck) Hung).—(**am**) **Sect. *Plagiostigma*, subsect. *Patellares***: *V. tokubuchiana* (photo © Masashi Igari).—(**an**) **Sect. *Plagiostigma*, subsect. *Stolonosae***: *V. pluviae* (photo © Kim Blaxland).—(**ao**) **Sect. *Rubellium***: *V. rubella* (photo © Pablo Silva).—(**ap**) **Sect. *Sclerosium***: *V. cinerea* (photo © Jerome Viard).—(**aq**) **Sect. *Spathulidium***: *V. pachyrrhiza* (photo © Dieter Zschummel).—(**ar**) **Sect. *Tridens***: *V. tridentata* (photo © larsonek).—(**as**) **Sect. *Viola*, subsect. *Rostratae***: *V. canina* (photo © Thomas Marcussen).—(**at**) **Sect. *Viola*, subsect. *Viola***: *V. collina* (photo © Thomas Marcussen).—(**au**) **Sect. *Xanthidium***: *V. flavicans* (photo © Instituto de Botánica Darwinion).—(**av**) **Sect. *Xylinosium***: *V. arborescens* (photo © Abdelmonaim Homrani Bakali).—All images used in this figure were cropped and gamma-corrected.—Links to the online sources for the images used in this figure under a creative commons (CC) licence: *V. abyssinica*, http://www.westafricanplants.senckenberg.de/root/index.php?page_id=47&id=2203#image=22051, (accessed on 5 May 2022) © Robert v. Blittersdorff; *V. arborescens*, https://www.teline.fr/en/photos/violaceae/viola-arborescens#photo-7 (accessed on 11 May 2022), © Abdelmonaim Homrani Bakali, CC BY-NC 4.0; *V. nummulariifolia*, https://www.inaturalist.org/observations/89240299 (accessed on 5 May 2022), © Sylvain Piry, CC BY-NC 4.0; *V. banksii*, https://inaturalist.nz/observations/25048808 (accessed on 12 May 2022), © Dewald du Plessis, CC BY-NC 4.0; *V. boliviana*, https://ecuador.inaturalist.org/photos/57308380 (accessed on 5 May 2022), © Sam Wilson, CC BY-NC; *V. cazorlensis*, https://www.inaturalist.org/observations/109331467 (accessed on 5 May 2022), © Santiago Martín-Bravo, CC BY-NC 4.0; *V. cinerea*, https://www.inaturalist.org/observations/65108391 (accessed on 5 May 2022), © Jerome Viard, CC BY-NC 4.0; *V. decumbens*, https://www.inaturalist.org/observations/10807224 (accessed on 5 May 2022), © Magriet Brink, CC BY-NC 4.0; *V. filicaulis*, https://www.inaturalist.org/observations/67950856 (accessed on 5 May 2022), © Andrew Townsend, CC BY-NC; *V. formosana*, https://www.inaturalist.org/observations/25305342 (accessed on 5 May 2022), © Kuan-Chieh (Chuck) Hung, CC BY-NC 4.0; *V. hamiltoniana*, https://www.inaturalist.org/observations/74528783 (accessed on 5 May 2022), © Toshihiro Nagata, CC BY-NC 4.0; *V. langsdorffii*, https://www.inaturalist.org/observations/73470330 (accessed on 5 May 2022), © Jonathan Goff, CC BY-NC-SA 4.0; *V. maviensis*, https://www.inaturalist.org/observations/39272209 (accessed on 7 May 2022), © Karl Magnacca, CC BY-NC 4.0; *V. nannei*, https://www.inaturalist.org/observations/6527547 (accessed on 5 May 2022), © Neptalí Ramírez Marcial, CC BY-NC; *V. rafinesquei*, https://www.inaturalist.org/observations/111043498 (accessed on 5 May 2022), © Mary Ann Yaich, CC BY-NC 4.0; *V. rubella*, https://www.inaturalist.org/observations/93725094 (accessed on 5 May 2022), © Pablo Silva, CC BY-NC 4.0; and *V. tridentata*, https://www.inaturalist.org/observations/103467450 (accessed on 5 May 2022), © larsonek, CC BY-NC 4.0.—Links to the relevant CC licences: CC BY 4.0 (https://creativecommons.org/licenses/by/4.0/; accessed on 5 May 2022), CC-BY-NC and CC BY-NC 4.0 (https://creativecommons.org/licenses/by-nc/4.0/; accessed on 5 May 2022), CC BY-NC-SA 4.0 (https://creativecommons.org/licenses/by-nc-sa/4.0/; accessed on 5 May 2022).

**Figure 4 plants-11-02224-f004:**
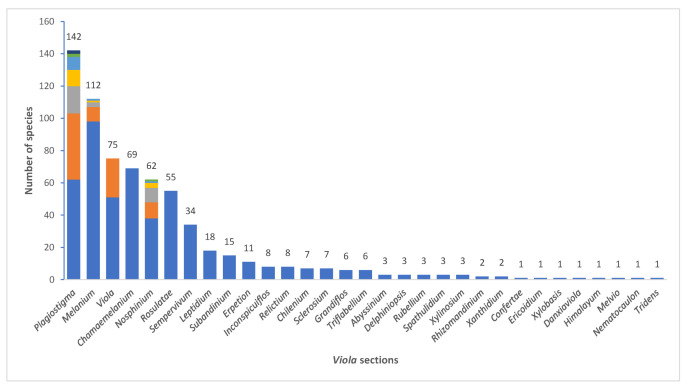
Stacked bar plot showing species richness among the 31 sections of *Viola*. Species counts are indicated above each bar. Sections containing subsections are indicated as stacked bars with the distribution of species among subsections indicated in different colours. For details on each subsection, see Table 1.

**Figure 5 plants-11-02224-f005:**
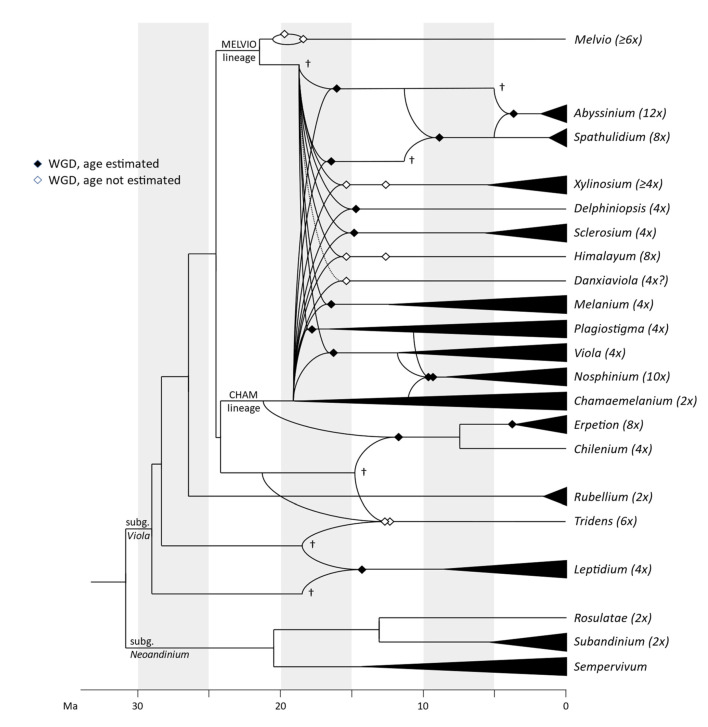
Dated phylogeny for monophyletic sections of *Viola*, updated from Marcussen et al. [28]. Estimated base ploidy is indicated after the name of each *Viola* section. Curved lines indicate parental lineages of an allopolyploid lineage and filled diamonds indicate the estimated time of allopolyploidisation [28]. Daggers denote parental lineages that are extinct or unsampled and extant only as a subgenome of an allopolyploid lineage. Horizontal black triangles indicate the estimated crown node of a section (and is prone to increase as more taxa are added). No phylogenetic data exist for sect. *Nematocaulon* and sect. *Xanthidium* of subg. *Viola* and for another seven sections of subg. *Neoandinium*.

**Figure 6 plants-11-02224-f006:**
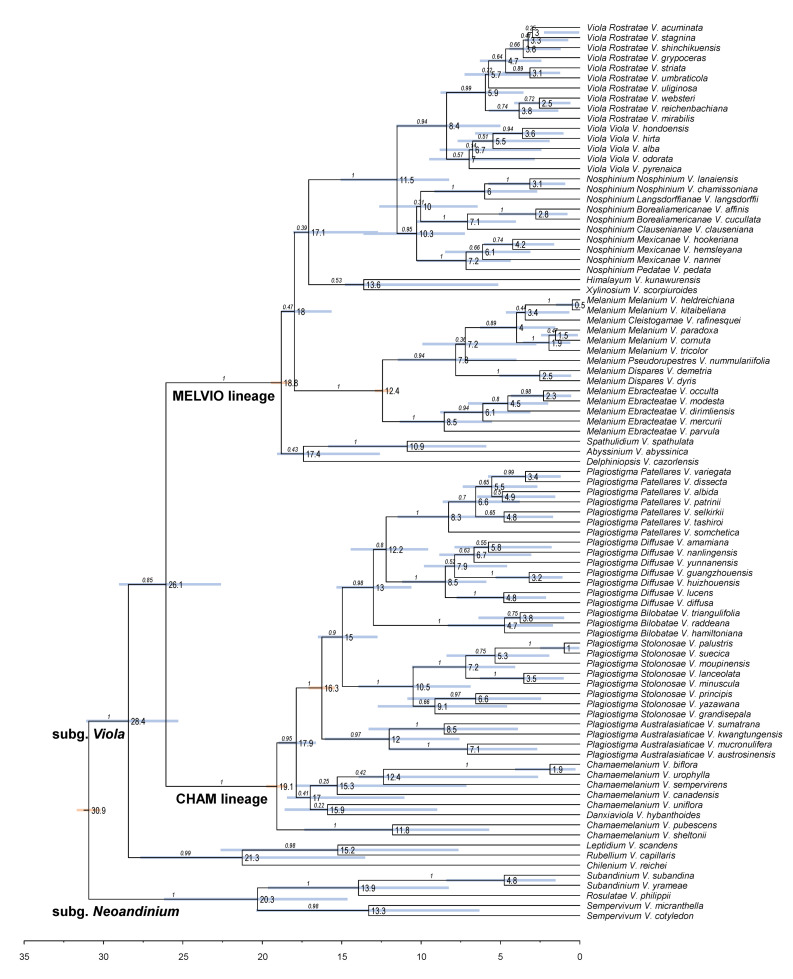
Dated BEAST phylogeny of the Internal Transcribed Spacers (ITS) 1 and 2 in selected taxa of *Viola* and secondary age calibration of five internal nodes. Tip names are shown as section, subsection (if available), and species. Node bars indicate the posterior 95% credibility interval for node height; nodes with bars indicated in brown were used for age calibration. Mean ages are indicated on nodes. The outgroup (*Melicytus*) was pruned.

**Figure 7 plants-11-02224-f007:**
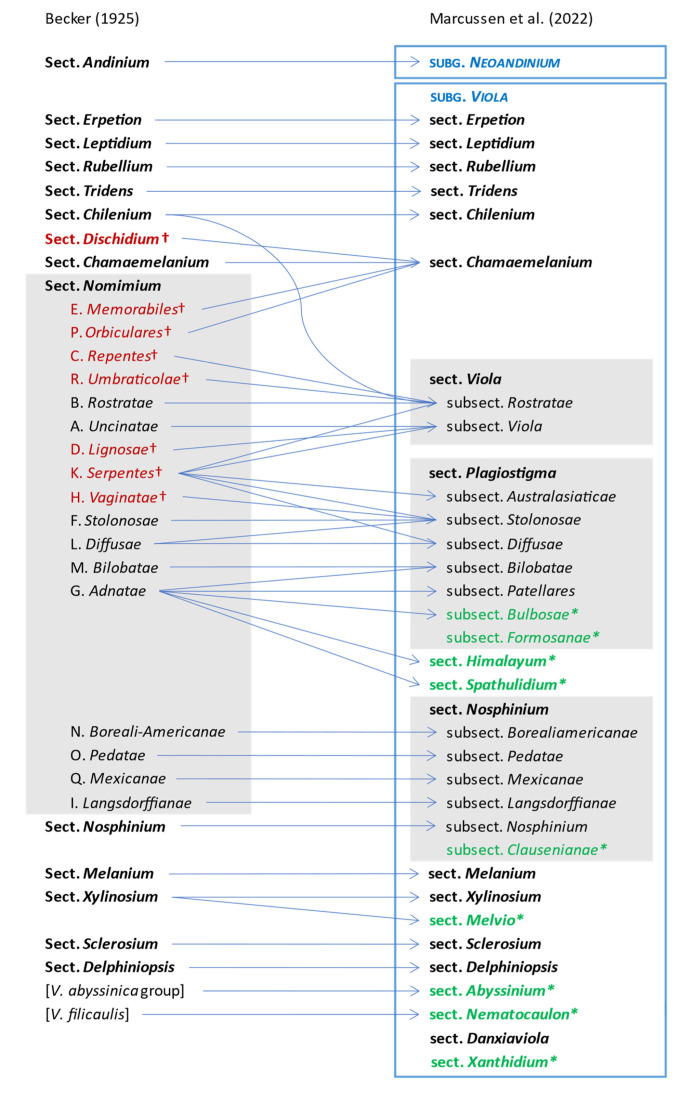
Wire diagram illustrating the major taxonomic differences between the phylogenetic classification of *Viola* proposed here (“Marcussen et al. 2022”) compared to the morphological classification proposed by Becker (1925 [1]). Merging lines denote lumping of two or more of Becker’s infrageneric groups into one taxon, while splitting lines denote segregation into two or more taxa. Taxa placed on the same line and interlinked with a horizontal line are synonymous, but may differ in delimitation, rank, or name (for reasons of priority). Taxa indicated with a dagger have been reduced to synonymy under another taxon. Taxa indicated with an asterisk are new infrageneric segregates described here. We do not show infrasectional taxa for sect. *Chamaemelanium*, which are not accepted here, or for sect. *Melanium* and the sect. *Andinium*/subg. *Neoandinium* pair, for which our treatments differ substantially from that by Becker.

**Figure 8 plants-11-02224-f008:**
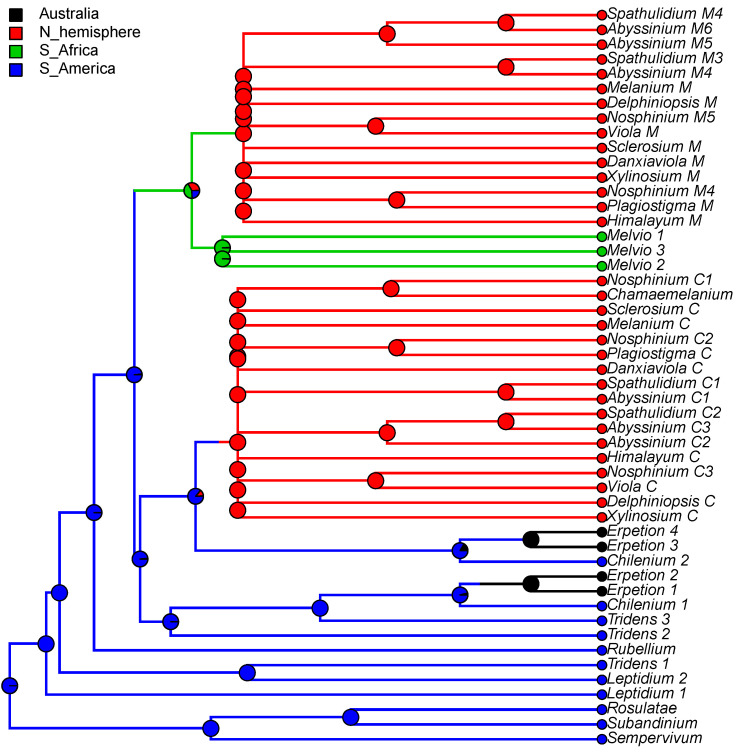
Discrete historical biogeography of *Viola* sections, showing the South American origin of the genus and independent dispersal into the northern hemisphere by the CHAM and MELVIO lineages. Ancestral states were inferred by stochastic character mapping [103] using a 1-rate model and 1000 replicates, given the monoploid multilabelled timetree with 50 leaves that results from unfolding the network in Figure 5 to a tree. Sections *Nematocaulon* (New Zealand) and *Xanthidium* (South America), for which data are lacking, are not included.

**Figure 9 plants-11-02224-f009:**
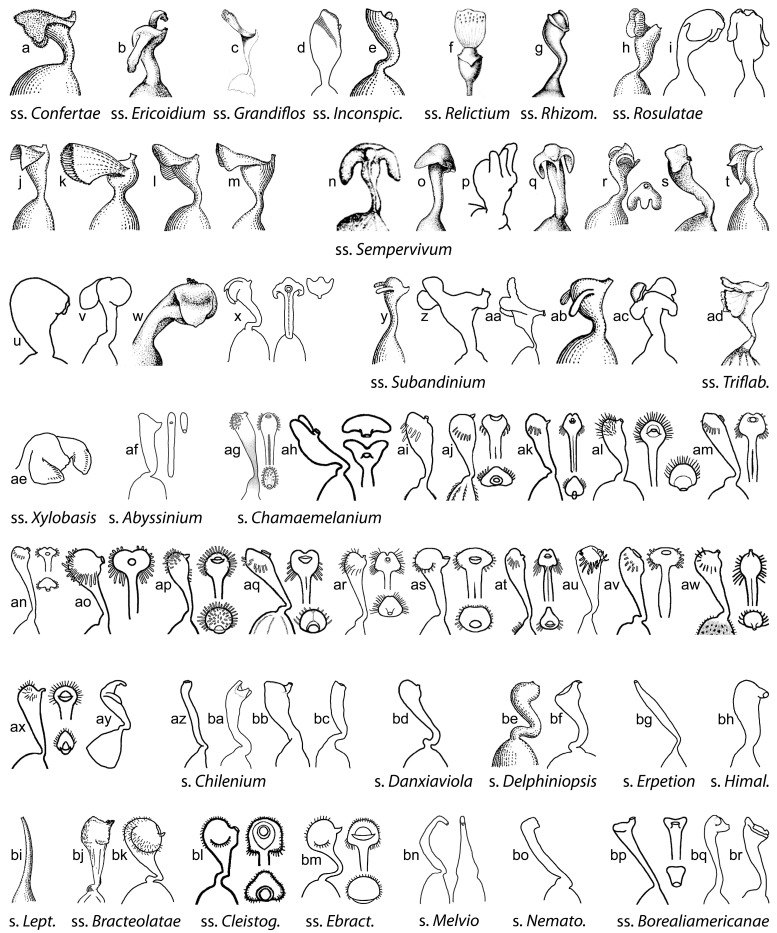
Style shapes in *Viola*. (**a**–**ae**) Subg. *Neoandinium*. (**af**–**fe**) Subg. *Viola*.—**Sect. *Confertae***: (**a**) *V. nassauvioides* [1].—**Sect. *Ericoidium***: (**b**) *V. fluehmannii* [80].—**Sect. *Grandiflos***: (**c**) *V. acanthophylla* [122].—**Sect. *Inconspicuiflos***: (**d**) *V. lilliputana*, (**e**) *V. membranacea* [1].—**Sect. *Relictium***: (**f**) *V. ovalleana* [48].—**Sect. *Rhizomandinium***: (**g**) *V. escondidaensis* [80].—**Sect. *Rosulatae***: (**h**) *V. aurantiaca* [1], (**i**) *V. kermesina*, (**j**) *V. niederleinii* [1], (**k**) *V. replicata* [1], (**l**) *V. rugosa* [1], (**m**) *V. volcanica* [1].—**Sect. *Sempervivum***: (**n**) *V. atropurpurea* [80], (**o**) *V. auricolor* [80], (**p**) *V. bangii*, (**q**) *V. coronifera* [80], (**r**) *V. cotyledon* [1], (**s**) *V. dasyphylla* [80], (**t**) *V. hieronymi* [80], (**u**) *V. micranthella*, (**v**) *V. pygmaea*, (**w**) *V. sacculus* [80], (**x**) *V. sempervivum.*—**Sect. *Subandinium***: (**y**) *V. araucaniae* [1], (**z**) *V. polypoda*, (**aa**) *V. pusilla*, (**ab**) *V. subandina* [80], (**ac**) *V. weberbaueri.*—**Sect. *Triflabellium***: (**ad**) *V. triflabellata* [80].—**Sect. *Xylobasis***: (**ae**) *V. beati*, tip of style [123].—**Sect. *Abyssinium***: (**af**) *V. abyssinica.*—**Sect. *Chamaemelanium***: (**ag**) *V. beckwithii*, (**ah**) *V. biflora*, (**ai**) *V. canadensis*, (**aj**) *V. charlestonensis*, (**ak**) *V. cuneata*, (**al**) *V. douglasii*, (**am**) *V. flettii*, (**an**) *V. frank-smithii*, (**ao**) *V. guadalupensis*, (**ap**) *V. hallii*, (**aq**) *V. lithion*, (**ar**) *V. lobata*, (**as**) *V. nuttallii*, (**at**) *V. ocellata*, (**au**) *V. orientalis* (redrawn from [78]), (**av**) *V. scopulorum*, (**aw**) *V. sheltonii*, (**ax**) *V. trinervata*, (**ay**) *V. wallichiana* (redrawn from [78]).—**Sect. *Chilenium***: (**az**) *V. commersonii* [80], (**ba**) *V. maculata*, (**bb**) *V. reichei*, (**bc**) *V. stuebelii.*—**Sect. *Danxiaviola***: (**bd**) *V. hybanthoides* (redrawn from [90]).—**Sect. *Delphiniopsis***: (**be**) *V. cazorlensis* [1], (**bf**) *V. delphinantha.*—**Sect. Erpetion**: (**bg**) *V. banksii.*—**Sect. *Himalayum***: (**bh**) *V. kunawurensis* (redrawn from [124]).—**Sect. *Leptidium***: (**bi**) *V. stipularis* [1].—**Sect. *Melanium*, subsect. *Bracteolatae***: (**bj**) *V. cornuta* [29], (**bk**) *V. tricolor* (redrawn from [125]).—**Sect. *Melanium*, subsect. *Cleistogamae***: (**bl**) *V. rafinesquei.*—**Sect. *Melanium*, subsect. *Ebracteolatae***: (**bm**) *V. dirimliensis.*—**Sect. *Melvio***: (**bn**) *V. decumbens.*—**Sect. *Nematocaulon***: (**bo**) *V. filicaulis.*—**Sect. *Nosphinium*, subsect. *Borealiamericanae***: (**bp**) *V. brittoniana*, (**bq**) *V. cucullata*, (**br**) *V. palmata*, (**bs**) *V. pedatifida*, (**bt**) *V. pratincola*, (**bu**) *V. sagittata*, (**bv**) *V. septemloba*, (**bw**) *V. sororia*, (**bx**) *V. viarum*, (**by**) *V. villosa.*—**Sect. *Nosphinium*, subsect. *Clausenianae***: (**bz**) *V. clauseniana.*—**Sect. *Nosphinium*, subsect. *Langsdorffianae***: (**ca**) *V. howellii*, (**cb**) *V. langsdorffii.*—**Sect. *Nosphinium*, subsect. *Mexicanae***: (**cc**) *V. grahamii*, (**cd**) *V. guatemalensis*, (**ce**) *V. hookeriana*, (**cf**) *V. humilis*, (**cg**) *V. nannei*, (**ch**) *V. nubicola*, (**ci**) *V. oxyodontis.*—**Sect. *Nosphinium*, subsect. *Nosphinium***: (**cj**) *V. kauaensis*, (**ck**) *V. maviensis* [1].—**Sect. *Nosphinium*, subsect. *Pedatae***: (**cl**) *V. pedata.*—**Sect. *Plagiostigma*, subsect. *Australasiaticae***: (**cm**) *V. austrosinensis* (redrawn from [126]), (**cn**) *V. annamensis*, (**co**) *V. kwangtungensis* (redrawn from [78]), (**cp**) *V. sumatrana* (redrawn from [74]).—**Sect. *Plagiostigma*, subsect. *Bilobatae***: (**cq**) *V. arcuata* (redrawn from [75]), (**cs**) *V. merrilliana* (redrawn from [74]), (**ct**) *V. triangulifolia* (redrawn from [78]), (**cr**) *V. hamiltoniana* [29].—**Sect. *Plagiostigma*, subsect. *Bulbosae***: (**cu**) *V. bulbosa* (redrawn from [76]), (**cv**) *V. tuberifera* (redrawn from [76]).—**Sect. *Plagiostigma*, subsect. *Diffusae***: (**cw**) *V. diffusa* [29], (**cx**) *V. huizhouensis* (redrawn from [127]), (**cy**) *V. jinggangshanensis* (redrawn from [128]), (**cz**) *V. nagasawae* (redrawn from [75]).—**Sect. *Plagiostigma*, subsect. *Formosanae***: (**da**) *V. formosana* (redrawn from [75]), (**db**) *V. stolonifolora* (redrawn from [97]).—**Sect. *Plagiostigma*, subsect. *Patellares***: (**dc**) *V. dactyloides* [1], (**dd**) *V. japonica* [29], (**de**) *V. macroceras* [1], (**df**) *V. patrinii* [1], (**dg**) *V. pinnata* [29], (**dh**) *V. selkirkii.*—**Sect. *Plagiostigma*, subsect. *Stolonosae***: (**di**) *V. suecica*, (**dj**) *V. blanda*, (**dk**) *V. brevipes*, (**dl**) *V. davidii* (redrawn from [78]), (**dm**) *V. fargesii* (redrawn from [78]), (**dn**) *V. grandisepala* (redrawn from [78]), (**do**) *V. jalapaensis*, (**dp**) *V. lanceolata*, (**dq**) *V. macloskeyi*, (**dr**) *V. minuscula*, (**ds**) *V. occidentalis*, (**dt**) *V. palustris*, (**du**) *V. pluviae*, (**dv**) *V. primulifolia*, (**dw**) *V. renifolia*, (**dx**) *V. rossii* (redrawn from [76]).—**Sect. *Rubellium***: (**dy**) *V. capillaris* [1], (**dz**) *V. portalesia* [1], (**ea**) *V. rubella.*—**Sect. *Sclerosium***: (**eb**) *V. stocksii*, (**ec**) *V. etbaica* [1], (**ed**) *V. somalensis* [1].—**Sect. *Spathulidium***: (**ee**) *V. spathulata.*—**Sect. *Tridens***: (**ef**) *V. tridentata* [1].—**Sect. *Viola*, subsect. *Rostratae***: (**eg**) *V. acuminata* (redrawn from [76]), (**eh**) *V. adunca*, (**ei**) *V. appalachiensis*, (**ej**) *V. canina*, (**ek**) *V. elatior* [29], (**el**) *V. huidobrii* [80], (**em**) *V. jordanii*, (**en**) *V. labradorica*, (**eo**) *V. papuana* (redrawn from [74]), (**ep**) *V. rostrata*, (**eq**) *V. rupestris* [29], (**er**) *V. stagnina* [29], (**es**) *V. striata*, (**et**) *V. uliginosa* [29], (**eu**) *V. umbraticola*, (**ev**) *V. walteri.*—**Sect.**
***Viola*, subsect. *Viola***: (**ew**) *V. barhalensis* (redrawn from [129]), (**ex**) *V. chelmea* [1], (**ey**) *V. hirta* [29], (**ez**) *V. odorata*, (**fa**) *V. pilosa* [1].—**Sect. *Xanthidium***: (**fb**) *V. flavicans*, (**fc**) *V. pallascaensis.*—**Sect. *Xylinosium***: (**fd**) *V. arborescens* [1], (**fe**) *V. scorpiuroides*.—All drawings by Kim Blaxland, H.E.B, and T.M. except where indicated.

**Figure 10 plants-11-02224-f010:**
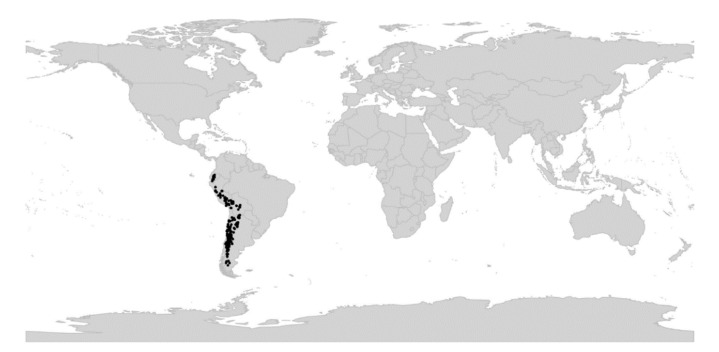
Global distribution of *Viola* subg. *Neoandinium*.

**Figure 11 plants-11-02224-f011:**
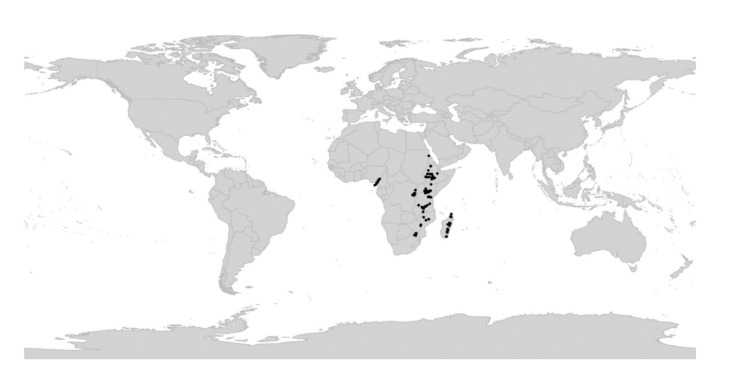
Global distribution of *Viola* sect. *Abyssinium*.

**Figure 12 plants-11-02224-f012:**
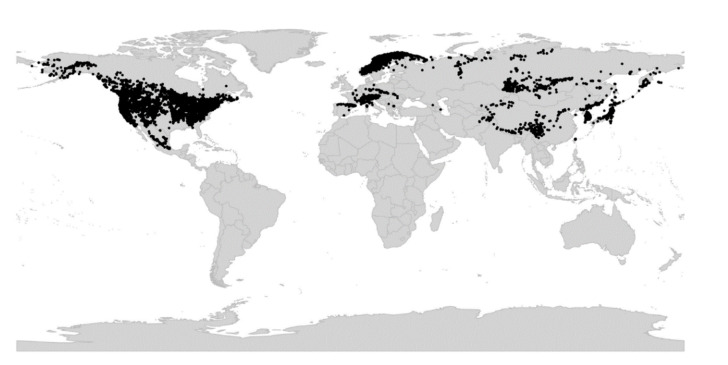
Global distribution of *Viola* sect. *Chamaemelanium*.

**Figure 13 plants-11-02224-f013:**
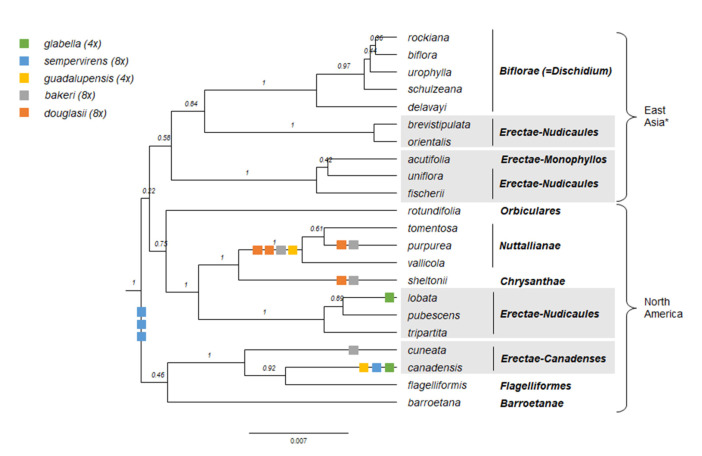
Ultrametric phylogeny of diploid taxa of *Viola* sect. *Chamaemelanium*, showing the basal irresolution among otherwise well-supported infrasectional lineages and the non-monophyly of Becker’s greges at both the diploid and allopolyploid level. Outgroups have been trimmed. The analysis was performed on a concatenated matrix of four loci (*GPI*, *ITS*, *NRPD2a*, *trnL-trnF*). The squares indicated on branches show the approximate phylogenetic placement of homoeologs of five North American allopolyploids [60], *V. bakeri* (8*x*), *V. douglasii* (8*x*), *V. glabella* (4*x*), and *V. guadalupensis* (4*x*). Branch support is given as posterior probabilities. * *Viola biflora* has a circumboreal distribution.

**Figure 14 plants-11-02224-f014:**
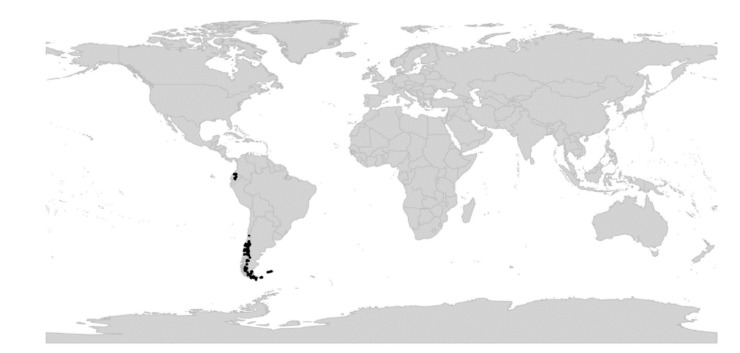
Global distribution of *Viola* sect. *Chilenium*.

**Figure 15 plants-11-02224-f015:**
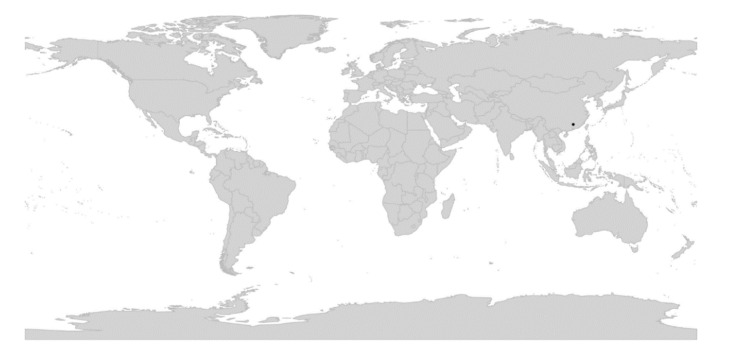
Global distribution of *Viola* sect. *Danxiaviola*.

**Figure 16 plants-11-02224-f016:**
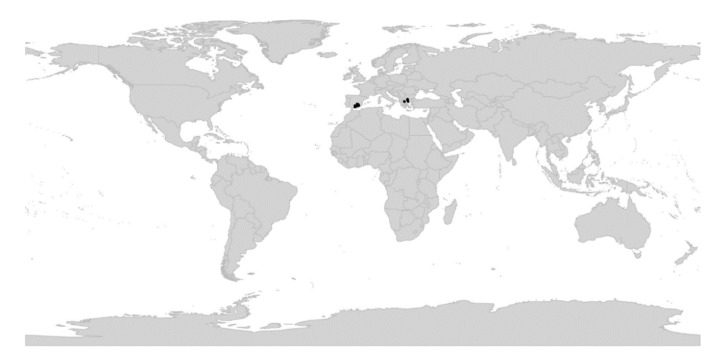
Global distribution of *Viola* sect. *Delphiniopsis*.

**Figure 17 plants-11-02224-f017:**
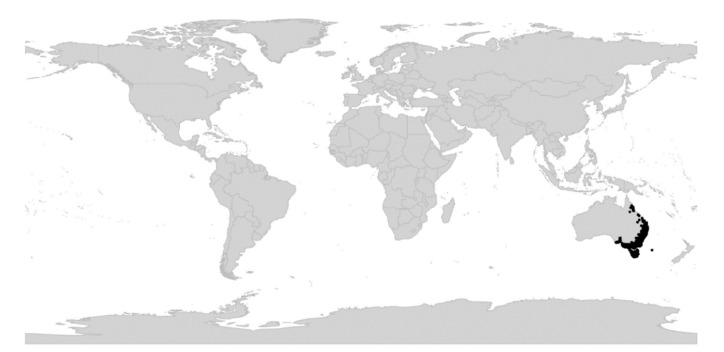
Global distribution of *Viola* sect. *Erpetion*.

**Figure 18 plants-11-02224-f018:**
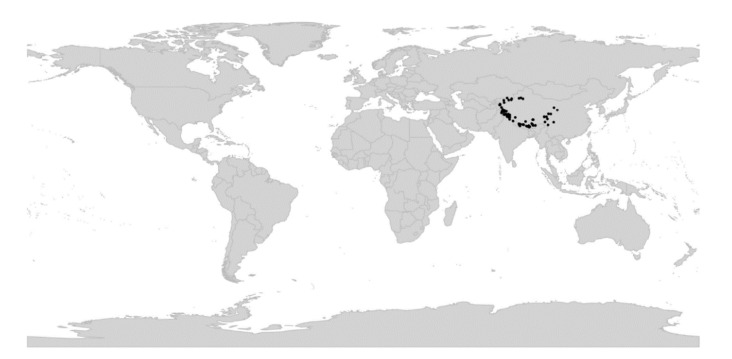
Global distribution of *Viola* sect. *Himalayum*.

**Figure 19 plants-11-02224-f019:**
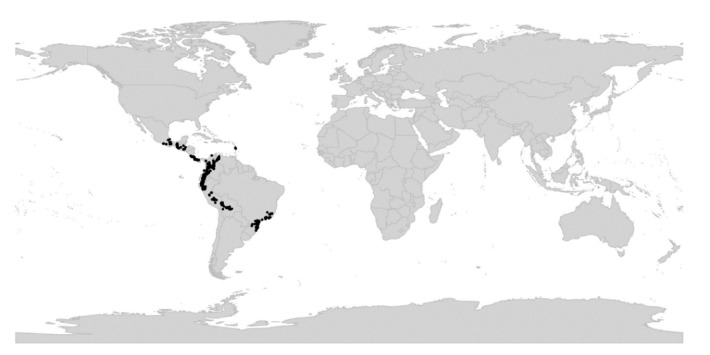
Global distribution of *Viola* sect. *Leptidium*.

**Figure 20 plants-11-02224-f020:**
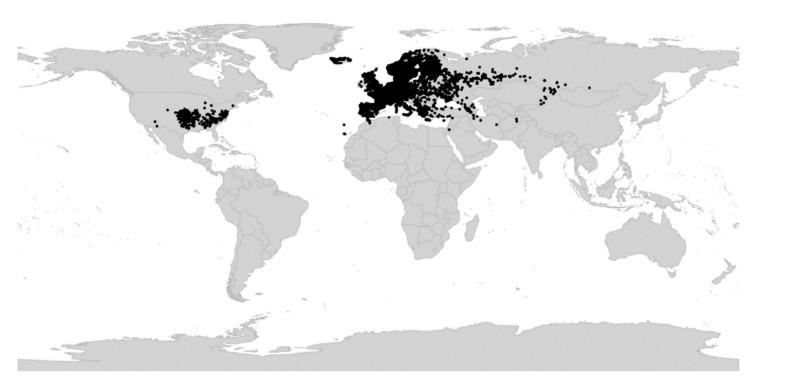
Global distribution of *Viola* sect. *Melanium*.

**Figure 21 plants-11-02224-f021:**
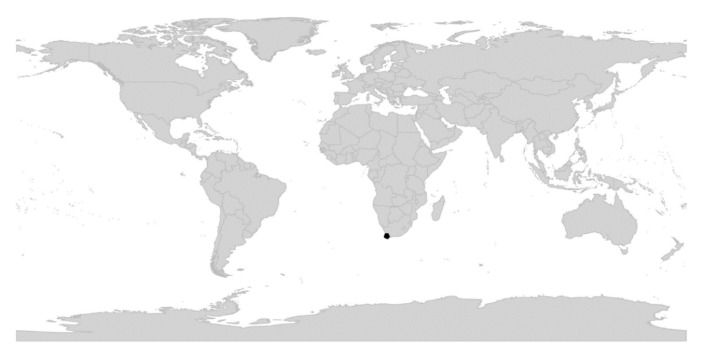
Global distribution of *Viola* sect. *Melvio*.

**Figure 22 plants-11-02224-f022:**
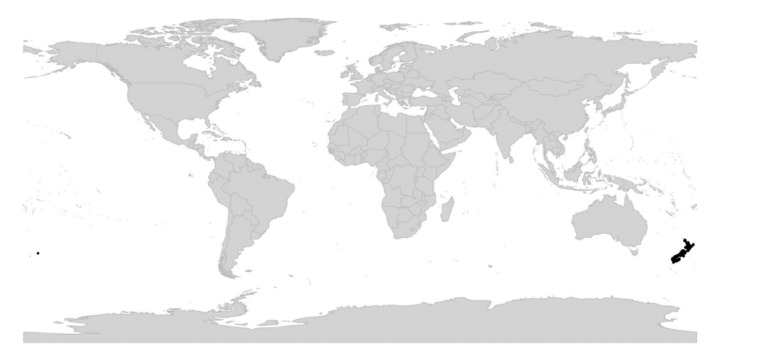
Global distribution of *Viola* sect. *Nematocaulon*.

**Figure 23 plants-11-02224-f023:**
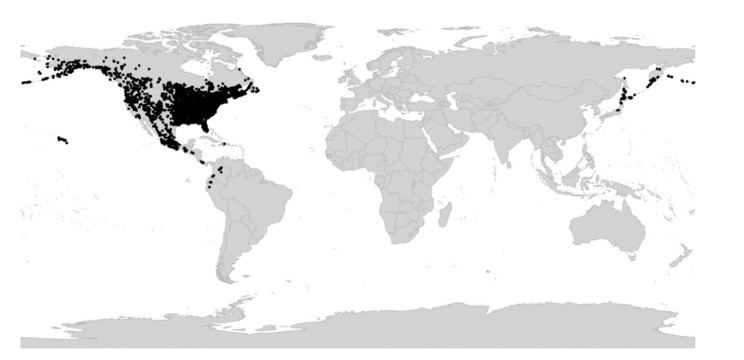
Global distribution of *Viola* sect. *Nosphinium*.

**Figure 24 plants-11-02224-f024:**
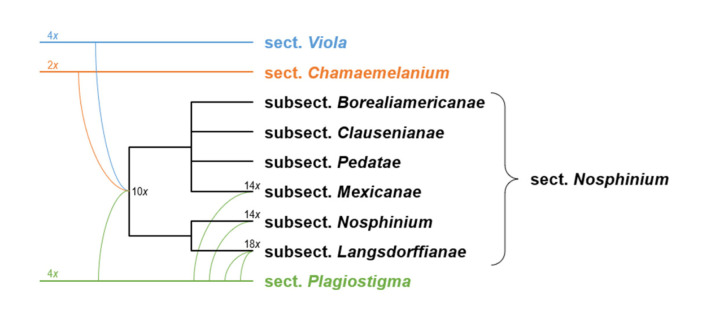
Reticulate allopolyploid phylogeny of *Viola* sect. *Nosphinium*, simplified from Marcussen et al. [45]. Allopolyploidisations are indicated by curved lines. Ploidy (*x*) is indicated.

**Figure 25 plants-11-02224-f025:**
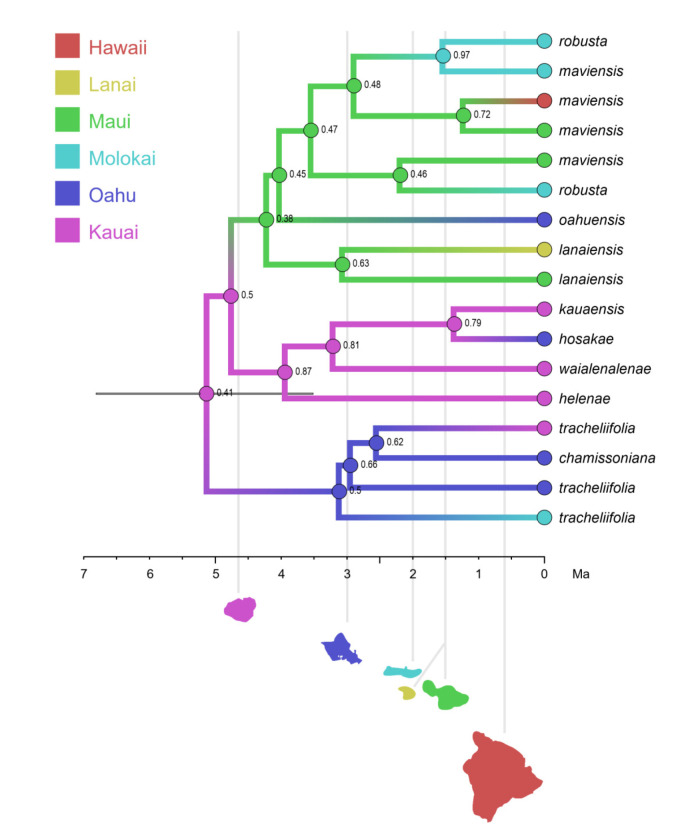
Historical biogeography of the Hawaiian violets, *Viola* subsect. *Nosphinium*, based on ITS sequences and simultaneous estimation of phylogeny and discrete biogeography using BEAST. Kauai is indicated as the most likely island of colonisation, based both on age, which excludes all the other islands, and on receiving the highest posterior probability (*pp*) by ancestral state reconstruction. The 95% credibility interval for node age is shown as a node bar on the crown node. Ancestral states are colour-coded according to island and indicated on each node along with the *pp*. Each island is shown as a silhouette and its age is indicated by a vertical line. Outgroups have been trimmed.

**Figure 26 plants-11-02224-f026:**
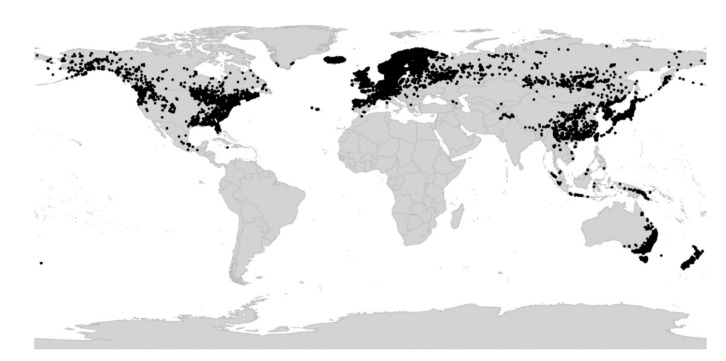
Global distribution of *Viola* sect. *Plagiostigma*.

**Figure 27 plants-11-02224-f027:**
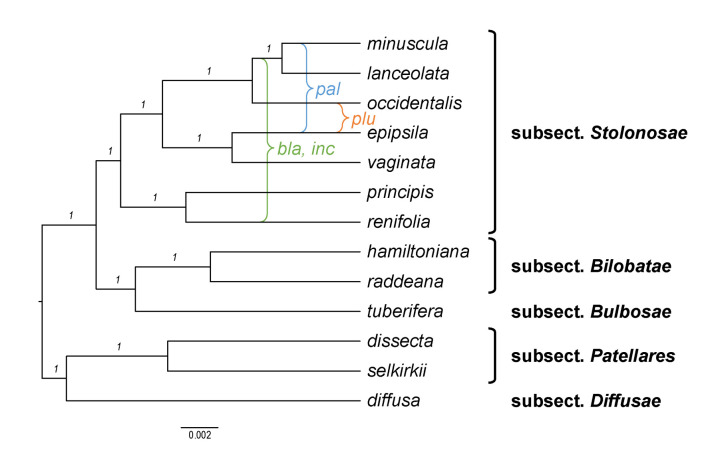
Phylogeny of *Viola* sect. *Plagiostigma* showing the delimitation of subsections (4*x*) with known allopolyploids (8*x*) superimposed, based on concatenated sequences of eight nuclear gene loci (*GPI-C*, *GPI-M*, *ITS-C*, *ITS-M*, *NRPD2a-C*, *NRPD2a-M*, *SDH-C*, and *SDH-M*). Outgroups have been pruned. The relative ages for polyploids are not to scale. Branch support is given as posterior probabilities. Abbreviations: bla, inc = *V. blanda* and *V. incognita*; pal = *V. palustris*; plu = *V. pluviae*.

**Figure 28 plants-11-02224-f028:**
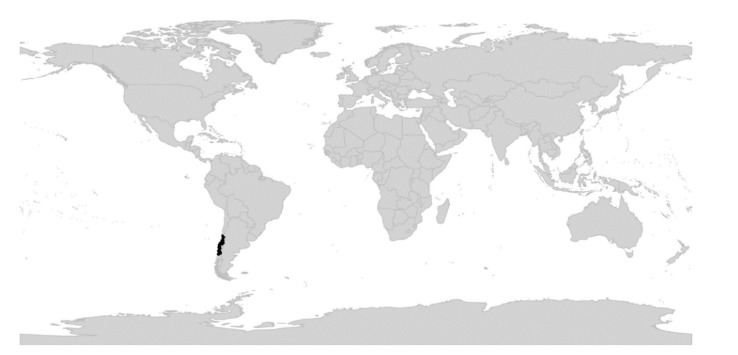
Global distribution of *Viola* sect. *Rubellium*.

**Figure 29 plants-11-02224-f029:**
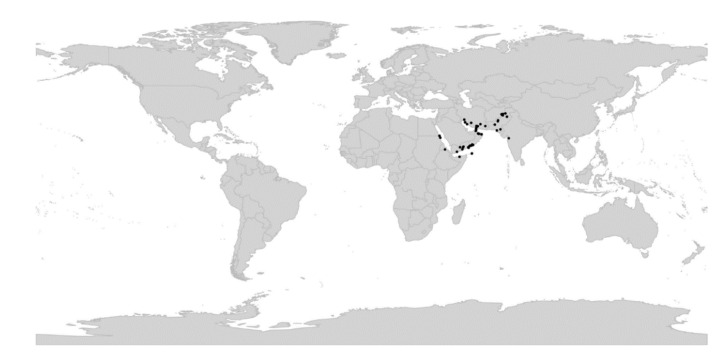
Global distribution of *Viola* sect. *Sclerosium*.

**Figure 30 plants-11-02224-f030:**
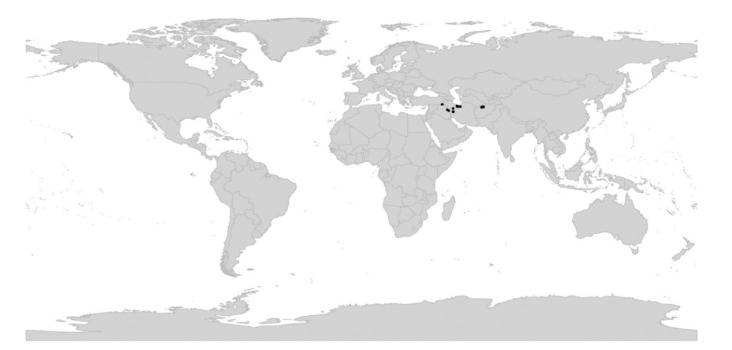
Global distribution of *Viola* sect. *Spathulidium*.

**Figure 31 plants-11-02224-f031:**
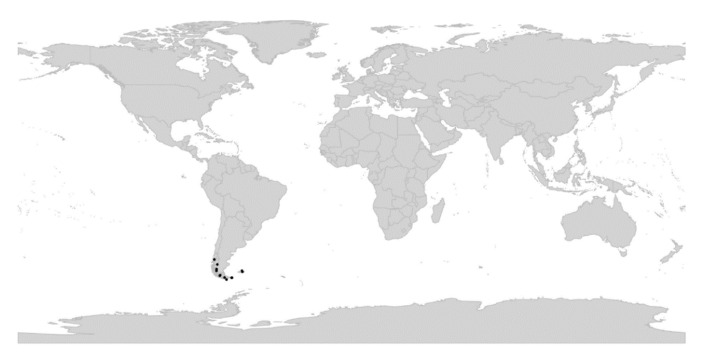
Global distribution of *Viola* sect. *Tridens*.

**Figure 32 plants-11-02224-f032:**
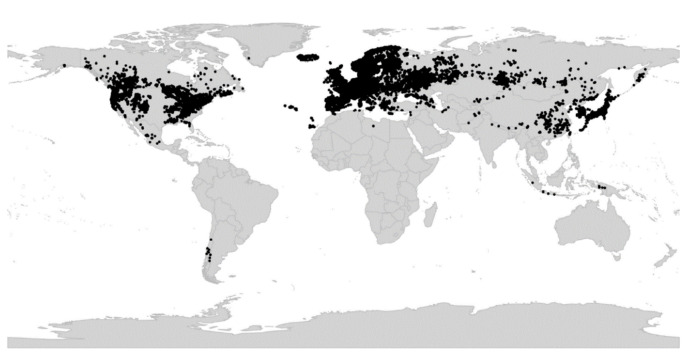
Global distribution of *Viola* sect. *Viola*.

**Figure 33 plants-11-02224-f033:**
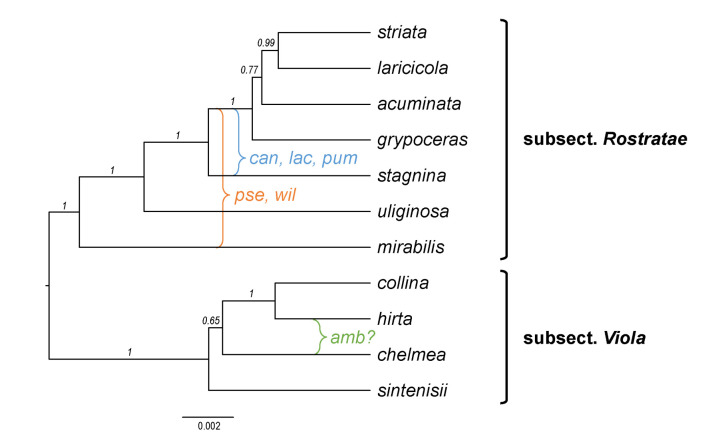
Phylogeny of *Viola* sect. *Viola* showing the delimitation of subsections (4*x*) with known allopolyploids (8*x*) superimposed, based on concatenated sequences of eight nuclear gene loci (*GPI-C*, *GPI-M*, *ITS-C*, *ITS-M*, *NRPD2a-C*, *NRPD2a-M*, *SDH-C*, and *SDH-M*). Outgroups have been pruned. The ages and placements for polyploids are not to scale. Branch support is given as posterior probabilities. Abbreviations: amb = *V. ambigua*; can, lac, pum = *V. canina*, *V. lactea*, and *V. pumila*; pse, wil = *V. pseudomirabilis* and *V. willkommii*.

**Figure 34 plants-11-02224-f034:**
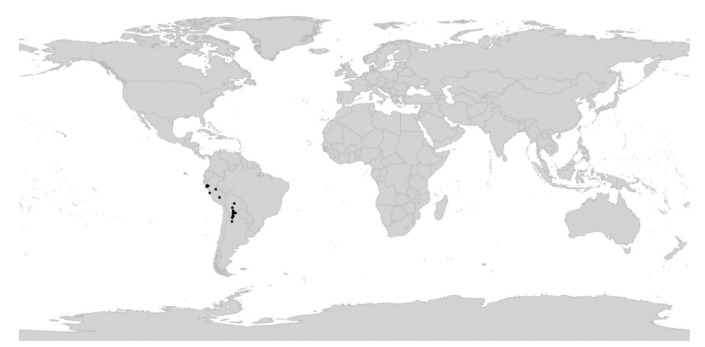
Global distribution of *Viola* sect. *Xanthidium*.

**Figure 35 plants-11-02224-f035:**
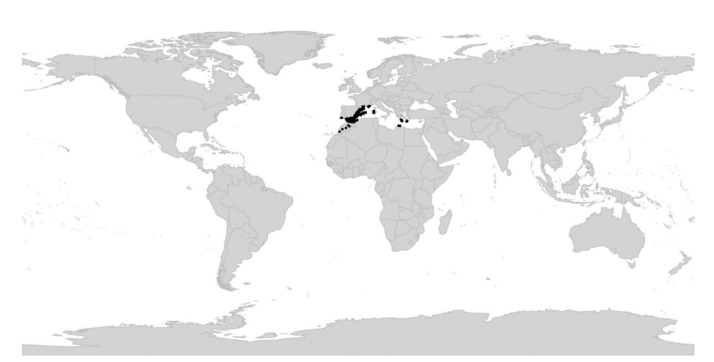
Global distribution of *Viola* sect. *Xylinosium*.

**Table 1 plants-11-02224-t001:** The proposed infrageneric classification of the 664 recognised species of *Viola* into two subgenera, 31 sections and 20 subsections. “-“ indicates missing data. “?” indicates unknown state. Chromosome numbers within square brackets indicate expected numbers based on ploidy and ancestry.

Genus Segregate	Figure	Ploidy (*x*)	Base 2*n*	Species	Distribution	Type Species
**Subg. *Neoandinium***	**-**	**2*x***	**14**	**139**	**South America**	** *Viola pygmaea* **
sect. *Confertae*	3a	-	-	1	Chile?	*V. nassauvioides*
sect. *Ericoidium*	3b	-	-	1	Argentina, Chile	*V. fluehmannii*
sect. *Grandiflos*	3c	-	-	6	Argentina, Chile	*V. truncata*
sect. *Inconspicuiflos*	3d–e	-	-	8	Peru	*V. lilliputana*
sect. *Relictium*	3e	-	-	8	Chile	*V. huesoensis*
sect. *Rhizomandinium*	3g	-	-	2	Argentina	*V. escondidaensis*
sect. *Rosulatae*	3h	2*x*	14	55	South America	*V. rosulata*
sect. *Sempervivum*	3i	-	-	34	South America	*V. atropurpurea*
sect. *Subandinium*	3j	2*x*	-	15	South America	*V. subandina*
sect. *Triflabellium*	3k	-	-	7	Argentina, Chile	*V. triflabellata*
sect. *Xylobasis*	3l	-	-	1	Argentina	*V. beati*
**Subg. *Viola***	**-**	**2*x***	**14?**	**525**	**cosmopolitan**	** *V. odorata* **
sect. *Abyssinium*	3m	12*x*	c. 72	3	Africa	*V. abyssinica*
sect. *Chamaemelanium*	3n	2*x*	12	69	northern hemisphere	*V. canadensis*
sect. *Chilenium*	3o	≥4x	-	7	South America	*V. maculata*
sect. *Danxiaviola*	3p	4*x*?	20	1	China: Guangdong	*V. hybanthoides*
sect. *Delphiniopsis*	3q	4*x*	20	3	southern Europe	*V. delphinantha*
sect. *Erpetion*	3r	8*x*	50	11	Australia	*V. hederacea*
sect. *Himalayum*	3s	8*x*	20?	1	central Asia	*V. kunawurensis*
sect. *Leptidium*	3t	4*x*	54	18	Latin America	*V. stipularis*
sect. *Melanium*	-	4*x*	?	112	northern hemisphere	*V. tricolor*
—subsect. *Bracteolatae*	3u	8*x*	?	98	western Palearctic	*V. tricolor*
—subsect. *Cleistogamae*	3v	8*x*	34	1	eastern North America	*V. rafinesquei*
—subsect. *Dispares*	3w	4*x*?	?	3	Mediterranean region	*V. dyris*
—subsect. *Ebracteatae*	3x	4*x*	?	9	Mediterranean region	*V. modesta*
—subsect. *Pseudorupestres*	3y	4*x*?	14	1	Alps & Corsica	*V. nummulariifolia*
sect. *Melvio*	3z	6*x*?	-	1	South Africa	*V. decumbens*
sect. *Nematocaulon*	3aa	-	72	1	New Zealand	*V. filicaulis*
sect. *Nosphinium*	-	10*x*	[56]	62	mainly North America	*V. chamissoniana*
—subsect. *Borealiamericanae*	3ab	10x	54	38	North America	*V. cucullata*
—subsect. *Clausenianae*	3ac	10*x*	c. 44?	1	USA: Utah	*V. clauseniana*
—subsect. *Langsdorffianae*	3ad	14*x*	[80]	3	Amphiberingian	*V. langsdorffii*
—subsect. *Mexicanae*	3ae	14*x*	[80]	10	Mexico to Ecuador and Venezuela	*V. humilis*
—subsect. *Nosphinium*	3af	14*x*	[80]	9	Hawaiian Islands	*V. chamissoniana*
—subsect. *Pedatae*	3ag	10*x*	54	1	eastern North America	*V. pedata*
sect. *Plagiostigma*	-	4*x*	24	142	cosmopolite except Africa	*V. palustris*
—subsect. *Australasiaticae*	3ah	8*x*?	46?	10	southeastern Asia and Malesia	*V. sumatrana*
—subsect. *Bilobatae*	3ai	4*x*	24	8	eastern Asia and Australasia	*V. arcuata*
—subsect. *Bulbosae*	3aj	4*x*	24	2	eastern Himalayas and central China	*V. bulbosa*
—subsect. *Diffusae*	3ak	4*x*	24	17	southeastern Asia	*V. diffusa*
—subsect. *Formosanae*	3al	4*x*	22	2	Taiwan and Okinawa	*V. formosana*
—subsect. *Patellares*	3am	4*x*	24	62	northern hemisphere	*V. selkirkii*
—subsect. *Stolonosae*	3an	4*x*	24	41	northern hemisphere	*V. palustris*
sect. *Rubellium*	3ao	2*x*	12	3	Chile	*V. rubella*
sect. *Sclerosium*	3ap	4*x*	22	7	Indian Ocean monsoon region	*V. cinerea*
sect. *Spathulidium*	3aq	8*x*	-	3	Iraq, Iran, Afganistan	*V. spathulata*
sect. *Tridens*	3ar	6*x*	40	1	southern South America	*V. tridentata*
sect. *Viola*	-	4*x*	20	75	near cosmopolite	*V. odorata*
—subsect. *Rostratae*	3as	4*x*	20	51	near cosmopolite	*V. riviniana*
—subsect. *Viola*	3at	4*x*	20	24	Palearctic	*V. odorata*
sect. *Xanthidium*	3av	-	-	2	central South America	*V. flavicans*
sect. *Xylinosium*	3aw	8*x*?	52	3	Mediterranean region	*V. arborescens*

**Table 2 plants-11-02224-t002:** Inferred ploidy for 11 species of *Viola* sect. *Melanium* based on *GPI* homoeolog number [28] and estimated genome size as gigabases (Gb) and 1C. Genome size data were downloaded from the Plant DNA C-values Database [253] and Genomes on a Tree (GoaT; accessed on 10 March 2022), which presents genome-relevant metadata for Eukaryotic taxa to be sequenced by the Earth Biogenome Project [233]. “-” indicates missing data.

Species	Inferred Ploidy	2*n* =	*GPI* Homoeologs	Genome Size (Gb)	Genome Size (1C, pg)	Subsect.
*V. dirimliensis*	4*x*	8	2	-	1.07 (herein)	*Ebracteatae*
*V. rafinesquei*	8*x*	34	4	-	-	*Cleistogamae*
*V. beckiana*	8*x*	20	-	1.32	1.35 [219]	*Bracteolatae*
*V. cornuta*	8*x*	22	-	1.25	1.18 [221]	*Bracteolatae*
*V. elegantula*	8*x*	20	-	1.32	1.35 [220]	*Bracteolatae*
*V. kitaibeliana*	8*x*	16	-	-	1.10 [218]	*Bracteolatae*
*V. tricolor*	12*x*	26	6 [236]	2.07	1.76 to 1.78 [218]	*Bracteolatae*
*V. arvensis*	16*x*	34	-	-	2.23 [218]	*Bracteolatae*
*V. calcarata*	16*x*	40	8	2.82	2.89 [221]	*Bracteolatae*
*V. lutea*	20*x*	48	-	-	2.75 [218], 3.13 [221]	*Bracteolatae*
*V. bubaniii*	20*x*	52	-	3.32	3.39 [221]	*Bracteolatae*

**Table 3 plants-11-02224-t003:** Taxonomic characters delimiting subsections within *Viola* sect. *Melanium*.

Character\Subsection	Subsect. *Pseudorupestres*	Subsect. *Ebracteatae*	Subsect. *Dispares*	Subsect. *Cleistogamae*	Subsect. *Bracteolatae*
Life history/durancy	perennial	annual	annual or perennial	annual to biennial	annual to perennial
Cleistogamous flowers	not produced	not produced	not produced	produced	not produced
Lamina of basal leaves	entire	entire or subcrenate	entire or crenate	crenate	entire or crenate
Colour of corolla throat	cream	bright yellow	bright yellow	bright yellow	bright yellow, rarely cream
Ploidy	probably 4*x*	4*x*, 8*x*, >8*x*	probably 4*x*, 8*x*	8*x*	8*x*, 12*x*, 16*x*, 20*x*
Bottom petal length, spur included (mm)	9.5–11.5	2–11.5	5–13	8–10	5.5–34
Spur length (mm)	2.3–2.5	0.9–3	1–3.5	1–1.5	1.8–16
Calycine appendage length (mm)	1.2–1.5	0.5–5.3	0.3–1	0.5–2	0.9–4.7
Pollen apertures	3	3 or heteromorphic 4	3 or 4	4	4 or 5 heteromorphic

**Table 4 plants-11-02224-t004:** GenBank sequence IDs for the *ITS* sequences used in the phylogeny in Figure 6. Taxon names have been edited to correspond to the nomenclature used herein. Metadata for each sequence are available at GenBank/NCBI.

Infrageneric Classification	Species	GenBank Sequence IDs
sect. *Rosulatae*	*Viola philippii*	MH792062
sect. *Sempervivum*	*V. cotyledon*	ON133602
sect. *Sempervivum*	*V. micranthella*	AF097222, AF097268
sect. *Subandinium*	*V. subandina*	MH781265
sect. *Subandinium*	*V. yrameae, ined.*	ON133601 (as *Viola* sp. MVN-2022a)
sect. *Abyssinium*	*V. abyssinica*	MN723993
sect. *Chamaemelanium*	*V. biflora*	DQ055348
sect. *Chamaemelanium*	*V. canadensis*	AF097231, MG234951
sect. *Chamaemelanium*	*V. pubescens*	DQ006044
sect. *Chamaemelanium*	*V. sempervirens*	MG235908
sect. *Chamaemelanium*	*V. sheltonii*	AF097226, AF097272
sect. *Chamaemelanium*	*V. uniflora*	AY582167, AY541600
sect. *Chamaemelanium*	*V. urophylla*	MH117805
sect. *Chilenium*	*V. reichei*	AF097223, AF097269
sect. *Danxiaviola*	*V. hybanthoides*	KF011244 (as *Viola* sp. LWB-2013a)
sect. *Delphiniopsis*	*V. cazorlensis*	AY148230, AY148250
sect. *Himalayum*	*V. kunawurensis*	NCBI accession PRJNA805692 (as *V. “kunawarensis”*)
sect. *Leptidium*	*V. scandens*	AF097221, AF097267
sect. *Melanium* subsect. *Bracteolatae*	*V. cornuta*	AY582166, MT367013
sect. *Melanium* subsect. *Bracteolatae*	*V. heldreichiana*	MT367025
sect. *Melanium* subsect. *Bracteolatae*	*V. kitaibeliana*	AY148235, KX166474, MT367029
sect. *Melanium* subsect. *Bracteolatae*	*V. paradoxa*	MT367093
sect. *Melanium* subsect. *Bracteolatae*	*V. tricolor*	DQ055396
sect. *Melanium* subsect. *Cleistogamae*	*V. rafinesquei*	MG235080 (as *V. bicolor*)
sect. *Melanium* subsect. *Dispares*	*V. demetria*	MT367018
sect. *Melanium* subsect. *Dispares*	*V. dyris*	MT367069
sect. *Melanium* subsect. *Ebracteatae*	*V. dirimliensis*	ON129460
sect. *Melanium* subsect. *Ebracteatae*	*V. mercurii*	MT367115
sect. *Melanium* subsect. *Ebracteatae*	*V. modesta*	MT367084
sect. *Melanium* subsect. *Ebracteatae*	*V. occulta*	HM851453
sect. *Melanium* subsect. *Ebracteatae*	*V. parvula*	AY148240, AY148260
sect. *Melanium* subsect. *Pseudorupestres*	*V. nummulariifolia*	MT367090
sect. *Nosphinium* subsect. *Borealiamericanae*	*V. affinis*	AF097251, AF097297
sect. *Nosphinium* subsect. *Borealiamericanae*	*V. cucullata*	AF097252, MG237103
sect. *Nosphinium* subsect. *Clausenianae*	*V. clauseniana*	AF097300, AF097254
sect. *Nosphinium* subsect. *Langsdorffianae*	*V. langsdorffii*	AF097259, MG235517
sect. *Nosphinium* subsect. *Mexicanae*	*V. hemsleyana*	AF097258, AF097304
sect. *Nosphinium* subsect. *Mexicanae*	*V. hookeriana*	AF097257, AF097303
sect. *Nosphinium* subsect. *Mexicanae*	*V. nannei*	AF097255, AF097301
sect. *Nosphinium* subsect. *Nosphinium*	*V. chamissoniana*	AF115955, AF115959
sect. *Nosphinium* subsect. *Nosphinium*	*V. lanaiensis*	JN682058
sect. *Nosphinium* subsect. *Pedatae*	*V. pedata*	AF097253, MG237117
sect. *Plagiostigma* subsect. *Australasiaticae*	*V. austrosinensis*	OM406228
sect. *Plagiostigma* subsect. *Australasiaticae*	*V. kwangtungensis*	OM406230
sect. *Plagiostigma* subsect. *Australasiaticae*	*V. mucronulifera*	FJ002910
sect. *Plagiostigma* subsect. *Australasiaticae*	*V. sumatrana*	OM406231
sect. *Plagiostigma* subsect. *Bilobatae*	*V. hamiltoniana*	AY928283 (as *V. verecunda*)
sect. *Plagiostigma* subsect. *Bilobatae*	*V. raddeana*	AY928279
sect. *Plagiostigma* subsect. *Bilobatae*	*V. triangulifolia*	FJ002912
sect. *Plagiostigma* subsect. *Diffusae*	*V. amamiana*	JF830899
sect. *Plagiostigma* subsect. *Diffusae*	*V. diffusa*	MH711723
sect. *Plagiostigma* subsect. *Diffusae*	*V. guangzhouensis*	MW683479
sect. *Plagiostigma* subsect. *Diffusae*	*V. huizhouensis*	MW683486
sect. *Plagiostigma* subsect. *Diffusae*	*V. lucens*	FJ002913
sect. *Plagiostigma* subsect. *Diffusae*	*V. nanlingensis*	FJ002916
sect. *Plagiostigma* subsect. *Diffusae*	*V. yunnanensis*	FJ002915
sect. *Plagiostigma* subsect. *Patellares*	*V. albida*	DQ787762 (as *V. chaerophylloides*)
sect. *Plagiostigma* subsect. *Patellares*	*V. dissecta*	JQ950564
sect. *Plagiostigma* subsect. *Patellares*	*V. patrinii*	AY928298
sect. *Plagiostigma* subsect. *Patellares*	*V. selkirkii*	AY928307
sect. *Plagiostigma* subsect. *Patellares*	*V. somchetica*	HM851457
sect. *Plagiostigma* subsect. *Patellares*	*V. tashiroi*	JF830885
sect. *Plagiostigma* subsect. *Patellares*	*V. variegata*	KC330743
sect. *Plagiostigma* subsect. *Stolonosae*	*V. suecica*	MG237736 (as *V. epipsila*)
sect. *Plagiostigma* subsect. *Stolonosae*	*V. grandisepala*	FJ002903
sect. *Plagiostigma* subsect. *Stolonosae*	*V. lanceolata*	MG235616
sect. *Plagiostigma* subsect. *Stolonosae*	*V. minuscula*	AF097236, AF097282 (as *V*. *macloskeyi* subsp. *pallens*)
sect. *Plagiostigma* subsect. *Stolonosae*	*V. moupinensis*	FJ002900
sect. *Plagiostigma* subsect. *Stolonosae*	*V. palustris*	KX166144
sect. *Plagiostigma* subsect. *Stolonosae*	*V. principis*	FJ002904
sect. *Plagiostigma* subsect. *Stolonosae*	*V. yazawana*	AY928289
sect. *Rubellium*	*V. capillaris*	AF097220, AF097266
sect. *Spathulidium*	*V. spathulata*	HM851456
sect. *Viola* subsect. *Rostratae*	*V. acuminata*	AY928273
sect. *Viola* subsect. *Rostratae*	*V. grypoceras*	AY928280
sect. *Viola* subsect. *Rostratae*	*V. mirabilis*	MK828560
sect. *Viola* subsect. *Rostratae*	*V. reichenbachiana*	DQ055382
sect. *Viola* subsect. *Rostratae*	*V. shinchikuensis*	FJ002885
sect. *Viola* subsect. *Rostratae*	*V. stagnina*	KX166475
sect. *Viola* subsect. *Rostratae*	*V. striata*	AF097247, MG234688
sect. *Viola* subsect. *Rostratae*	*V. uliginosa*	KU949386
sect. *Viola* subsect. *Rostratae*	*V. umbraticola*	AF097244, AF097290
sect. *Viola* subsect. *Rostratae*	*V. websteri*	AY928274
sect. *Viola* subsect. *Viola*	*V. alba*	EU413916
sect. *Viola* subsect. *Viola*	*V. hirta*	EU413946
sect. *Viola* subsect. *Viola*	*V. hondoensis*	AY928272
sect. *Viola* subsect. *Viola*	*V. odorata*	EU413922
sect. *Viola* subsect. *Viola*	*V. pyrenaica*	JF683824
sect. *Xylinosium*	*V. scorpiuroides*	MT367099
Outgroup	*Melicytus obovatus*	EF635462

**Table 5 plants-11-02224-t005:** Taxa and GenBank sequence IDs for the phylogenetic analysis of sect. *Chamaemelanium*. A dash indicates missing data. Metadata for each sequence are available at GenBank/NCBI.

Species	*GPI*	*ITS*	*NRPD2a*	*trnL-trnF*
*Viola barroetana*	-	AF097224, AF097270	-	-
*V. biflora*	JF767023	AY928309	GU289574	JF767165
*V. brevistipulata*	JF767032	AY928275	GU289575	JF767167
*V. canadensis*	JF767034	AF097231, AF097277	GU289576	JF767163
*V. delavayi*	-	FJ002908	-	-
*V. fischeri*	-	AY582168, AY541601	-	-
*V. flagelliformis*	-	AF097233, AF097279	-	-
*V. lobata*	JF767080	-	-	JF767161
*V. orientalis*	-	AY541602, AY582169	-	DQ085929
*V. pubescens*	JF767117	DQ006044	GU289580	JF767162
*V. purpurea*	JF767118	MG235177	KJ138061	JF767160
*V. rotundifolia*	JF767122	AF097241, AF097287	KJ138062	JF767168
*V. schulzeana*	-	FJ002907	-	-
*V. sheltonii*	JF767130	AF097226, AF097272	KJ138070	JF767159
*V. tomentosa*	JN620193	-	-	JN620205
*V. tripartita*	OP256029	OP256030	-	-
*V. uniflora*	JF767146	AY582167, AY541600	KJ138083	JF767166
Outgroup: *V. congesta*	JF767046	MH781265 (*V. subandina*)	GU289564	JF767154
Outgroup: *V. capillaris*	JF767035	AF097220, AF097266	KJ138036	JF767156

**Table 6 plants-11-02224-t006:** Taxa, island biogeography, and GenBank sequence IDs of ITS1 and ITS2 for the combined dating and biogeographic analysis of subsect. *Nosphinium*. Biogeography for the outgroup was not coded and was entered as missing (“?”) in the analysis. Metadata for each sequence are available at GenBank/NCBI.

Species	Biogeography	GenBank Sequence IDs
*Viola chamissoniana*	Oahu	AF115955, AF115959
*V. helenae*	Kauai	AF097260, AF097306
*V. kauaensis* var. *hosakae*	Oahu	AF115957, AF115961
*V. kauaensis* var. *kauaensis*	Kauai	AF097262, AF097308
*V. lanaiensis*	Lanai	FJ895310, FJ895319
*V. lanaiensis*	Maui	JN682058
*V. maviensis*	Maui	AF097263, AF097309
*V. maviensis*	Molokai	FJ895311, FJ895320
*V. maviensis*	Maui	FJ895312, FJ895321
*V. maviensis*	Hawaii	FJ895313, FJ895322
*V. oahuensis*	Oahu	FJ895314, FJ895323
*V. robusta*	Molokai	AF115956, AF115960
*V. robusta*	Molokai	FJ895315, FJ895324
*V. tracheliifolia*	Kauai	AF097261, AF097307
*V. tracheliifolia*	Oahu	FJ895316, FJ895325
*V. tracheliifolia*	Molokai	FJ895317, FJ895326
*V. waialenalenae*	Kauai	AF115958, AF115962
Outgroup: *V. selkirkii*	?	AY928307
Outgroup: *V. spathulata*	?	HM851456
Outgroup: *V. langsdorffii*	?	AF097259, AF097305
Outgroup: *V. langsdorffii*	?	FJ895309, FJ895318
Outgroup: *V. mirabilis*	?	DQ358858, DQ358835
Outgroup: *V. nannei*	?	AF097255, AF097301
Outgroup: *V. odorata*	?	EU413918
Outgroup: *V. pedata*	?	AF097253, AF097299
Outgroup: *V. reichenbachiana*	?	DQ055382

**Table 7 plants-11-02224-t007:** Taxa and GenBank sequence IDs for the combined phylogenetic analysis of sect. *Plagiostigma* and sect. *Viola* (Figure 27 and Figure 33). A dash indicates missing data. “ibid.” is used for taxa with genomic data whose loci have the same GenBank ID (given under *GPI-C*). Taxon names have been edited to correspond to the nomenclature used herein. Metadata for each sequence are available at GenBank/NCBI.

Species	*GPI-C*	*GPI-M*	*ITS-C*	*ITS-M*	*NRPD2a-C*	*NRPD2a-M*	*SDH-C*	*SDH-M*
**Sect. *Plagiostigma***								
*V. dissecta*	SRX11632715	ibid.	ibid.	-	ibid.	ibid.	ibid.	-
*V. diffusa*	JF767047	JF767048	GQ434456	-	KJ138043	KJ138044	KJ138112	KJ138113
*V. epipsila*	JF767049	JF767050	MG237736	-	GU289587	GU289588	KJ138115	KJ138116
*V. hamiltoniana*	JF767150	JF767151	AY928283	-	GU289591	GU289592	-	KJ138153
*V. lanceolata*	JF767069	JF767070	MG235616	-	KJ138051	KJ138052	KJ138119	-
*V. minuscula*	JF767089	JF767090	AF097236, AF097282	-	-	-	-	-
*V. occidentalis*	JF767088	JF767087	-	-	OP256031 ^1^	OP256032 ^1^	OP256033 ^1^	-
*V. principis*	JF767115	JF767116	FJ002904	-	KJ138059	KJ138060	KJ138128	-
*V. raddeana*	SRX9916745	ibid.	ibid.	-	ibid.	ibid.	ibid.	ibid.
*V. renifolia*	JF767120	JF767121	JN999695	-	-	-	-	-
*V. selkirkii*	JF767128	JF767129	MG234698	-	GU289590	GU289589	KJ138143	KJ138144
*V. tuberifera*	JF767142	JF767143	-	-	OP256034 ^1^	OP256035 ^1^	OP256036 ^1^	OP256037 ^1^
*V. vaginata*	JF767148	JF767149	-	-	OP256038 ^1^	OP256039 ^1^	OP256040 ^1^	OP256041 ^1^
**Sect. *Viola***								
*V. acuminata*	SRX11632718	ibid.	-	ibid.	ibid.	ibid.	ibid.	ibid.
*V. chelmea*	JF767036	JF767037	-	-	KU949390	KU949396	KU949402	KU949407
*V. collina*	JF767044	JF767045	-	EU413938	KU949389	KU949395	KU949401	KU949406
*V. grypoceras*	SRX14846970	ibid.	-	ibid.	ibid.	ibid.	ibid.	ibid.
*V. hirta*	JF767065	JF767066	-	DQ358856, DQ358833	GU289581	GU289582	KJ138117	KJ138118
*V. laricicola*	JF767078	JF767079	-	-	KU949387	KU949393	KU949399	KU949404
*V. mirabilis*	JF767085	JF767086	-	MK828558	GU289583	GU289584	KJ138120	KJ138121
*V. sintenisii*	-	-	-	DQ358859, DQ358836	KU949391	KU949397	-	-
*V. stagnina*	JF767133	JF767134	-	KX166475	-	KU949392	KU949398	KU949403
*V. striata*	JF767135	JF767136	-	AF097247, AF097293	KU949388	KU949394	KU949400	KU949405
*V. uliginosa*	JF767144	JF767145	-	KU949386	GU289585	GU289586	KJ138151	KJ138152
**Outgroups**								
*V. congesta*	JF767046	JF767046	MH781265	MH781265	GU289564	GU289564	KJ138104	KJ138104
*V. capillaris*	JF767035	JF767035	AF097220, AF097266	AF097220, AF097266	KJ138036	KJ138036	KJ138135	KJ138135

^1^ Sequences published herein. Metadata are available in [45].

## Data Availability

The data (scripts and analysis files) presented in this study are available in the Appendix A.

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
