# Peer review of "A Revised Phylogenetic Classification for Viola (Violaceae)"

_plants, 2022, doi:10.3390/plants11172224_

Round 1
Reviewer 1 Report
This is a very complete work, which attempts to cover all aspects of the morphology, phylogeny and distribution of the genus Viola. However, there are some questions to be reviewed, as there are aspects that have not yet been studied, so that it would be premature to make a review of this genus.
Table of contents
The table of contents should change, and so should the organisation of the work:
1. Introduction
2. Results and discussion
2.1. Phylogeny and classification
2.1.1. Genus phylogeny
2.1.2. Justification for taxonomic levels and classification
2.1.3. Changes to Becker’s original system for Viola
2.2. Patterns of evolution within Viola
2.2.1. Historical biogeography of Viola
2.2.2. Hybridisation and allopolyploidy
2.2.3. Base chromosome numbers in Viola
2.2.4. Morphology, anatomy, adaptations
2.2.5. Biotic interactions
2.2.6. Fossil record of Viola
2.3. The “known unknowns”: outstanding research in Viola
2.3.1. Phylogeny of Viola
2.3.2. Chromosome counts and ploidy
2.3.3. Fossil record
2.3.4. Alpha taxonomy
2.3.5. Transcriptomes and genomes
2.4. Taxonomic treatment of Viola
3. Materials and Methods
Some comments on the origin of Viola. Lines 500-504, 815-818, 833-846
The authors hypothesise, but do not prove, that Viola originated in South America. This is contradicted by the large number of species in the Northern Hemisphere. They also indicate that there are lineages that dispersed to the Northern Hemisphere 20-25 Ma ago. If the formation of the Central American corridor did not culminate until the Pliocene (Ibaraki 1997, Rowley & Garzione 2007), how could this have happened?
If Viola's ancestors appeared in Eurasia 15 Ma ago, how did Viola originate in South America 25 Ma ago, If there is no certainty that the Central American corridor was closed?
The fact that there are neither pollen nor seed fossils of Viola in South America is very significant. Whereas in Eurasia there are many fossils indicating that the genus was already widely dispersed in the Miocene.
Therefore, the origin of Viola needs to be clarified or at least an in-depth commentary with the relationship of Viola dispersion to geological aspects of the uplift of the Andean Cordillera and the formation of the Central American corridor.
For the lines 922-926, and also 929-934:
If there is a lack of knowledge about the phylogeny of Viola and DNA sequences are of poor resolution, how can results on its phylogeny and geographical distribution be provided?
Line 1415 and others:
In which herbaria are the species types?
Material and methods
I think that a work of this kind should be based on bibliography, databases... but also on herbarium sheets. Which herbaria have been consulted, in which herbaria are the types of the species cited? Are the species typified?
Line 3893: Which herbaria have been consulted? For the online images, it’s necessary to take into account that images are not conducive to solving many taxonomic problems; they can complement the study of herbarium sheets. For example, the use of databases to verify species is scientifically erroneous if they are not based on herbarium sheets. There are species with such similar characters that it is very doubtful to verify them with images.
From what type of material have the DNA sequences been obtained? From herbarium sheets? From which herbarium sheets? From fresh material? Where does this fresh material come from? Where is it kept? These questions come from the fact that in the text, it says that the authors have used GenBank, but they have also used PCR.
Appendix A
Accepted species should be accompanied by a reference to the types and the herbaria where they are kept.
Small remarks
Line 27, 28 and others
It is common to see the abbreviation 'subgen.' in floristic works, but not 'subg.'. If you use the abbreviation 'gen.' for the genus, you should also use 'subgen.'.
Line 137: Please include the word 'gibbose' without separation.
Line 488: Set Viola in italics.
Line 753: Delete ‘all’ once
Line 3523-3524: Put the species names in italics.
Reviewer 2 Report
This paper is of exceptional quality, made by authors, most of whom have studied and are well acquainted with the Viola genus and who have carefully pondered the affirmations and taxonomic framework they propose. the article is a sum of the knowledge on the genus on a world scale, based both on the traditional morphological taxonomy and on all the knowledge acquired with the molecular approach. I think it will be a point of reference on this genus for a long time.
I therefore believe that the article can be published on Plants in its current form.
Author Response
We wish to thank Reviewer 2 for her/his overwhelmingly positive response. The reviewer did not suggest any points to improve.